

# Observational data of Arctic Sea Ice Melt Ponds: a Systematic Review of Acquisition and Processing Approaches

Sara Aparício[1], Simon Driscoll[2], Daniela Flocco[3]

[1]CENSE - Center for Environmental and Sustainability Research, NOVA School of Science and Technology, NOVA University, Lisbon, Portugal

[2]Department of Applied Mathematics and Theoretical Physics, University of Cambridge, Cambridge, Cambridgeshire, UK

[3]Dipartimento di Scienze della Terra, dell'Ambiente e delle Risorse, Università degli Studi di Napoli "Federico II", Napoli, Italy

*Correspondence to*: Sara Aparício (s.aparicio@campus.fct.unl.pt)

**Abstract.** This review synthesizes current methods for acquiring and processing Earth observation (EO) data relevant to Arctic sea ice melt ponds (MPs), pools of meltwater that form on the ice surface during the polar summer. By reducing albedo, MPs amplify the ice–albedo feedback and alter the sea ice energy budget, exerting a strong influence on the Arctic climate system. Robust observational records are therefore essential for improving sea ice prediction in a rapidly changing and highly sensitive polar environment. Despite this importance, melt pond parameterizations remain underdeveloped in many sea ice models. Advancing these parameterizations, through refinement of existing schemes and integration of novel approaches, is a critical priority for better constraining sea ice evolution and its role in the climate system.

Here we review the main EO methods used in MP studies, including active and passive optical sensors (multispectral and LiDAR) and microwave instruments (synthetic aperture radar, radiometers, and scatterometers). We also summarize melt pond signatures across the electromagnetic spectrum, outlining the strengths and limitations of each sensor. Complementary in situ observations from field campaigns, together with key processing techniques, are discussed, alongside a synthesis of available MP datasets from satellite missions and ground-based campaigns. Persistent EO data gaps, such as cloud cover, limited temporal sampling, and spatial constraints that lead to underrepresentation of different Arctic regions and ice types, remain a major challenge, highlighting the need for future missions with improved resolution, coverage, and spectral capacity.

By compiling and critically assessing these datasets and methods, and identifying current knowledge gaps, this paper provides the most comprehensive review of melt pond observations currently available. It is designed to support refinement of parameterizations and the development of multi-modal modelling approaches, crucial for addressing observational gaps and ultimately advancing the understanding and prediction of Arctic change.



## 1. Introduction

The Arctic is one of the most vulnerable components of the Earth system to climate change, warming nearly four times faster than the globe since 1979 (Rantanen et al., 2022). The observed amplified warming, has far-reaching consequences (Previdi, et al., 2021; Shu et al., 2022) which correlates with a decline in Arctic sea ice volume, thickness and extent since the beginning of the satellite record (Stroeve et al., 2012a; Stroeve and Notz, 2018). By reflecting away a substantial part of the incoming solar radiation, sea ice acts as a regulator of the Earth's temperature. However, in the past decades not only the Arctic sea ice has reached record September minima, it is also thinning (Giles et al., 2008) with ice moving towards a regime where multi-year ice (MYI) is being replaced by first-year ice (FYI) (Sumata et al., 2023). Melt ponds (MPs) are among the most distinctive features of this rapidly evolving Arctic sea ice surface: these are pools of meltwater that form on the sea ice surface (Figure 1) from the snow and sea ice melting in Spring and Summer, as well as from rainfall accumulation during the melt season. Melt ponds can cover up to 50% of the Arctic sea ice surface with peaks of 80% of cover on flat first year ice (Flocco et al., 2015), wherein they play an extremely large role in thermodynamic sea ice process and hence playing a pivotal role in the Arctic energy budget (Fetterer and Untersteiner, 1998; Curry et al., 1995; Eicken et al., 2004).

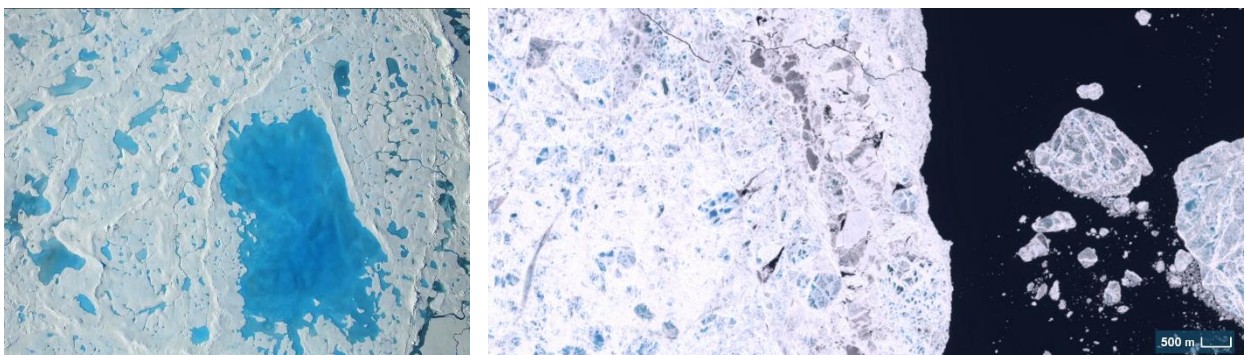

**Figure 1. (Left): Aerial photograph showing melt ponds on the surface of Arctic sea ice. Credits: Eric Fraim/NASA Operation IceBridge, July 25, 2017; (Right) Satellite image of Copernicus Sentinel-2 True colour of Arctic sea ice melt ponds (illustrating the large variability of pond color), leads and open ocean. Acquired on 17th June 2024 in the Northeastern coast of Greenland.**

The colour of melt ponds varies from light bluish to dark as it can be seen in Figure 1, largely depending on depth of the pond, type of ice (Buckley et al., 2020) and the properties of underlying ice (Lu et al., 2018) resulting in a considerable range of the value of melt pond albedo. It can vary from 0.2 over dark melt ponds to 0.4 for light melt ponds whereas the typical values for melting ice range between 0.6-0.7 (Perovich et al., 2002). Because of their low albedo, compared to snow and bare ice, melt ponds contribute to a positive ice-albedo feedback (Polashenski et al., 2012) leading to further melting. Pistone et al. (2014) found that the albedo change resulting from sea ice loss was equivalent to 25% of the direct forcing from $CO_2$ during the same period.





Numerous studies confirm that, the Arctic melt season is lengthening (Perovich et al., 2008; Markus et al., 2009; Rösel and Kaleschke, 2012; Pistone et al., 2014, Stroeve et al., 2014), suggesting an increase of the impact that the presence of melt ponds have in earlier months of the melting season. This is linked also to a shift in the Arctic sea ice spring predictability barrier, which imposes limits on regional forecasts initialised prior to spring (Bushuk et al., 2020). Several other studies, though, link melt pond state even to pre-melt season (Fuchs et al., 2025). In fact, melt ponds have a potential to predict the observed sea ice September minimum (Petty et al., 2017, Schröder et al. 2014), and it has been shown that the presence of, or improved parametrisations of melt ponds in Global Circulation Models (GCMs) can improve physical modelling of sea ice (Holland et al. 2012, Schröder et al., 2014; Liu et al., 2015), suggesting that melt ponds may provide a route through the Spring predictability barrier.

Notable uncertainty still remains: Driscoll et al. (2024a) showed how models demonstrate a substantial sensitivity to uncertain melt pond parameters. Simulated sea ice thickness over the Arctic ocean can vary by as much as a meter after only a decade of simulation by altering the melt pond parameters in a sea ice model alone, while Polashenski et al. (2012) showed that different melt pond schemes only partially capture the evolution of the observed pond coverage. It has also been suggested that discrepancies between the rates of sea ice decline presented in IPCC reports are due to fundamental missing processes in sea ice models such as those related to melting and melt pond evolution (Stroeve et al., 2007; Stroeve et al., 2012b; Bianco et al., 2024).

Numerous field campaigns (notably SHEBA and MOSAiC), have provided invaluable observations that significantly increased our insight and understanding of melt pond evolution, but single field campaigns still suffer from being local, and within a specific year, therefore being affected by the meteorological conditions of that time. Such campaigns, whilst offering high quality data are not necessarily representative of pan-Arctic evolution, and struggle to capture longer term trends in sea ice. Remote sensing data can provide consistent, spatially extensive observations of melt pond evolution across the entire Arctic basin, overcoming the spatial limitations of in situ campaigns. Satellite data such as MODIS, Sentinel-2, and ICESat-2 have been used to derive melt pond fraction, albedo variability, and surface topography at high spatial and temporal resolutions. These datasets allow for the tracking of melt pond onset, expansion, and decay over multiple years, offering insights into interannual variability and long-term trends. Furthermore, the integration of remote sensing observations into data assimilation frameworks and machine learning models holds promise for improving seasonal forecasts of sea ice extent and advancing our understanding of the role of melt ponds in modulating the Arctic surface energy budget.

A sound representation incorporating all available data sources could potentially avoid timing errors that can affect presently available parametrisations, break sea ice Spring predictability barriers, and provide sound predictions for the future of sea ice loss. Spatial and temporal gaps in data highlight a potential need for multi-modal approaches - ones that integrate a variety of diverse data sources - to improve the robustness of model physics and parametrisations, whilst furthering understanding of sea ice predictability. The increasing importance of these processes in a changing Arctic makes the case for a deeper understanding of melt ponds, including their observation, and inclusion of their parameterization in global climate models.





The objective of this literature review is to provide a systematic overview of Arctic melt ponds studies using Earth observation data. Specifically, this review aims to: (1) present a fundamental overview of melt pond observation techniques

(2) outline different strengths and limitations of various remote sensing techniques, including airborne, spaceborne, optical, radar, and microwave approaches, for sensing melt ponds across different ice types; (3) review the key findings from Earth observation studies; (4) discuss the methodological advances and challenges in processing remote sensing data, including algorithm development, data fusion, resolution, and accuracy limitations; and finally, to (5) identify knowledge gaps and research opportunities.

This literature review is primarily intended for the physical modelling and Earth observation communities and further supports data-driven research. It also provides a useful reference for climate scientists, operational forecasting centres, and policy stakeholders engaged in improving the understanding and prediction of Arctic melt pond processes, while highlighting existing gaps in EO data and methods that warrant further attention.

## 2.   Melt pond optical properties and seasonal evolution

The presence of melt ponds significantly alters the optical properties of sea ice, primarily by reducing surface albedo and enhancing solar energy absorption and accelerating ice melt (Polashenski et al., 2012; Pistone et al., 2014). Observational studies have shown that pond-covered ice can experience melt rates two to three times greater than adjacent bare ice surfaces (Fetterer and Untersteiner, 1998). This enhanced absorption reinforces the ice–albedo feedback, making surface albedo one of the most influential parameters in governing the evolution of Arctic sea ice and the regional energy budget (Dickinson et

al., 1983; Qu et al., 2015). Figure 2 illustrates the spectral dependence of albedo for various sea ice surface conditions, highlighting the pronounced reduction in reflectivity associated with melt pond development.

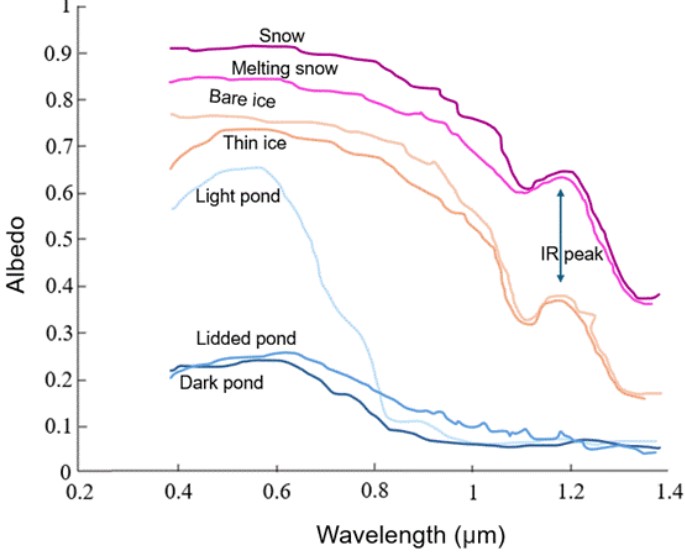

**Figure 2: Spectral albedo of different surfaces. Adapted from Light et al. (2022)**





Bare ice (light orange curve in Figure 2) has a substantial surface scattering layer, providing a stable high reflectance varying
between 0.75-0.8 in the blue-green region of the spectrum (at 450-500 nm) (Perovich et al., 2002). This highly scattering
surface forms on melting, drained sea ice during spring and summer and its scattering properties change as melt progresses
(Smith et al., 2022). The increase in snow grain size (which occurs with rising liquid water content in snow) is responsible
for reducing the surface's overall reflectivity (Warren, 1982, 2019).

Besides colour and albedo, melt ponds also exhibit substantial spatial and temporal variability in terms of shape, and depths,
and often form interconnected networks that evolve dynamically throughout the melt season (Fetterer and Untersteiner,
1998; Eicken et al., 2002). This morphological and radiometric complexity presents considerable challenges for their
detection, classification, and parameterization in both remote sensing applications and climate models (Tschudi et al., 2008;
Rösel et al., 2012). Accurate representation of melt pond characteristics is critical for improving simulations of the surface
energy balance and for better constraining ice–albedo feedback mechanisms within coupled sea ice–climate models
(Schröder et al., 2014; Liu et al., 2015).

The shape, colour, connectivity and distribution of melt ponds are primarily influenced by two main factors: meltwater
balance and ice surface features. Meltwater inflows and outflows on the sea ice surface depend on multiple variables such as
melt rate, precipitation, snow cover, pond catchment size area, sea ice type, and salinity (Eicken et al., 2004; Perovich et al.,
2009; Polashenski et al., 2012; Kim et al., 2018). Meltwater moves vertically through percolation via connected pore
structure in the ice (see Fig.3b, side view) or horizontally across the surface via cracks and brine drainage channels (see
Fig.3-a-b-c, nadir view.). Melt ponds, forming above sea level, create hydrostatic pressure that drives fresh meltwater
downward through porous sea ice, flushing out salt. Some of this meltwater refreezes within the ice, reducing porosity and
limiting drainage (Polashenski et al., 2012).

Melt ponds begin forming in late May and cover large portions of the sea ice by mid-June. They deepen and expand through
July, beginning to refreeze by late August or early September (Fetterer and Untersteiner, 1998, Rösel and Kaleschke, 2012).
Figure 3 illustrates the seasonal cycle following four stages: (a) melt onset, (b) drainage, (c) melt evolution, and (d) freeze-
up. During onset, meltwater accumulates in depressions and cracks, lowering albedo to ~0.6, and further to ~0.3 as ponds
form. In the drainage phase, increasing permeability allows percolation and drainage via macroscopic flaws, slightly raising
albedo to ~0.5. Meltwater also spreads horizontally, interconnecting ponds. This interconnection follows predictable scaling
patterns, with pond fractal dimension increasing around a critical area of 100 m², as smaller ponds coalesce to form large
connected regions with boundaries resembling space-filling curves for ponds larger than 1000 m² (Hohenegger et al., 2012).
This third stage marks the seasonal peak in pond coverage, with mature ponds showing albedo below 0.3 (Polashenski et al.,
2012). Refreezing begins in the final stage but can occur intermittently throughout the season, forming ice lids that halt
inflow and reduce albedo effects which sometimes is masked further by snowfall (Grenfell and Perovich, 2004; Flocco et al.,
2015; Anhaus et al., 2021).




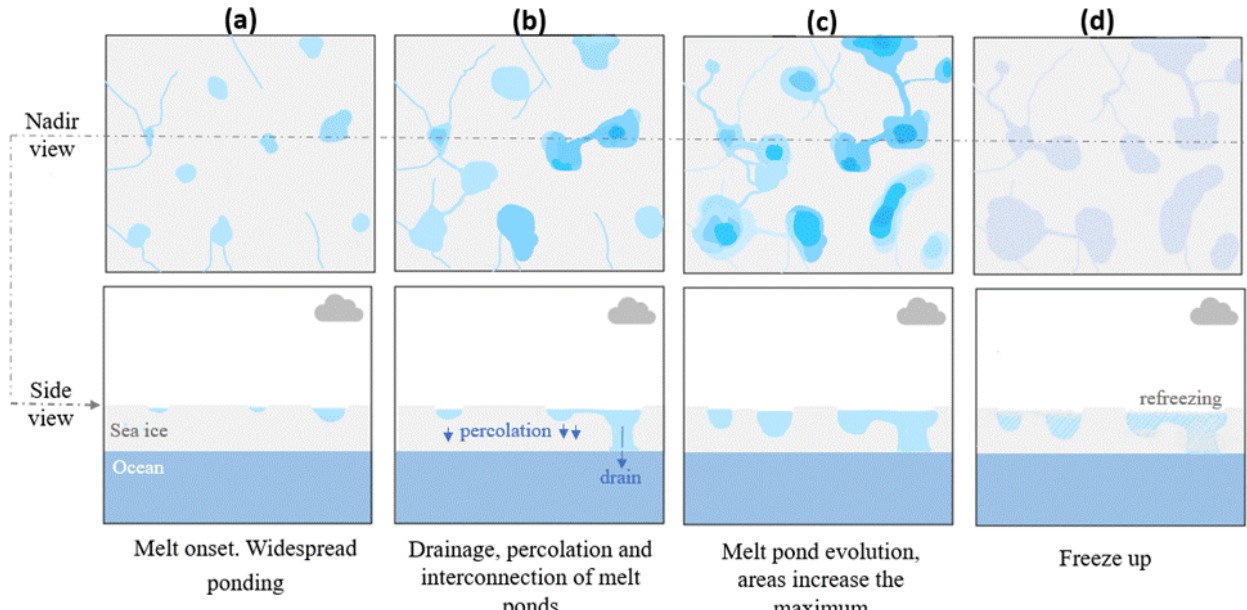

**Figure 3: Main characteristics of the 4 stages of melt pond evolution: a) Melt onset and widespread ponding, (b) Drainage, percolation and interconnection of melt ponds, (c) Melt pond continue to evolve and area covered increases to the maximum and (d) Freeze up**

The influence of ice types and associated surface features, becomes especially apparent within this seasonal context, as they govern where and how ponds form and evolve. Melt ponds form more rapidly on flatter FYI surfaces, where water spreads freely and may melt through the ice by season's end (Fetterer and Untersteiner, 1998). On MYI, more complex topography is shaped by snow dunes and past melt patterns, trapping water in defined pools, inhibiting lateral spread and forming drainage channels (Eicken et al., 2004; Wright et al., 2020; Buckley et al., 2020). FYI also contains more brine and fewer air bubbles

than MYI, enhancing absorption and melt due to reduced scattering (Scott and Feltham, 2010). In addition to seasonal and long-term trends, melt ponds also show diurnal thermal variability influenced by cloud cover. This variability decreases once ponds are ice-covered or snow-lidded.

## 3.    Earth observation methods for melt pond detection and monitoring

A wide range of Earth observation (EO) techniques are currently employed to detect and monitor MPs. This section provides

an overview of spaceborne and in situ/field campaign observational approaches. Particular emphasis is placed on the parameters retrieved and the signatures of melt ponds across the electromagnetic spectrum (EM), focusing both on the optical and microwave regions, for active and passive systems (Section 3.1), followed by considerations on associated limitations. In section 3.2, *in situ* and field campaigns comprising ship-based and airborne surveys are discussed. Section 3.3 addressed data processing techniques to derive MP-related information from EO data, with a focus on recent advances in

algorithm development and multi-sensor integration.



## 3.1. Spaceborne observations - melt ponds signatures across the electromagnetic spectrum

The logistical and financial challenges of in situ campaigns in remote Arctic regions make Earth observation, particularly the
ones from space, a valuable tool for monitoring sea ice. EO sensors operating across the electromagnetic spectrum can retrieve diverse physical parameters related to melt pond presence and properties. Figure 4 illustrates the two primary spectral domain systems: the optical and microwave regions used for the EO of melt ponds. It summarises the type of parameters retrieved (relative to different ways that active and passive systems interact with matter), the 'signature' of MP within each range of parameters, along with main applications and examples of corresponding main satellite missions.


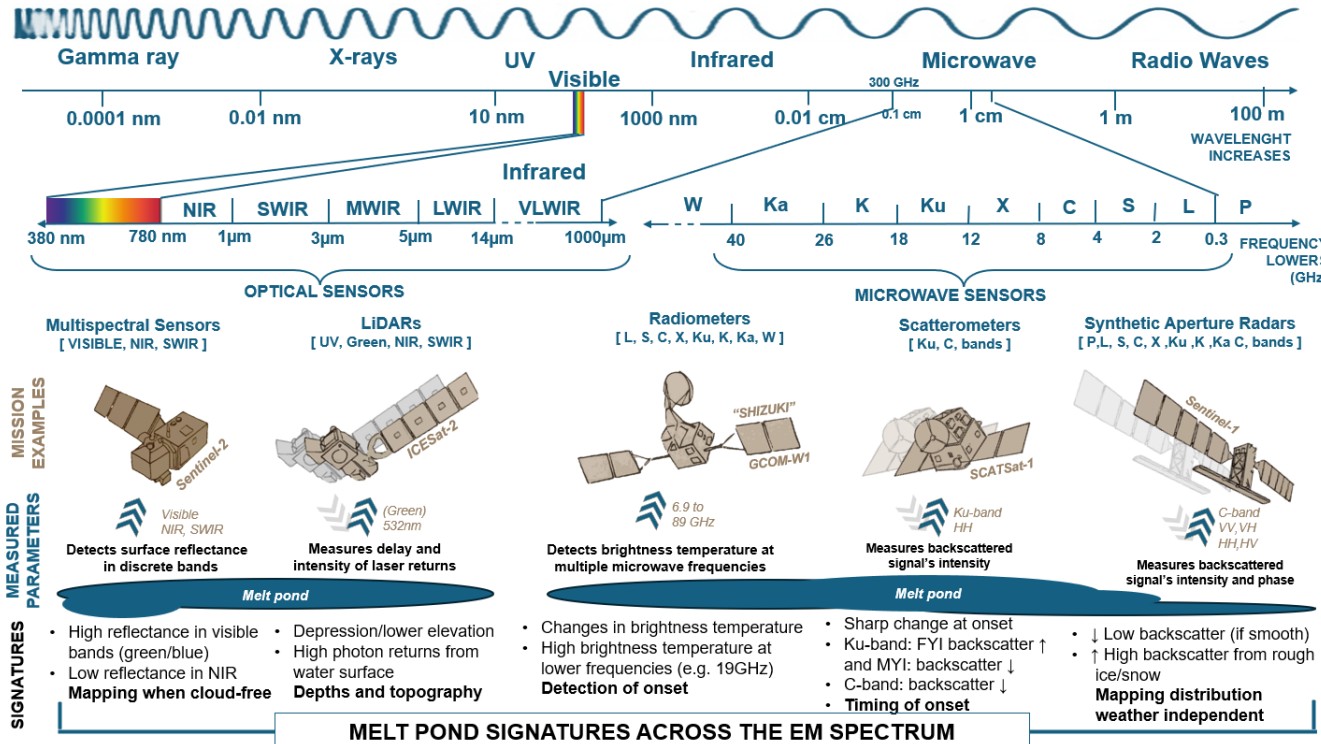

**Figure 4: Overview melt pond signatures and parameters detected across different Earth observation (EO)**






### 3.1.1. Optical sensors: passive (multispectral) and active (LiDAR) systems

**Passive optical systems**

Spaceborne optical monitoring of melt ponds began in the 1970s (Grenfell and Maykut, 1977), and has since been widely used for its ability to capture reflected sunlight across multiple wavelengths. Passive optical systems, exploit the spectral characteristics of melt ponds by measuring surface reflectance, typically across the visible (VIS, ~0.4–0.7 μm), near-infrared (NIR, ~0.75–1 μm) and shortwave infrared (SWIR, ~1 to 3 μm) regions of the EM (see Fig. 4). Melt ponds exhibit low reflectance in the NIR and SWIR due to strong absorption by water in these wavelengths, in contrast to the higher reflectance of surrounding sea ice. Conversely, in the blue-green region of the VIS portion, melt ponds exhibit high reflectance (Malinka et al., 2018). This spectral signature and accentuated differences in certain bands enables effective distinction of melt ponds from surrounding ice and snow, through spectral indices and classification algorithms (see Section 3.3 and Appendix A for methodological approaches). Key applications of passive optical data are melt pond classification, monitoring and retrievals of melt pond fractions (MPF). These applications are influenced by the resolution, which can span from coarse/medium (250 m to 1 km), to high (10 - 60) and very high resolution (0.3-10 m).

Coarse-resolution instrument, such as the Moderate Resolution Imaging Spectroradiometer (MODIS) on board Aqua/Terra satellites have been used for large-scale monitoring of melt ponds, thanks to their frequent temporal coverage and extensive historical record (Tschudi et al., 2008; Rösel et al., 2012; Rösel and Kaleschke, 2012; Ding et al., 2020) as well as the Medium Resolution Imaging Spectrometer (MERIS) aboard ENVISAT (Zege et al., 2015; Istomina et al., 2015, 2025). Higher resolutions, such as the Enhanced Thematic Mapper Plus (ETM+) aboard Landsat7 (Markus et al., 2003; Rösel and Kaleschke, 2011; Qin et al., 2021) have been effectively employed to detect and monitor Arctic melt ponds. Data from the Ocean and Land Colour Instrument (OLCI) on Sentinel-3 (Niehaus et al., 2024), and the Multispectral Instrument (MSI) on Sentinel-2 with visible and near-infrared bands, have been increasingly employed for high-accuracy melt pond fractions retrievals (Wang et al., 2020; Niehaus et al., 2022), and more recently bathymetric studies (Xiong and Li, 2025). For very high-resolution imagery, from the DigitalGlobe's WorldView (WV) constellation has enabled detailed local analyses of melt pond properties and classification (Wright and Polashenski, 2018), providing finer spatial insights, complementing the broader-scale insights provided by coarser-resolution data.

Despite their advantages, optical methods face intrinsic challenges and limitations:

- *Illumination dependency:* A key fundamental operational constraint is the requirement of sunlight to function, rendering optical sensors ineffective during dark periods. Likewise, the persistent presence of cloud cover significantly limits their operational utility in the Arctic's predominantly overcast conditions (Comiso and Kwok, 1996; Fetterer et al., 2008). Moreover, atmospheric scattering and absorption, from aerosols, haze and low solar elevation angles in the Arctic can distort top-of-the-atmosphere (TOA) reflectance causing degradation of image quality, altering spectral signatures (as noted by Zege et al. (2015)), further limiting data quality and availability.



Data unavailability contributes to temporal coverage limitations, limiting understanding of pond evolution and drainage events that occur within hours during melt season.

- *Resolution-Coverage Trade-offs:* Melt ponds can range from one to hundreds meters in diameter (Perovich et al., 2002), making resolution critical. However, optical sensors face fundamental trade-offs between resolution and swath/coverage. Medium/Coarser systems cannot resolve smaller ponds or narrow features (5-10 m wide), causing significant underestimation. Another limitation comes with the mixed pixel complexity, when trying to accurately determine coverage of different surface types within coarser-resolution sensor footprints. MODIS and MERIS offer wider coverage (>1000 km swath) allowing for daily pan-Arctic coverage but much coarser resolution (250-1000 m), with MODIS L1B showing striping artefacts that bias retrievals (Lee et al., 2020). Moreover, Landsat-7 which has higher resolution (30 m resolution, 185 km swath), has proven to not be well-suited for melt pond mapping (Markus et al., 2003). Additionally, for high resolution systems, for instance Sentinel-2 (10 m resolution for the visible and near-infrared NIR bands), despite its considerably higher resolution it also still suffers from mixed pixels (Buckley et al., 2023) and limited Arctic coverage (up to 82.8°) and coastal waters within 20 km of the shore. Similarly, higher-resolution commercial platforms like DigitalGlobe and Pléiades are limited in polar coverage (to 82°N) and availability, requiring license for data access. In order to understand the importance of EO resolution on the MPF retrieval, Buckley et al. (2023) found 18% melt pond fraction difference between Sentinel-2 (7.6%) and WorldView (25.5%) for identical areas, with biases up to 20%.

- *Reflectance signature and Spectral Ambiguity*: Melt ponds and open water/leads can be indistinguishable due to overlapping spectral properties (Lee et al., 2020), contributing further to misclassification challenges, for instance, fresh snow increases NIR reflectance, mimicking melting ice signatures. This can cause considerable MPF underestimations (Istomina et al., 2025). On the other hand, early-stage freeze ponds resemble liquid ponds spectrally, leading to MPF overestimations (Xiong and Ren, 2023).

**Active optical systems**

Light Detection and Ranging (LiDAR) technology commonly operates at wavelengths ranging between 905 nm and 1550 nm in commercial systems, while scientific applications, including satellite and airborne LiDAR, often use 1064 nm (short-wave infrared, SWIR) and 532 nm (green light) (see Fig. 4), tailored for atmospheric and bathymetric applications, respectively. In LiDAR, laser pulses composed of photons are emitted at these wavelengths. By measuring the round-trip time and return intensity of these photons, LiDAR enables the determination of surface elevation and reflectance. More recently, active optical sensors have enabled measurements of sea ice topography and pond depths, by analysing the time delay and intensity of returned laser pulses reflected from the ice and water surfaces. Green laser wavelength penetrates clear water, enabling measurements of shallow waterbody depths (Szafarczyk and Toś, 2023). The Advanced Topographic Laser Altimeter System (ATLAS), aboard the NASA's ICESat-2 satellite, operating at 532 nm, has been used to map sea ice topography and detect melt pond presence and their depths (Farrell et al., 2020; Tilling et al., 2020; Buckley et al., 2023;





Herzfeld el al. 2023), however, as of now, no operational ICESat-2 data product automatically includes pond depth measurements.

The main limitations of LiDAR consist in laser penetration and consequent ability for depth determination. Laser penetration
through melt pond water is limited by absorption and scattering effects, particularly in turbid or deeper ponds. Moreover, height retrievals suffer from insufficient signal photons, leading to reduced measurement accuracy. The highly reflective nature of MP surfaces causes detector saturation, leading to non recorded photons.

### 3.1.2. Microwave sensors: Active (radars) and passive (radiometers) systems

The wavelengths of microwave sensors (Fig. 4), are not affected by atmospheric particles operating independently of lighting
and weather conditions, contributing to their demonstrated capabilities of detecting melt ponds in the Arctic sea ice (Scharien et al., 2014; Marshall et al., 2019).

**Active microwave systems**

Of the four active microwave sensor types: Synthetic Aperture Radar (SAR), scatterometers, altimeters, and weather radars,
this section focuses on SAR and scatterometers, given their relevance to melt pond studies. Despite the wide range of frequencies covered by SAR systems (see Fig. 4), SAR-based melt pond studies primarily rely on three frequency bands: C-band (~5.4 GHz, ~5.6 cm), X-band (~9.6 GHz, ~3.1 cm) and L-band (~1.3 GHz, ~23.5 cm). During the melting season, the liquid water in snow increases its dielectric constant ($\varepsilon*$) enhancing the absorption of microwave energy, reducing microwave returns (also due to the specular nature), creating the characteristic low backscatter ($\sigma\circ$) signature of melt ponds.
However, this signature is very complex as it is highly dependent on the intercombination of key radar parameters such as frequency (band), incidence angle and polarization, and the surface roughness and dielectric properties of melt ponds (and their surroundings) (Scharien et al., 2014a; Han et al., 2016; Li et al., 2017).

Data retrieved by missions using C-band, such as Sentinel-1 and RADARSAT-2, are the most commonly used, due to their balance between spatial resolution (5-40 and 3-100m, respectively and depending on the acquisition mode), penetration
depth and sensitivity to surface roughness. Comparatively, X-band SAR systems like TerraSAR-X (9.65 GHz, 3.1 cm), provide higher spatial resolution (0.25-40 m), are more sensitive to fine-scale surface features (Fig. 5), making them suitable for melt pond boundary detection, or early melt detections (Kern et al., 2010; Fors et al., 2017). L-band missions like ALOS-2, though less frequently used, allows greater penetration through wet snow due to their longer wavelengths, proving valuable for melt onset and drainage stages (Mahmud et al., 2020; Tavri et al., 2023).
Polarization is a critical parameter to consider on melt pond signatures detection. SAR systems have evolved from single- to quad-polarization (Scharien et al., 2010), enabling the use of co-polarization ratios (co-pol, VV/HH for instance) to reduce geometric effects and emphasize scattering mechanisms (Woodhouse, 2015). Co-pol ratios effectively indicate liquid melt pond fractions on FYI, even under wind-roughened conditions (Scharien et al., 2012), while cross-pol ratios, less affected by incidence angle, are sensitive to volume scattering and useful for ice classification (Johansson et al., 2017). Cross-pol also





aids in distinguishing FYI from MYI during pond drainage and detecting freeze events due to snow cover contrast (Scharien et al., 2012). Incidence angle also has a significant impact on melt pond signatures: as it increases, the backscatter intensity generally decreases, however, the optimal incidence angle for melt pond retrievals depends on a number of factors, such as frequency band used (Fors et al., 2017).

While SAR systems could, in principle, overcome some of the optical-related limitations, offering all-weather, day-and-night

imaging capabilities, it remains a non-trivial task to retrieve melt pond fractions at broad spatial scales using radar-based methods (Huang et al., 2016). To date, no comprehensive SAR-based MPF product exists with a reliability comparable to that of optical aerial or satellite observations (see Tables 2 and 4). The challenge is particularly pronounced for rough or more complex ice surfaces such as MYI, as correlations between radar signatures and MPFs has been mostly studied for smooth landfast FYI ( Mäkynen et al., 2014; Scharien et al., 2014; Tanaka et al., 2016; Li et al., 2017). In fact, most studies

using SAR had limitations in retrieving melt ponds since the $\sigma\circ$, polarisation ratios, or frequencies were insufficient to discriminate melt ponds from sea ice and open ocean water, leading to cases of under- or overestimations (Li et al., 2017). Even with present-day high-resolution SAR sensors (e.g., TerraSAR-X, COSMO-SkyMed, RADARSAT-2, Sentinel-1), which can achieve spatial resolutions between 1 m and 10 m, large-scale melt pond mapping remains limited. Key challenges include surface roughness, speckle noise, and radar parameters, as outlined below:

- *Polarization limitation and polarimetic features*: Early SAR studies relied on single-polarization (single-pol, VV or HH) backscatter, which is primarily sensitive to surface roughness. As a result, smooth melt ponds           under calm conditions were poorly distinguishable (Yackel and Barber, 2000; Kim et al., 2013; Mäkynen et al., 2014). Co-pol ratios (VV/HH for example) showed promise for melt ponds on FYI, exploiting dielectric property contrasts (Fors et al., 2017; Scharien et al., 2012, 2014). However, systematic overestimations

occur because melt ponds share similar polarimetric characteristics with open ocean water (Li et al., 2017). Multi-pol SAR data can enhance discrimination, but dual co-pol data have substantially smaller swath widths, reducing suitability for regional monitoring.

- *Incidence angle dependencies*: Radar backscatter changes with the incidence angle, meaning that the same pond may have different signatures at different acquisition geometry. X-band optimal performance occurs at 29-40°

incidence angles (Fors et al., 2017), while TerraSAR-X dual-pol performs best at 20-30° for MYI monitoring (Han et al., 2016). At C-band, melt pond backscatter is weak at lower angles (20-25°) but has a marked increase at ~30° due to specular reflection (Yackel and Barber, 2000). It should be emphasized, however, that the optimal incidence angle for melt pond detection likely depends on a combination of factors, including ice type and condition, as well as the imaging system's specifications. For this reason, constructing large-scale SAR mosaics remains a challenge

due incidence angle variability (Howell et al., 2020).

- *Environmental sensitivities: ice type, wind conditions and seasonality*: Wind-induced surface roughness significantly limits X-band systems compared to C-band, particularly above 5 m/s. Optimal retrieval occurs at intermediate wind speeds (~6.3 m/s for X-band), resulting in narrow operational windows (Fors et al., 2017;



Scharien et al., 2014). Under calm conditions (<3 m/s), melt ponds yield specular scattering, producing lower
backscatter (Yackel and Barber, 2000; Scharien et al., 2010, 2012; Fors et al., 2017). Melt pond signatures also vary
seasonally due to changing pond depths and ice evolution. Overall, C-band performs best during melt onset, early
melt stages and peak conditions, while L-band provides improved separability during post-drainage due to deeper
penetration and sensitivity to volumetric scattering (Tavri et al., 2023). During refreezing, particularly in FYI, C-
band backscatter increases due to rough surface scattering and volume scattering from desalinated upper ice cover
(Scharien et al., 2012). It should be noted that most studies focus on FYI, but melt ponds on MYI exhibit different
physical and microwave scattering characteristics, requiring further examination of polarimetric SAR retrievals for
MYI (Han et al., 2016).

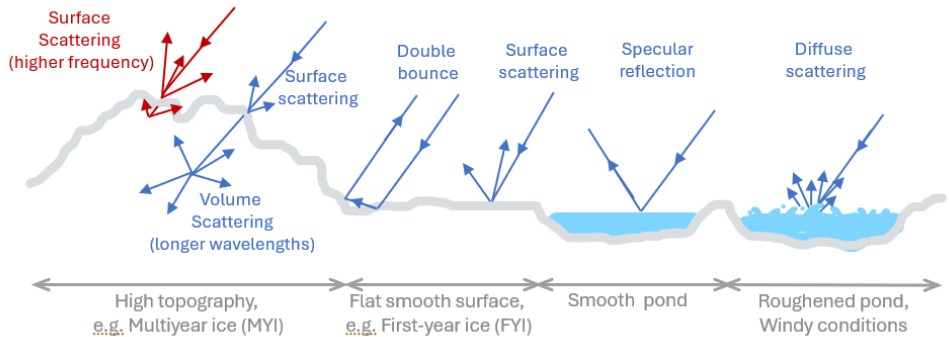

**Figure 5: Radar scattering mechanism. Different interactions of microwave radar with different sea ice types and melt pond under**
**different wind conditions**

While SAR provides high spatial resolution imagery, scatterometers, generally operating in C-band and Ku-band (~12-18
GHz), analyze backscatter from multiple viewing angles. This makes them valuable for large-scale sea ice monitoring due to
their frequent temporal coverage, albeit at a coarser spatial resolution than SAR (on the order of kilometers). Scatterometers
are particularly sensitive to water content of the surface. Differently to SAR, Ku-band scatterometers have been mostly used
to infer the timing and extent of Arctic melt onset. As the amount of liquid water in the snow cover increases, the wet snow
leads to a significant decrease in radar backscatter (Forster et al., 2001).

Three examples of scatterometers instruments used for the detection of melt onset are the SeaWinds sensor onboard the
QuikSCAT satellite which operated in Ku-band (13.6 GHz), discontinued in 2009 (Howell et al., 2005), and the ISRO
ScatSat-1 (13.5 GHz) (Geldsetzer et al., 2023), discontinued more recently, in 2021. The Advanced Scatterometer (ASCAT),
operating in the C-band (5.3 GHz) aboard multiple satellite missions (MetOp-A, MetOp-B), has also been used in melt onset
detections, extending the records from QuickSCAT despite operating at a different frequency band (Mortin et al., 2014). C-
band and Ku-band show different sensitivities to melt due to their varying depth penetration depths. Ku-band shows a
significantly stronger sensitivity towards melt onset due to its shallow penetration as scattering dominates above the snow-
ice interface (Ulaby et al., 1984). For this reason ASCAT's C-band melt signature differs from that of Ku-band: backscatter



values drop as the first appearance of wet snow and melt ponds, fluctuate during melt season and summer, remaining low during summer, and gradually increase during winter. Also conversely to Ku-band, C-band backscatters tend to be stronger and less variable. Regarding scatterometers main limitations:

- *Multiparameter dependency*: Scatterometers show the same nature of limitations described for SAR systems, showing complex responses varying with ice type, polarization ratio used and wind-induced roughness (Geldsetzer
et al., 2023).

- *Footprint contamination issues*: Additionally, scatterometer data can be significantly affected by noise from land or MYI contamination, especially in smooth FYI regions, due to its wide antenna footprint and beam orientation (Maknun et al., 2024).

**Passive microwave systems**

Passive microwave radiometers measure brightness temperature, which is the intensity of upward-traveling microwave radiation across multiple frequencies typically ranging from 6.9 to 89.0 GHz. Key instruments include the Advanced Microwave Scanning Radiometer-Earth Observing System (AMSR-E) and its successor Advanced Microwave Scanning Radiometer 2 (AMSR-2) aboard NASA's Aqua and JAXA's Global Change Observation Mission 1st-Water (GCOM-W1)
satellites, respectively. These sensors have been used to derive melt pond fractions through gradient ratios and through brightness temperature measurements at 6.925, 10.65, 18.7, 23.8, 36.5, and 89.0 GHz (Tanaka et al., 2016; Tanaka and Scharien, 2022). A key radiometer is the Special sensor microwave imager (SSM/I), onboard the United States Air Force Defense Meteorological Satellite Program (DMSP). It operates at four frequency intervals centred at 19.35, 22.235, 37.0 and 85.5 GHz allowing for melt onset detection (Marshall et al., 2019).

The SMMR–SSM/I microwave sensors are very sensitive to emissivity changes caused by the presence of small amounts of liquid water in the snowpack and on the sea ice surface (Comiso and Kwok, 1996). The fundamental principle exploits the significant emissivity difference between water and ice in microwave frequencies. Water's much higher microwave emissivity causes brightness temperatures to increase substantially when liquid water appears in snow or forms melt ponds (Mote et al., 1993), enabling detection of even small liquid water amounts. Daily brightness temperature variations serve as
reliable indicators for Arctic melt and freeze onset dates. The 19 GHz horizontal polarization frequency is preferred for melt detection as it shows the lowest brightness temperature for dry firn, enhancing temperature contrast from liquid water emergence (Liu et al., 2006). While other factors such as air temperature, water content, snow grain size and internal ice layers, also influence daily brightness temperature, surface liquid water presence is the dominant driver of daily variation (Tedesco, 2007).

Like active microwave systems, radiometers face fundamental shortcoming related to microwave scattering, emission signatures and spatial resolutions constraints:

- *Radiometric ambiguity of water features*: Melt pond brightness-temperature signals resemble open ocean water, making them indistinguishable from open water. Moreover, at microwave frequencies of ≥6 GHz, melt ponds



(freshwater) and open ocean (saltwater) also exhibit similar brightness temperature (Gogineni et al., 1992). , leading them to be misclassified as open leads or cracks.

- *Low penetration Depth*: At frequencies of 19, 37, and 89 GHz, microwave radiation penetrates only ~1 mm into liquid water, preventing sensors from distinguishing sea water from melt ponds on ice (Ulaby et al., 1986), being the cause of introducing uncertainties in sea ice concentration retrievals (Cavalieri et al., 199; Comiso and Kwok, 1996). This limitation restricts passive microwave remote sensing to areas with high sea ice concentration (>95%), where open water influence can be neglected (Tanaka et al., 2016).

- *Coarse resolution:* Passive microwave sensors generally have coarse spatial resolution, limiting their ability to detect small melt ponds and fine-scale features. When ponds are smaller than sensor resolution, the resulting signal becomes a mix of sea ice and leads (Comiso and Kwok, 1996). The varying sea ice microwave signature, distinct emission signatures of dry versus wet snow/ice, and varying water surface roughness make MPF retrieval overall challenging across the microwave sensors (Xiong and Ren, 2023).

## 3.2. In situ and field/campaign observations

While spaceborne observations are powerful tools for monitoring melt ponds, in situ ice observations complement them by providing information that is difficult or impossible to obtain remotely, such as very high spatial resolution images or depth measurements. Table 1 provides a summary of various field campaigns, which although not focused solely on melt ponds, have nonetheless contributed to the acquisition of in situ melt pond data. It is worth noting that some instruments used in airborne and ship-based observations share similar sensor characteristics with those used in spaceborne platforms. For this reason, due to their similar nature, such instruments are not discussed separately here, having been covered in the previous section.

We have classified these campaigns depending on their duration: there are three main types of observational campaigns: 1) year-long ice-drifting expeditions; 2) shorter cruises or icebreaker-based campaigns, typically lasting a few weeks and often combining ship-based observations with helicopter or drone deployments; and 3) airborne surveys conducted from aircraft. Airborne platforms normally include aerial photography (using digital cameras) (Perovich et al., 2002) or microwave sensors (Kim et al., 2013; Han et al., 2016) onboard of drones, helicopters or unpiloted aerial vehicles (UAVs) (Tschudi et al., 2008; Fetterer et al., 2008).



**Table 1: Campaign summary table**

| Campaign | Type | Duration | Location | Primary focus | MP-specific data acquisition | Data Access |
|---|---|---|---|---|---|---|
| MOSAiC | Year-long Ice Drifting-Base | Sep 2019 - Oct 2020 (1 year) | Central Arctic Ocean | Coupled Arctic climate system processes | Hyperspectral, bathymetry & biogeochemical measurements | https://mosaic-expedition.org/mosaic-data/ |
| SHEBA | | Oct 1997 - Oct 1998 (1 year) | ~570 km NE of Prudhoe Bay, Alaska | Surface heat budget & ice-albedo feedback processes | Basic broadband albedo and coverage measurements | https://www.eol.ucar.edu/field_projects/sheba |
| ICE212 (NPI) | Week(s)-long Icebreaker cruise-Based | Jul 26 - Aug 3, 2012 (8 days) | North of Svalbard (82.3°N) | In situ albedo measurements and aerial imagery | | n.a. |
| HOTRAX | | Aug 5 - Sep 30, 2005 (66 days) | Central Arctic Ocean and North of Svalbard | Physical properties of the polar ice pack | Pond depths and aerial photography | https://www.ncei.noaa.gov/access/paleo-search/study/14169 |
| Operation IceBridge | Airborne surveys (multi-instrument) | 2009-2021 (13 years) | Arctic, Antarctic, Alaska | Ice sheet and sea ice monitoring via remote sensing | Aerial imagery and altimetry of melt features | https://nsidc.org/data/icebridge/data |
| THINICE | | August 2022 | Svalbard | Arctic Cyclone–Cloud–Ice Interactions | n.a. | https://ralithinice.aeris-data.fr/# |

### 3.2.1. Ship-based drifting campaigns

**MOSAiC**

From September 2019 to July 2020, the Multidisciplinary Drifting Observatory for the Study of Arctic Climate (MOSAiC)
(Fig. 7) stands as the largest Arctic research expedition in history (Shupe et al., 2021). Led by the Alfred Wegener Institute,
the German research icebreaker RV Polarstern was deliberately frozen into the Arctic sea ice to drift across the central Arctic
Ocean for an entire year, including the polar winter. The central observatory consisted of many fixed installations on RV
Polarstern itself, as well as a wide array of sampling and measurements: flux stations, meteorological sensors, oceanographic
profilers, biogeochemical sensors and GPS position-tracking bows, allowing for comprehensive measurements of conditions
across scales of 10-100 km in the coupled atmosphere-ice-ocean system (Rabe et al., 2024), with contributions to melt pond
distribution (Neckel et al., 2023; Sperzel et al., 2023) and spatiotemporal studies (Webster et al., 2022b).



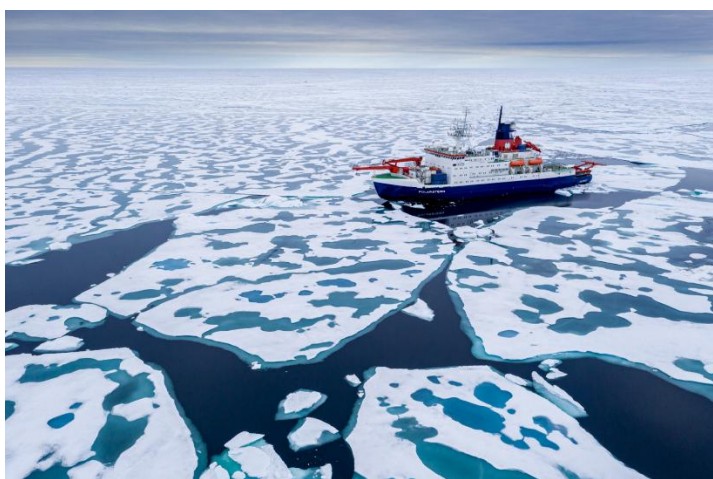

**Figure 6: The Polarstern ship amongst melting ice and melt ponds whilst part of the MOSAiC campaign. Image credits: Steffen Graupner/AWI/MOSAiC**

**SHEBA**

About 22 years earlier, conducted from October 1997 to October 1998, the Surface Heat Budget of the Arctic Ocean (SHEBA) was a comprehensive field experiment aimed at investigating the interactions among the atmosphere, sea ice, and ocean in the Arctic . The Canadian icebreaker Des Groseilliers was intentionally embedded within the Arctic ice pack, approximately 570 km northeast of Prudhoe Bay, Alaska, serving as a drifting research station. Over this yearly period researchers were able to track an icepack and take direct observations of the sea ice evolution over a full melt season (Uttal et al., 2002). SHEBA also utilized a diverse array of instruments and platforms, including aircraft, meteorological towers, surface radiometers, oceanographic and meteorological instruments housed in structures on the ice, helicopters and buoys. Both SHEBA and MOSAiC extensive observations over approximately a year, substantially aided the improvement and refining of melt pond parametrisations, and advanced our understanding of the Arctic in general (Wang et al., 2024). Data collected in these campaigns have also been used in numerous melt ponds studies as validation data (see Appendix C).

**3.2.2. Cruise icebreaker-based campaigns**

**ICE212 (NPI)**

The ICE212 (NPI) campaign consisted of an 8-day ice drifting flow experiment on R/V *Lance* experiment carried out by the Norwegian Polar Institute in the Arctic north of Svalbard at 82.3∘N from 26 July to 3 August 2012. The study collected in situ albedo measurements and aerial imagery. The in situ albedo measurements were representative of the four main surface types: bare ice, dark melt ponds, bright melt ponds and open water. Images were acquired by a helicopter-borne camera system during ice survey flights covering about 28 km2 (Divine et al., 2015).



**HOTRAX**

The Healy Oden Trans-Arctic Expedition (HOTRAX), during the 5 August to 30 September 2005 conducted a
comprehensive survey of the physical properties of the polar ice pack. The research program encompassed four main
categories of snow and ice characterization: transit observations, ice station measurements, helicopter survey flights, and
autonomous ice mass balance buoy deployments. Notably, pond fractions were substantial early in the expedition at the
southern ice pack boundary. Researchers documented melt pond coverage alongside sediment-laden and biologically rich
ice areas. The expedition included 28 ice stations where teams measured snow depth, ice thickness, and pond depths,
creating a comprehensive dataset of Arctic ice conditions. The helicopter photographic survey flights were conducted 1-3
times per week using a digital camera (Nikon D70), and allowed to detect smaller-scale features such as leads, ridges and
melt ponds that satellite imagery could not resolve (Perovich et al., 2009). HOTRAX generated melt pond fraction datasets
that have been used validation dataset for multiple studies focused on MPF retrievals (Peng, et al., 2022; Ding et al.,2020,
Tanaka et al., 2016; Istomina et al., 2015) - see Appendix B. More details of these datasets are addressed later in section
4.1.2 which is dedicated to observational datasets.

### 3.2.3. Airborne surveys

**OPERATION ICEBRIDGE**

NASA's Operation IceBridge (OIB) was a 13-year (2009-2021) airborne mission dedicated to survey land and sea ice across
the Arctic, Antarctic and Alaska. In the summers of 2016 and 2017, the NASA Operation IceBridge (OIB), obtained (to that
date), the most widespread airborne survey of summer sea ice conditions, using multiple aircraft throughout the years
(MacGregor et al., 2021). The NASA 524 HU-25C Guardian aircraft carried multiple instruments on board (radar, laser
altimeter and a digital camera), and covered both MYI and FYI in the Beaufort Sea (in 2016) and MYI in the Lincoln Sea
and central Arctic Ocean (in 2017). Although data from this campaign was used to retrieve melt pond fraction (MPF)
(Wright and Polashenski; 2018; Buckley et al., 2020), it has also been widely used as an independent validation dataset in
the development of MPF retrieval algorithms (Wang et al., 2020; Xiong and Li, 2025), as is later described in Table 2, in
section 4.



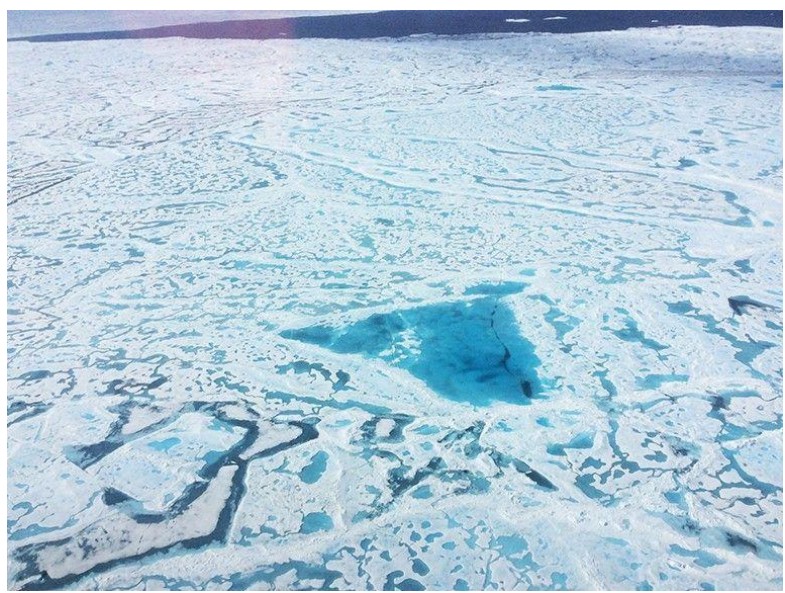

**Figure 7: A large pool of meltwater over sea ice, as seen from an Operation IceBridge flight over the Beaufort Sea on July 14, 2016. Credits: NASA/Operation IceBridge**

**THINICE**

The field campaign *THINICE* was based in Svalbard in August 2022. Its intent was to provide unique observations of summertime Arctic cyclones and the associated tropospheric and sea ice conditions (Rivière et al., 2024). Arctic cyclones are known to cause rapid sea ice loss. Cavallo et al. (2025) investigated the variability in changes in sea ice extent for a period of less than 18 days and their association to Arctic cyclones as well as to tropopause polar vortices. They discover the increased

importance of understanding Arctic cyclones for sea ice depletion - accordingly observations in these conditions could be useful for understanding sea ice impacts also. Two research aircraft were deployed, the ATR42-SAFIRE equipped with radar-lidar remote sensing instrumentation and the BAS Twin Otter. The THINICE campaign was not extensively intended to be a melt pond observational campaign, but nonetheless it collected high-resolution and accurate melt pond data during its operations under unique conditions. It may therefore reveal insights into melt ponds in previously unobserved/poorly

observed conditions that are essential to determining the future of sea ice loss. As such, the THINICE campaign can be considered a valuable addition to melt pond datasets for improving physical and machine learning models, not only because melt ponds are clearly identifiable through the high quality observations, but also due to the unique conditions in which these observations were taken. These observations are already being used to determine parametrisation settings in models.

To date there is no dedicated platform to access all campaign and melt pond-related observational datasets, however,

regarding specifically shipborne Arctic sea ice observation data, there is a program debuted in the summer of 2012 that supports the collection of data from multiple campaigns. IceWatch is coordinated by the Norwegian Meteorological Institute, the University of Alaska Fairbanks, the International Arctic Research Center and the Geographic Information Network of Alaska, and through a web-based application name ASSIST, it assists in the collection and archiving of sea ice observations



recorded during research cruises in the Northern Hemisphere and can be accessed at https://climate-cryosphere.org/ice-
watch-assist/.

### 3.3. Post-processing techniques for melt pond observations applications

Processed melt pond data, derived from satellite or in situ campaigns, typically serves two main applications: (1) the
generation of melt pond classifications that categorize pixels into surface types such as ice, melt pond, or open ocean for
surface mapping and temporal change monitoring; and (2) for the retrievals of melt pond fractions (MPFs). MPF retrievals,
defined as either the fraction of ponded ice or as the fraction of melt ponds within a satellite scene, are particularly relevant
for sea ice evolution studies, general circulation models (Flocco and Feltham, 2007; Polashenski et al., 2012), and seasonal
sea ice extent forecasting (Flocco et al., 2012; Schröder et al., 2014; Howell et al., 2020; Ding et al., 2020; Feng et al., 2021).
Over time, different methods have been developed to generate melt pond classifications and retrieve MPF from data
acquisitions. Figure 8 illustrates the main families of technical approaches, that include techniques that are sensor-specific or
sensor independent and methods used when combining different data types. Appendix A provides a more detailed
description of each technique. Early approaches for optical data, comprise fundamental statistical techniques, including pixel
intensity and histogram analysis of specific bands for MP pixel classification (Webster et al., 2015; Huang et al., 2016,
Buckley et al., 2023). More advanced approaches adopted in later studies, enabled MPFs retrievals through two main
techniques. First, spectral unmixing methods that decompose mixed optical pixels into endmembers representing different
surface types (Kern et al., 2016; Yackel et al., 2018; Xiong and Ren, 2023). Second, physically-based algorithms such as
Melt Pond Detector 1 and 2, and LinearPolar (see Appendix A) use radiative transfer and bidirectional reflectance
distribution function (BRDF) models (Niehaus et al., 2024; Istomina et al., 2025; Niehaus et al., 2022; Qin et al., 2021).
These optical-domain methods inherently rely on reflectance signatures for their effectiveness. Microwave data processing
employs distinct methodologies, that geophysical inversions, conversion of backscatter to MPF and empirical relationship
between co-polarization and compact polarization (Yackel and Barber, 2020; Scharien et al., 2014; Li et al., 2017). Though
these approaches are largely sensor-specific until the development of multi-sensor integration techniques.



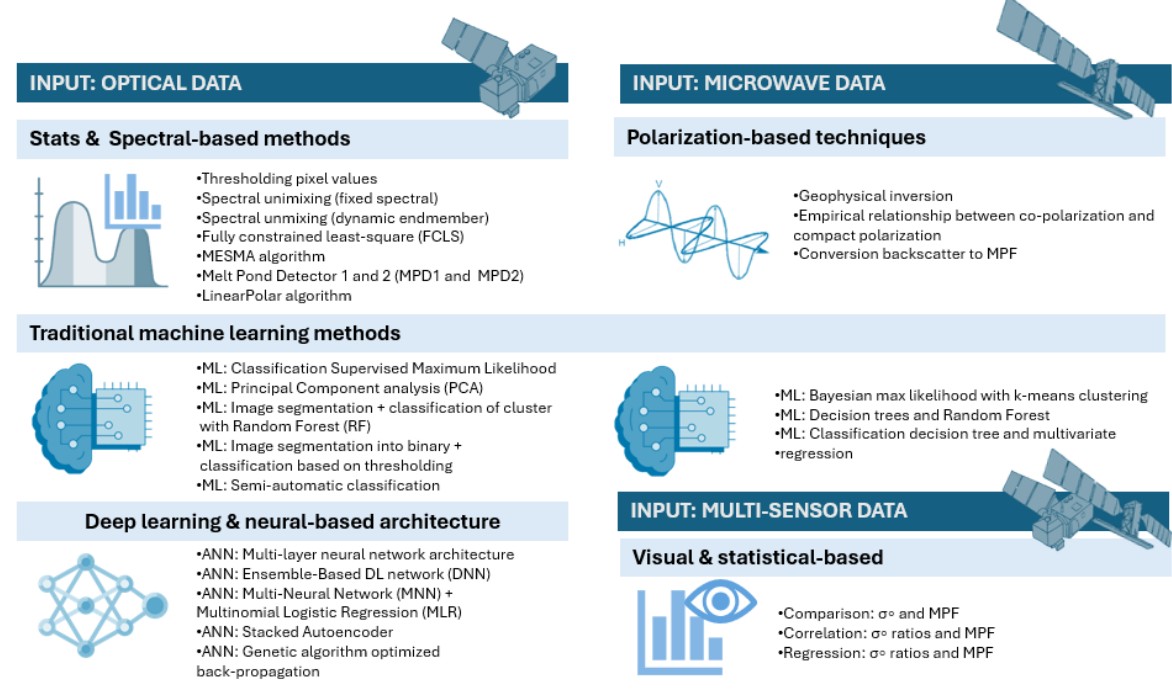

**Figure 8: Summary of the main family of approaches for both optical, microwave and combination of input data.**

Since 2014, MPFs retrieval based on multiple sensors have been possible, using correlation and regression approaches (Mäkynen et al., 2014; Tanaka et al., 2016; Fors et al., 2017; Scharien et al., 2017; Ramjan et al., 2018; Howell et al., 2020; Tanaka and Scharien, 2022). These fusion methods link microwaves to optical-based MPF and are fusion-suited since they are essentially empirical mappings that can integrate information from multiple sensors. Likewise, machine learning methods span across both sensor types: traditional techniques such as supervised classifiers, clustering, decision trees and random forests have been predominantly applied for melt pond classifications both for optical data (Fetterer et al., 2008; Sankelo et al., 2010; Renner et al., 2013; Miao et al., 2015; Divine et al., 2015; Wright and Polashenski, 2018) but also microwave data (Kern et al., 2010; Han et al., 2016; Ramjan et al., 2018). Deep learning methods, while not inherently sensor-specific, have proliferated significantly after 2020, with multiple ANN architectures (ensemble-based deep neural networks (DNN), multi-neural networks, stacked autoencoders) being used essentially for MPFs retrievals based on optical acquisitions (Rösel et al., 2012; Ding et al., 2020; Lee et al., 2020; Lee and Stroeve, 2021; Feng et al., 2021; Peng et al., 2022). A wide variety of approaches may indicate that the field has not yet convergent on a widely accepted approach or optimal method, suggesting that different techniques are better suited to specific data types, or that optimal solutions are still being developed.



## 4. Melt pond observations: available dataset and lead studies

### 4.1. Melt ponds and melt pond fraction generated datasets

This section provides a comprehensive overview of available melt pond observational datasets and melt pond fraction (MPF) products derived from satellite imagery, aerial photography, and ship-based observations. We present two distinct categories of Arctic melt ponds datasets, which differ in spatial coverage (pan-Arctic versus regional), spatial resolution (kilometer versus meter scale), and the nature of the data (satellite-based continuous monitoring versus campaign-specific programs): (1) Pan-Arctic MPF datasets which offer continuous, multi-year coverage at coarser resolutions (see Table 2 in section

4.1.1), and (2) high resolution regional MP and MPF products that provide finer resolution but spatially limited observations from a combination of field campaigns and targeted remote sensing efforts (see Appendix B). The second category of datasets, which include melt pond imagery, melt pond classification maps and MPF products at meter-scale resolution, are typically confined to specific Arctic regions and time periods. By examining both tables 2 and 4, we highlight their complementary nature and respective strengths.

### 4.1.1. Pan-Arctic satellite-based data melt pond fraction datasets

While numerous melt pond-related datasets exist, only a few provide pan-Arctic coverage and multi-year continuity essential for large-scale studies. Table 2 presents an overview of these pan-Arctic MPF datasets, which in this article are identified by their sensor name and lead author (their alternative names used in other studies are provided in parentheses). The table summarizes key characteristics, highlighting the sensor used, temporal span, coverage area, spatial and temporal resolutions,

data format, access information, and references. The evolution of pan-Arctic MPF datasets reflect advancing sensor technology and algorithm development. Among the pioneering efforts, the MODIS-RÖSEL MPF, developed at Hamburg University, established baseline coverage that spans from 2000 to 2011 (Rösel et al., 2012; Rösel et al., 2015). Concurrently, the MERIS-ZEGE MPF dataset (version V1.7) covers a similar time span, and was produced at University of Bremen, represents an improved iteration of earlier versions (Zege et al., 2015; Istomina et al., 2015). Building on this foundation, the

same team of authors subsequently released OLCI-ISTOMINA MPF (version 1.7), which extends seasonal coverage to include May through September (plus the first week of October for 2021). This dataset uses the same retrieval algorithm as the one used to generate MERIS-ZEGE MPF. The authors have recently developed an updated version of the algorithm by adapting the MPD2 algorithm (see Appendix A) to OLCI sensor, effectively extending the MERIS-based MPF (2002-2011) with recent data (2017 to present) (Istomina et al., 2025). Complementing these efforts, several MODIS-based products offer

extended temporal coverage. The MODIS-DING MPF dataset spans from 2000 to 2019, covering each year from May to September (Ding et al., 2020). Likewise, the MODIS-LEE MPF dataset, covers nearly the same period (although with one year longer) from June 2000 to August 2020, with a slighter wider coverage (regarding latitude) and a higher resolution compared to all MPF datasets mentioned so far (Lee and Stroeve, 2021). Additionally, MODIS-PENG MPF covers a similar time span, but is available in a different data format compared to the other datasets.





**Table 2: Open pan-Arctic multi-year melt pond fraction (MPF) datasets**

| Dataset Name | MODIS-RÖSEL MPF (UH-MPF) | MERIS-ZEGE MPF (MERIS MPF 1.7 or UB-MPF) | OLCI-ISTOMINA MPF (UB-OLCI) | OLCI-NIEHAUS (MPD2 MPF) | MODIS-DING MPF (BNU-MPF) | MODIS-LEE MPF | MODIS-PENG MPF (NENU-MPF) |
|---|---|---|---|---|---|---|---|
| Sensor | MODIS | MERIS | OLCI | OLCI | MODIS | MODIS | MODIS |
| Format | NetCDF | NetCDF | NetCDF | NetCDF, jpg | NetCDF | NetCDF | .img |
| Time span | 2000 - 2011 | 2002-2011 | 2017-2023 | 2017-2023 | 2000 - 2019 | 2000-2020 | 2000-2020 |
| Coverage | Lat .> 60° | Lat .> 60° | Lat .> 60° | Lat .> 60° | Lat .> 60° | Lat .> 57.8° | Lat .> 60° |
| Spatial res. | 12.5 km | 12.5 km | 12.5 km | 12.5 km | 12.5 km | 1 km < 10 km | 12.5 km |
| Temp. res. | Weekly | Daily | Daily | Daily, Weekly | Weekly | Daily, Weekly, Monthly | Daily |
| Data access | https://www.cen.uni-hamburg.de/en/icdc/data/cryosphere/arctic-meltponds.html#beschreibung | https://data.seaice.uni-bremen.de/meris/mecosi/ | https://data.seaice.uni-bremen.de/olci/ | https://data.seaice.uni-bremen.de/MeltPonds-Albedo/MPD2/ | https://doi.pangaea.de/10.1594/PANGAEA.933280 | https://ramadda.data.bas.ac.uk/repository/entry/show?entryid=b91ea195-fd3d-4171-bae4-198c46575c16 | https://ieee-dataport.org/documents/nenu-mpf#files |
| Reference | Rösel et al. (2012); Rösel et al. (2015) | Zege et al. (2015); Istomina et al. (2015, 2025) | Istomina et al. (2025) | Niehaus et al. (2024) | Ding et al. (2021); Ding et al. (2020) | Lee et al. (2020); Lee and Stroeve, (2021) | Peng et al. (2022) |


These pan-Arctic datasets share common challenges related to their spatio-temporal continuity, partly due to cloud obscuration that creates data gaps. Despite these limitations, important distinctions exist in their accuracy and temporal coverage capabilities. MERIS-ZEGE MPF, was considered to have the highest estimation accuracy because it was derived using a physical-based model (Istomina et al., 2015; Zege et al., 2015). However, its short temporal span (2002-2011), 565 makes it unsuitable for long-term trends analysis and the study of MPF evolution. In contrast the MODIS-based MPFs and the recent dataset by Istomina et al. (2025), which bridges MERIS and OLCI, provides nearly two decades of observations better suited for studying MPF evolution and trends.



Two comprehensive intercomparison studies have evaluated some of these datasets' relative performance and consistency. Lee et al. (2024) compared MODIS-RÖSEL, MERIS-ZEGE and MODIS-LEE datasets (after standardizing them to 8-day composite and re-gridding). They also validated the three datasets against WorldView imagery and Landsat5-images classified following Wright and Polashenski (2018) methodology. The comparison revealed consistent spatial patterns among these datasets during early melt-season, with best agreement observed in June and July. However, dataset agreement deteriorated as the melt season progressed. Systematic differences emerged: MODIS-RÖSEL consistently reported the lowest MPFs values, while MODIS-LEE presented the highest. A reason for the considerably higher MPFs of MODIS-LEE, particularly during June, likely result from its reliance on normalized band ratios between the and near-infrared bands, which given its sensitivity to liquid water, could have led to wet snow, thin ice and leads to be wrongly classified as melt ponds. Despite this limitation, MODIS-LEE demonstrated the strongest correlation with high-resolution validation imagery and other melt-related variables, possibly due to its superior spatial resolution. Notably, another relevant conclusion was that none of the datasets exhibited significant MPF trends during 2002 to 2011 (except in July and August, but with no statistically significant), despite overall Arctic warming and earlier melt onset. Additionally, peak-season (July) monthly mean melt pond fractions remained below 35% across all datasets, in contrast with earlier studies (Perovich et al., 2002).

A separate intercomparison by Ding et al. (2020) examined MODIS-DING, MODIS-RÖSEL and OLCI-ISTOMINA MPF datasets for 2003–2011. All products demonstrated good agreement in MPF evolution throughout the melting season, though with distinct temporal characteristics. MODIS-DING and MODIS-RÖSEL showed slower June increases compared to OLCI-ISTOMINA, which peaked in early July, while MODIS products peaked approximately two weeks later. During late-season melt, MODIS-RÖSEL maintained elevated MPF values longer, with MODIS-DING showing intermediate behavior. Validation against observation evidence suggested MODIS-DING achieved the best overall agreement throughout the complete melt season.

### 4.1.2. High resolution regional melt pond and melt pond fraction products

High spatial resolution products have been increasingly available in the past 20 years. Campaign-based efforts have generated several high resolution regional datasets that are of invaluable support for moderate-resolutions acquisitions and pan-Arctic studies, serving as relevant validation references as well as a standalone monitoring system. Their limited regional coverage is a known trade-off for their high spatial resolution.

Appendix B, presents high resolution data that have been collected from several Arctic expeditions and research programs. It includes information on the development and source of data acquisition. High spatial resolution products have significantly contributed to sea ice studies and serve as independent validation resources (Peng et al., 2022), and as valuable training data for machine learning algorithms (Ding et al., 2020).

We have listed twelve available dataset adding information on their sources and related literature, time range, spatial coverage and resolution, datatype, availability of data (and possible URL), and visual example of the data described. Given



the heterogeneity of the datasets listed, we have also added a description of the data gathering/retrieval process for each dataset. We here give a general description of Appendix B and refer the reader to the full table in Appendix.

The datasets WV2, MEDEA, NSIDC and PANGEA are based on high resolution satellites, and are irregularly spaced in time. The remaining datasets are based on aerial photography - helicopter or drone-borne imagery from field campaigns described earlier in section 3.2.1 (e.g. HOTRAX and NPI) and in situ ship-based photographs (e.g. IceWatch datasets).

These data are available in various formats (GeoTIFF, PNG, JPEG, CSV) and have a limited geographical coverage. Some are one-time or limited-duration collections as their generation are linked to in situ campaigns. Worthy of mention are also the DLUT data (Huang et al., 2016; Lu et al., 2010) from helicopter collections, and the ship-based datasets JOIS (Tanaka et al., 2016) and PRIC-Lei (Lei et al., 2017). These dataset have been used in several studies, and although not available online, they are provided upon request.

High-resolution optical sensors Sentinel-2 (Niehaus et al., 2023) or WorldView (Wright and Polashenski, 2018) provide superior MPF retrievals with meter-scale accuracy. The SAR-based MPF retrievals (Han et al., 2016; Howell et al., 2020) also face coverage limitations and encounter fundamental challenges in distinguishing melt ponds from open water due to similar backscatter characteristics. Passive microwave-based MPF retrievals, using AMSR-E and AMSR-2 (Tanaka et al., 2016; Tanaka and Scharien 2022) on the other hand suffer from coarse spatial resolution and limited penetration in wet

conditions, making them less reliable over mixed or melting surfaces.

### 4.2. Key studies on Earth observations of melt ponds: a comprehensive collection

This section provides an overview of over forty research studies dedicated to Earth observations of melt ponds, highlighting their main objectives and methodological approach, areas of interest (AOIs), time period covered as well as data collected and their usage. Appendix C summarizes these aspects to offer a comprehensive reference for state-of-the-art Earth

observations of melt ponds, providing a perspective that helps to identify trends in melt pond research, particularly regarding their role in sea ice-related studies and advancements in remote sensing. We distinguish between validation (Va) datasets, used to assess study performance through accuracy or RSME metrics, and comparison (Co) datasets, employed when performing visual comparison of results.

This overview provides key findings and results of each research article, including description of melt pond and MPF

products and corresponding performance achieved; usage of campaign-based and multi-sensor observational records, as well as satellite-derived pan-Arctic products.

By compiling this extensive table, we have been able to extrapolate information and trends such as research trends, the evolution of methodological approaches, validation methods, changes of sampling location and summary of findings.

This section summarizes for brevity and refer the reader to the full table content in Appendix C.




- *Research Trends*: The classification of melt ponds and melt pond fraction retrievals has been a central focus of Earth observation-based research with studies consistently working to improve accuracy and reduce uncertainty. The seasonal evolution of melt ponds, along with melt pond onset followed by their characterization in relation to sea ice type, has also been an increasingly important focus of melt ponds studies.

- *Methodological Evolution*: The studies leveraged a variety of remote sensing technologies across multiple platforms, enabling refined estimation accuracy ultimately tracking seasonal and spatial variations. There has been a clear methodological evolution, from early thresholding and 3-4 class classification schemes (e.g. ice, ponds, open water). Similarly, sensor technology used has evolved significantly:

    o **Earlier studies (2000-2015)** established fundamental methods using optical sensors (MODIS, MERIS) with basic thresholding and class classification schemes. Studies like Fetterer et al., (2008) and Rösel et al. (2012) pioneered pan-Arctic MPF retrieval methods.

    o **Mid-period studies (2015-2019)** expanded to multi-sensor approaches incorporating SAR and microwave technologies. Advanced algorithms emerged, with studies like Han et al. (2006) and Scharien et al. (2017) exploring radar-based MPF detection, while classification schemes evolved to include other surface subtypes (dark/bright ponds, submerged ice).

    o **Recent work (2019-now)** emphasizes AI techniques, multi-sensor fusion and continuous long-term datasets. Studies by Niehaus et al.(2022), Xiong and Li (2025) demonstrated physical retrieval with improved accuracy and extended temporal coverage.

- *Validation and Performance Metrics:* Campaign datasets using airborne imagery (e.g. Canon EOS, UAVs, helicopter surveys) provide standard validation, with studies achieving accuracies ranging from 85-96% for MP classifications (Wright and Polashenski, 2018; Lee et al., 2020) and RMSE values of 0.05-0.18 for MPF retrievals (Peng et al., 2020; Lee et al., 2020). There is also a noticeable improvement on more recent studies regarding validation practices, which have become increasingly rigorous with multiple ground-truth datasets and cross-comparison between different MPF products as demonstrated in comprehensive intercomparison by Lee et al. (2024) and Ding et al. (2020).

- *Geographic and Ice Type Focus*: The literature shows concentrated research in accessible and relevant Arctic regions: Canadian Arctic Archipelago (15 studies), Beaufort/Chukchi Seas (12 studies), and Svalbard/Fram Strait (8 studies). Studies increasingly distinguish between first-year ice (FYI) and multi-year ice (MYI), with some documenting how the ongoing MYI-to-FYI transition might result in increased pond formation given FYI's smoother and flatter topography which facilitates wider pond spreading (Polashenski et al., 2012; Rösel and Kaleschke, 2012; Malinka et al., 2018) presenting wider and shallower features comparatively to MYI (Eicken et al., 2004; Polashenski et al., 2012).



- *Findings :*

    o **Seasonal Evolution:** Multiple studies show consistent seasonal patterns, with melt pond fractions presenting low values in May, increasing substantially in June and peaking in July. To be noted that some studies also noted a decay occurring half month later than previously reported (Ding et al., 2020), with greater interannual variability during decay phases.

    o **Long-term changes**: The pan-Arctic multi-annual MPF datasets, allowed to reveal complex temporal trends. While some studies found negative trend (2000-2011) linked to declining sea ice extent (Rösel and Kaleschke, 2012), despite MYI decline, more recent analyses show moderate increases (+0.75% per decade, Istomina et al. (2025)) with pronounced regional variability (-10% to +20% per decade). Feng et al. (2021) documented earlier melt onset and extended melting periods driving these changes.

    o **Spatial Patterns:** Studies reveal both consistent large-scale patterns but evolving regional dynamics. MYI-dominated regions, such as north of Greenland and Central Arctic Ocean, consistently show lower MPF values, while seasonal ice zones exhibit higher fractions. However, significant spatial shifts are occurring. Kara and Laptev seas are showing positive trends (Xiong and Ren, 2023), replacing earlier patterns seen on the Beaufort and Chukchi Seas (Ding et al., 2019; Istomina et al., 2025). This redistribution reflects earlier melt onset, FYI expansion in previously MYI-dominated regions, and shifting pond formation regimes across Arctic basins.

    o **Contradictory findings**: Despite the broad consensus linking ice type to MPF, contradictory findings have emerged that challenge this relationship. Perovich and Polashenski (2012) found no significant MPF trend between 2000–2011, despite substantial decreases in MYI. Their research suggested that local ice deformation and snow distribution were more influential factors than ice age in determining pond formation. This conclusion was further supported by Webster et al. (2015), who demonstrated that snow depth distribution and ice topography were more critical factors in melt pond formation than ice type alone. While Peng et al. (2022) reported long-term MPF declines which were attributed to declining sea ice extent, more recent studies (covering the 2000-2019 melting seasons May-August) documented significant upward trends in both overall MPF and specifically in multi-year ice MPF (Xiong and Ren, 2023; Istomina, 2025). These changes appear to be linked to earlier melt onset and extended melting periods, a trend noted by multiple researchers (Perovich et al., 2008; Markus et al., 2009; Rösel and Kaleschke, 2012; Pistone et al., 2014). SHEBA campaign data (Perovich et al., 2003) have revealed a constant linear relationship between pond fraction and pond depth, however more recent studies studies, have shown no relationship between pond depth and fraction (Polashenski et al., 2020, Webster et al., 2022a and Buckley et al., 2023).





## 5. Discussions

### 5.1. Considerations on knowledge gaps

Several critical limitations continue to hinder the comprehensive understanding and monitoring of melt ponds, despite the large interest in their dynamics. Knowledge of melt pond characteristics has traditionally been derived from observations conducted over land-fast ice (Yackel and Barber, 2000; Perovich et al., 2002; Eicken et al., 2004; Polashenski et al., 2012), primarily due to the relative ease of access and the feasibility of repeat in situ measurements using ships or aircraft. Although several campaign-based datasets have extended coverage into parts of the Arctic Ocean, these efforts often lack spatial and

temporal consistency, constrained by the significant logistical challenges associated with Arctic fieldwork. This has introduced a systematic observational bias, as remote and logistically inaccessible regions, characterized by different sea ice regimes, remain under-sampled due to the high costs and inherent risks involved in accessing them.

Addressing the knowledge gaps in melt pond observation requires a detailed examination of three interconnected challenges: (1) limited spatial coverage in remote and under-observed regions, particularly the central and high Arctic, which contributes

to observational biases related to ice type, as multi-year ice (MYI) remains substantially underrepresented compared to first-year ice (FYI), complicating comparative analyses; (2) temporal limitations, such as restricted seasonal sampling and infrequent satellite revisit times, which restrict continuous monitoring; (3) methodological challenges in data collection which result in scarcity of high-quality, labelled datasets that hampers the effectiveness of machine learning approaches and (4) sensor-based constraints. Each of these areas present important opportunities to advance melt pond research through

enhanced observation strategies and improved data integration:

*(1) Under-observed regions and ice type bias*: Coverage gaps are particularly problematic in the central Arctic Ocean, where data assimilation techniques are required to integrate satellite observations, ground observations, and numerical models. Most in situ observations remain confined to accessible coastal areas of lands-fast ice, leaving the remote central Arctic ocean very under-sampled. As an example, the MOSAiC expedition revealed regional

representativeness issues: along the central transect, average summer pond coverage was ~14%, compared to ~20% at the ice edge, highlighting the limitations of point-based observations for capturing broader spatial variability (Webster et al., 2022b). A particular example is the Canadian Arctic Archipelago (CAA). The CCA region exemplifies both the progress and limitations in melt pond research. Although several studies have addressed melt ponds on FYI in this region (Yackel and Barber, 2000; Scharien and Yackel, 2005), melt ponds over both MYI and

FYI remain under-investigated, despite recent efforts (Scharien et al., 2007; Wright and Polashenski, 2018; Buckley et al., 2020; Li et al., 2020). Tracking ponds on FYI in the CAA presents additional technical challenges, as missions like Copernicus Sentinel-2 are constrained to within 20 km of the coast which is a major limitation, given that FYI often retreats offshore along western Canada and Alaska. Overall, high-resolution and campaign-based studies are concentrated in a few regions, such as the CAA/North Greenland (Buckley et al.,2023; Howell et al.,

2020; Geldsetzer et al., 2023), North Svalbard/Nansen Basin (Renner et al., 2013; Mäkynen et al., 2014; Divine et





al., 2015; Fors et al., 2017) and the Beaufort/Chukchi Seas (Tschudi et al,. 2008; Kern et al., 2010; Webster et al., 2015; Han et al., 2016). These campaigns, often limited to specific regions and years, leave large portions of the Arctic basin under-sampled.

*(2) Temporal and Seasonal data gaps*: Airborne campaigns have demonstrated the potential to extract melt pond parameters from imagery, but most remain also temporally constrained. Many are limited to single-season studies (Miao et al., 2015) or narrow multi-year analyses (Tschudi et al., 2001; Perovich, 2002), offering only temporal snapshots rather than continuous monitoring. The temporal and seasonal data gap adds another layer of limitation, as it limits the ability to capture melt pond evolution processes. As a result, the ability to capture key transition phases in MP evolution, such as melt onset or the winter–spring transition, remains limited (Li et al., 2020). One example is the NSIDC MP dataset (listed in Appendix B), only 101 out of 1056 acquired images were suitable for pond statistics, and these were unevenly distributed across time (Fetterer et al., 2008). Finally, optical-based satellite systems face persistent challenges during the melt season due to cloud cover and low illumination, further reducing usable data during critical periods.

*(3) Metrics and validation:* Even in regions and periods with relatively good coverage, key physical metrics such as melt pond depth and volume remain sparsely measured and validated (Fuchs et al., 2024; Xiong and Li, 2025). In fact, differences in melt ponds over different sea ice types, in terms of depths, sizes but also evolution throughout the melting season are still a source of ambiguity. Further research is needed to generalise conclusions and parameters across ice types and sensor modalities. While dedicated in situ and airborne measurements could significantly improve our understanding of melting sea ice surfaces and complex pond bottom structures (Buckley et al., 2023), such studies would be most effective when collected beneath coincident satellite orbits. However, while satellite observations offer broader spatial coverage compared to in situ or airborne campaigns, they still suffer from limitations in resolution, revisit frequency, and retrieval accuracy, leaving room for improvement. As a result, there is a significant shortage of high-quality data in this area of research, especially considering the considerable demand for training datasets for data-hungry deep learning models (Sun et al., 2022). For example, MYI in particular is under-represented in high-resolution pond mapping, with most datasets being optimized for FYI or mixed ice types (Istomina et al., 2015; Tilling et al., 2020; Geldsetzer et al., 2023).

*(4) Sensor-based constraints*: Despite their potential, microwave and SAR-based melt pond retrieval methods still lag behind in situ methods, partly due to sea ice heterogeneity and suboptimal radar parameter settings (e.g., frequency, incidence angles and polarisation). Current research limitations stem from the narrow focus of existing studies, as efforts have been made towards single polarisation HH and mostly focused on RADARSAT over FYI regions (Scharien et al., 2017; Li et al., 2017; Ramjan et al., 2018; Howell et al., 2020). This approach has left significant gaps in understanding SAR behaviour across different ice types and sensor configurations.



### 5.2. Critical role of melt ponds in future cryospheric research

Improved understanding of melt ponds has the potential to significantly advance cryospheric research. Key areas of interest
include: (1) the role of melt pond fraction in enhancing sea ice extent (SIE) predictions; (2) its influence on sea ice
concentration (SIC) retrievals; and (3) the discrepancies between observed and modelled pond evolution. Persistent
challenges in melt pond parameterisation further limit the accuracy of current sea ice models. These issues highlight the
critical need for more comprehensive observational datasets and refined modelling approaches. The identified gaps inform
priority areas for future research, which are outlined in the concluding section.

### 5.2.1. Melt pond fraction and its role in sea ice extent predictions

Improved understanding of MPF and melt season duration is critical for advancing predictions of sea ice extent and assessing
the Arctic's radiative balance. Arctic summer sea ice has declined markedly, with a record minimum in 2012 (Stroeve et al.,
2012a), exceeding prior projections. Given the importance of summer SIE for both climate feedback and increasing Arctic
activity, more accurate seasonal forecasting is urgently needed (Eicken et al., 2013).
Several studies have demonstrated the predictive potential of MPF. Schröder et al. (2014) found that spring MPF,
particularly in May, strongly correlates with September SIE, driven by an albedo-melt feedback. In contrast, Liu et al. (2015)
concluded that MPF integrated over early to mid-summer offers greater predictive skill and suggested other factors, such as
melt onset and surface temperatures, may also influence model performance. Both studies underscore discrepancies between
modelled and observed MPF, highlighting the need for better integration of satellite data into models. Howell et al. (2020),
examined peak MPF in the Canadian Arctic Archipelago (CAA) and reported clearer predictive signals than pan-Arctic
studies, likely due to the CAA's more stable ice regime. Feng et al. (2021) extended the analysis spatially and temporally,
identifying strong correlations between early summer MPF and September SIE anomalies across the Beaufort, Chukchi,
Eastern Siberian seas, and the CAA. MPF anomalies were also linked to atmospheric patterns, such as negative Arctic
Oscillation phases, and even to distant climate effects, including winter temperatures in northern China.
Collectively, these findings emphasize the importance of both the time and spatial scale of MPF in sea ice forecasting.
Despite evidence linking May–July MPF to September ice conditions, this relationship remains underexplored. Future
research should prioritize the integration of satellite observations and model outputs, with a focus on understanding
interactions between MPF evolution, atmospheric forcing, and sea ice thermodynamics. These efforts are essential for
developing more accurate, physically based forecasting tools in a rapidly changing Arctic.

### 5.2.2. Impact of melt ponds on sea ice concentration retrievals

Sea ice concentration is a critical parameter for monitoring long-term changes in the Arctic and is extensively assimilated
into numerical models. This widespread use is largely enabled by the continuous passive microwave satellite record since
1978, which offers the significant advantage of providing consistent observations regardless of cloud cover or darkness





during the polar night. Instruments such as AMSR-E and SSMIS capitalize on these capabilities to deliver near-daily SIC

data essential for climate monitoring and modeling. However, during the melt season, the presence of melt ponds significantly degrades SIC retrievals derived from satellite brightness temperatures. Melt ponds lower surface emissivity, causing algorithms to interpret pond-covered ice as open water, leading to systematic underestimation of SIC (Cavalieri et al., 1990; Comiso and Kwok, 1996). Additional uncertainties arise from changes in snow and ice surface properties. These limitations highlight the need to account for melt pond influence in SIC products, particularly during summer, to improve the

accuracy of sea ice monitoring and data assimilation.

As an example, Kern et al. (2016) found that when MPF reaches 40%, SIC is underestimated by 14–26%, depending on the algorithm used, while underestimation is negligible when MPF is around 20%. The same study also concluded that no single algorithm consistently outperforms others, but those more sensitive to melt ponds could be optimized more effectively, as the influence of unknown snow and sea ice surface property variations is less pronounced. Kern et al. (2020) compared 10

passive microwave satellite derived SIC products against MODIS and ship measurements, finding significant discrepancies linked to melt pond effects, which reveal major challenges in accurately retrieving SIC during melt seasons. This emphasizes the unknown contribution of melt pond signature within ice. Zhao et al. (2021) evaluated three SIC products against ship-based and MODIS observations from 60 Arctic cruises (2006–2020), and found a strong relationship between MPF and SIC underestimation: at 50% MPF, SIC was underestimated by 7–20%. Furthermore, this study also showed that applying MPF-

based correction significantly reduced SIC bias. Collectively, these studies strongly support the need for improved characterization of melt pond signatures, specifically, their influence on brightness temperature and SIC retrieval algorithms, and for their explicit inclusion in the retrieval process, to improve the accuracy of satellite-derived SIC, an ongoing and well-recognized challenge in remote sensing of Arctic sea ice.

### 5.2.3. Use of observational data in refining melt pond representations in models

Melt pond behavior is a known source of uncertainty in sea ice models (e.g. Flocco et al., 2012; Holland et al., 2012; Webster et al., 2015). Our understanding of melt pond evolution remains limited, and current climate models lack the resolution to capture such small-scale features. Long-term, pan-Arctic, high-resolution observations would be invaluable in constraining model parameters across diverse conditions. Key observations for melt pond studies have been identified from both modelling and observational perspectives. For instance, Ebert and Curry (1993) implicitly represented melt ponds via a

temperature-based albedo scheme, while more recent results emphasize the benefits of explicit melt pond representation. Highlighting this, Diamond et al. (2024) found that in 'near-future' simulations, implicit schemes never produced ice-free summers, whereas explicit schemes led to ice-free conditions 35% of the time, associated with Arctic air temperature increases of 5–8°C. This temperature difference is solely due to the choice of scheme. Driscoll et al. (2024a) further demonstrated that sea ice predictions are highly sensitive to melt pond parameter values, as also noted by Flocco and

Feltham (2007). While future developments may reduce differences between schemes, a divergence is also possible, underscoring the need for accurate parameterisation informed by detailed observations and potentially multi-modal methods.





Diamond et al. (2024) do not develop their own scheme but apply an existing one to assess impact. These results build on years of development and testing of melt pond models, including physical schemes (Taylor and Feltham, 2004; Lüthje et al., 2006; Flocco and Feltham, 2007; Skyllingstad et al., 2007; Hunke et al., 2013; Wang et al., 2024) and data-driven 830 approaches (Driscoll et al., 2024a,b). Observations have been essential to their development.

The 1997 - 1998 SHEBA campaign was central in refining parameters like pond fraction, depth, albedo, and drainage in most subsequent studies. MODIS data validated pond coverage and climatology in Flocco et al. (2012). Aerial imagery from Eicken et al. (2004) and in situ albedo measurements (e.g. Yackel et al., (2000) and Hanesiak et al., (2001)) have been used to evaluate schemes like Flocco and Feltham (2007) and Scott and Feltham (2010). A full review is beyond scope, but the 835 role of field campaigns and diverse data sources is clear. Despite the physically based foundation of these schemes, uncertainties remain, partly due to sparse observations. Sterlin et al. (2020) noted disagreement in melt pond fraction trends, raising concerns about model sensitivity to surface forcing. Differences in how melt ponds are distributed across ice thickness categories further affect sea ice evolution. Using MOSAiC data, Webster et al. (2022b) demonstrated the importance of drainage parameters and pond formation sequences, with new schemes showing differences of over 50% in 840 simulated pond coverage (Wang et al., 2024).

The importance of accurate melt pond parametrisations is indisputable. Their future likely lies in hybrid, data-informed approaches capable of leveraging multiple data streams. This is especially relevant as data-driven methods may facilitate the integration of varied sources. Diamond et al. (2024) noted that only ~11% of the 126 CMIP6 model configurations used explicit melt pond schemes, despite their major climate impact. Observations will continue to shape modelling efforts.

**5.2.4. Filling the knowledge gap: Future research and trends**

This section outlines priority areas for future research, along with emerging trends enabled by current missions and algorithmic advances, that hold the potential to close existing gaps in melt pond observational knowledge and enhance its impact across cryospheric science:

- *Physical Processes and Model Integration*: Understanding the physical processes driving melt pond fraction
evolution remains a key challenge. The relationship between pond depths, pond fraction, snow cover and type, could potentially benefit from the understanding of the interplay between surface physics and MPF variability. Emerging efforts to couple remote sensing observation with sea ice components of GCMs, hold the potential of improving and simulating pond onset, evolution and refreeze states. The significant influence of physical features like snow cover (Webster et al., 2015) or snow depth (Kim et al., 2018; Toyoda et al., 2024) underscores the
importance coupling these parameters in a modeling framework to resolve inconsistencies between MPF datasets and improve melt onset detection.

- *Improving Validation and Benchmarking Datasets*: Existing discrepancies between pan-Arctic MPF datasets (Table 2) in terms of spatial distribution and temporal behaviours, particularly in early season melt and MYI zones (Ding et al., 2020; Peng et al., 2022; Xiong and Ren, 2023), urges the need for improved validation protocols. Satellite-





derived MPF values still rely heavily on sporadic and geographically limited field campaigns. A coordinated effort to expand ground truth datasets, including autonomous platforms and better synchronization with satellite overpasses, along with open-access repositories, is essential. Furthermore, the absence of a standardized benchmarking/validation dataset hampers algorithm intercomparison and development. High-resolution imagery from commercial satellites (e.g., PlanetScope, WorldView-3) offers a promising, though underutilized, resource for
validation.

- *Generalised Microwave and SAR*: Despite their potential, microwave and SAR-based melt pond retrieval methods still lags behind in situ methods and limited research focus on single polarisation HH over FY regions. To address these limitations, extensive studies comparing surface roughness and incidence angles across different SAR frequencies and polarizations over different sea ice surface types could support MPs observations. Multifrequency
combinations show promise in improving MPF retrievals by reducing frequency-specific ambiguities (Kern et al., 2010). Future work would benefit from assessing multifrequency (C, L, and X band) and polarimetric behaviour during seasonal transitions in FYI and MYI (Scharien et al., 2010), alongside expanded summer-season studies and improved scattering models (Han et al., 2016). Additionally, increased in situ validation datasets could improve SAR-based method assessment, which currently relies mostly on visual comparison rather than true ground truth.

- *Machine Learning and Deep Learning Applications*: The rapid growth in AI applications (Wright, 2020; Ding et al., 2020; Feng et al., 2021; Peng et al., 2022), suggests that this trend will increase, in particular since key benefit of AI applications is the ability to bypass intermediate classification steps and move toward direct MPF estimations (Wright and Polashenski, 2020; Buckley et al., 2020; Niehaus et al., 2022). However, current models face significant limitations due to sparse labeled training data, leading to overfitting and poor generalization across
different regions and times. Limited training samples result in region- and season-dependent errors in MPF estimations, caused by the strong spatial and temporal variability of sea ice and melt pond reflectance. Current scarcity of in situ data may not yet be sufficient to develop robust ML-based MPF retrievals (Wright and Polashenski, 2020; Xiong and Ren, 2023). Addressing these challenges requires training models on larger and more diverse datasets, potentially through crowdsourced labeling or semi-supervised learning.

- *Standardization in surface classification*: The lack of a common taxonomy for cryospheric surface types, particularly regarding melt pond stages, introduces complexity and challenges. Surface class definitions vary considerably across the literature (Webster el al., 2015; Huang et al., 2016; Wright and Polashenski, 2018; Buckley et al., 2020), making it difficult to generalize findings. A standardized classification framework would facilitate reproducibility and improve model training, along with address. frequent misclassifications in mixed or ambiguous
pixels.

- *Sensor Synergies and Multi-Mission Approaches*: To overcome single-sensor limitations, integrating data from multiple sensor types (e.g., SAR, passive microwave, and optical) holds promise. Ensemble observations combining multi-polarization, multi-frequency SAR with microwave and optical could help resolve detection ambiguities and



increase spatial and temporal coherence. Few studies combine multiple sensors for single product generation. Moreover, hyperspectral and active optical sensors remain underutilized, yet could benefit distinction of pond types and mapping of pond depths, which to date does not exist.

## 6. Conclusions

This comprehensive review of observational methods for Arctic sea ice melt ponds outlines recent advances and current capabilities, while also identifying the persistent challenges that continue to constrain observations of these key components of the Arctic climate system.

The review examines how EO sensors operating across optical and microwave domains respond to different physical properties of melt ponds, and outlines domain-specific signatures that enable melt pond detection or characterization.

EO data from spaceborne platforms provide long-term and regional-to-pan-Arctic monitoring capabilities, while also validating existing model outputs and directly inform the development of melt pond schemes, particularly with respect to spatial coverage, timing of pond onset and drainage, as well as connections to ice concentration and type. Aerial imagery and very-high-resolution datasets from field campaigns further support the refinement of parameterisations by resolving small-scale variability, and they also provide essential ground truth data for validation of remote sensing algorithms.

Melt pond data processing has evolved from early statistical approaches to sophisticated methods, including spectral unmixing, radiative transfer model-based algorithms and machine learning techniques, with a parallel transition from single-sensor to multi-sensor fusion strategies. This progression reflects the field's ongoing search for optimal solutions, as different techniques prove better suited to specific datasets combination thereof. Optical systems currently lead pan-Artic product development despite limitations imposed by illumination and cloud cover, while microwave systems offer all-weather capabilities but struggle to distinguish melt ponds from water, limiting their operational use for large-scale applications. A fundamental trade-off exists between pan-Arctic datasets, which provide continuous monitoring at coarser resolutions, while high-resolution regional products deliver meter-scale accuracy but limited spatio-temporal coverage.

Analysis of more than forty leading studies on EO-based melt pond observations reveals that established paradigms of melt pond formation are being challenged, with local factors proving as influential as ice type. Differences between melt pond fractions, in particular, the lower agreement throughout the melt season and heterogenous temporal patterns and spatial dynamics findings reveals significant retrieval uncertainties.

Three major constraints further exacerbate technical limitations: geographical sampling biases toward accessible coastal regions leave the central Arctic under-observed (particularly problematic given the ongoing MYI to FYI transition, as MP behaviour across these different ice types remains poorly characterised at broad scales); temporal constraints prevent capture of critical transition phases; and methodological inconsistencies including absent standardized classification schemes and validation protocols hamper algorithm development and inter-study comparisons.



These observational limitations have cascading effects across multiple domains: melt pond observations demonstrate strong predictive potential for sea ice extent forecasting through albedo-feedback mechanisms, yet data quality limitations prevent full utilization of this capability. During melt seasons, ponds substantially degrade sea ice concentration retrievals, causing systematic underestimation that requires better characterization. Explicit parameterization of melt ponds is crucial for reliable climate model projections, as existing models are highly sensitive to parameterization choices; however, such
schemes remain insufficiently implemented.

This review highlights priority areas for future research to address current limitations. Key directions include enhanced physical process coupling between surface dynamics and melt pond variability; expanded ground truth datasets through coordinated observation campaigns; the integration of observational data with sea ice models; standardized surface classification frameworks; larger machine learning training datasets; and leveraging from methodological evolution and
multi-sensor synergies. The field would also benefit from a dedicated, centralized platform for accessing melt pond datasets, along with systematic co-location of EO products, field observations and model diagnostics. Given the importance of accurately representing Arctic feedback, such investments represent both scientific opportunities and part of essential building blocks to contribute to climate research.









# 7. Appendices

**Appendix A: Description of MP algorithms or combination of techniques used for MP and MPF applications**

| | Technique | Description of techniques, main limitations and outputs |
|---|---|---|
| **Spectral-based methods** | Thresholding | Applies value cutoffs to values of spectral bands or indices separating pixels into different classes (e.g., melt ponds, sea ice, open water). **Examples:** NDWI thresholds for water identification; RGB color thresholds from histogram analysis; red-channel intensity cutoffs; and panchromatic intensity thresholds accounting for neighboring pixel effects. **Limitation:** Sensitive to illumination and atmospheric effects; requires manual threshold selection. **Output:** Binary or multi-class classification maps. |
| | Spectral unmixing (Fixed spectral) | Decomposes mixed pixels into factional contributions of different surface types (sea ice, snow, melt ponds) using known spectral reflectance values (endmembers) from field measurements. Physically interpretable results; well-established methodology. **Limitations:** Assumes constant reflectance for each surface type; sensitive to endmember quality and representativeness. **Output:** Fractional coverage maps. |
| | Spectral unmixing (Dynamic endmember) | Advanced spectral unmixing methods that address fixed reflectance limitations: |
| | | **Fully Constrained Least Squares (FCLS):** Estimates fractional surface coverage with non-negativity and sum-to-one constraints. **Limitations:** Dependent on large, representative spectral libraries limited by reference dataset accuracy. |
| | | **MESMA (Multiple Endmember Spectral Mixture Analysis):** Allows number/type of endmembers to vary per pixel, improving flexibility over standard unmixing. **Limitations:** Still depends on quality of reference spectra; computationally more demanding |
| | MPD1 (Melt Pond Detection 1) | Uses a physical model to dynamically derive reflectance based on optical properties; simulates BRDF for melt ponds, ice, snow. **Limitations:** Computationally complex; mode assumptions may not capture all real-world variability. **Output:** Fractional coverage maps. |
| | MPD2 (Melt Pond Detection 2) | Built on MPD1 with physical forward model and prior info (e.g. temperature history) to improve retrievals, also includes open ocean. **Limitations:** Requires prior physical information; still limited by accuracy of physical inputs and computational cost. **Output:** Fractional coverage maps. |
| | LinearPolar algorithm | Transforms spectral data into polar coordinates along melt pond and sea ice axes; angle ($\theta$) distinguishes pond/ice, radius (r) captures variability. **Limitations:** Requires optimal band selection; still sensitive to spectral variability and noise; limited validation under some conditions. |
| **Polarization-based techniques** | Geophysical Inversion | Uses statistical variations in SAR backscatter ($\sigma°$) to estimate melt pond morphology, albedo, and surface properties; empirical regression relates $\sigma°$ to geophysical variables. Relies on empirical correlations; accuracy limited by variability in surface conditions and calibration. |



| | | |
|---|---|---|
| | Polarization Ratio Methods (VV/HH, cross-pol) | Uses ratios of SAR backscatter between polarizations (VV/HH, cross-pol) to distinguish surface roughness. **Advantages:** Exploits fundamental scattering differences; relatively simple implementation. **Limitations:** Sensitive to incidence angle and surface roughness; prone to overestimation. **Output:** Surface roughness classifications and MPF estimates. |
| | Compact Polarization SAR | Uses hybrid polarization parameters with tilted-Bragg scattering models for MPF retrieval. **Advantages:** Reduced data volume compared to full polarimetric; maintains scattering information. **Limitations:** Systematic overestimation; model assumptions may not capture all scattering behavior. **Output:** MPF retrievals |
| Traditional machine learning methods | PCA (Principal Component Analysis) | Reduces dataset complexity by transforming variables into uncorrelated components for surface type separation. **Advantages:** No training data required (unsupervised approach); identifies main spectral variations. **Limitations:** Sensitive to noise; spectral overlap reduces accuracy; limited physical interpretability. **Output:** Classification maps based on principal components. |
| | Maximum Likelihood Classification | Uses labeled training data to compute class statistics (mean, covariance) for pixel classification. **Advantages:** Well-established statistical foundation; handles multiple classes effectively. **Limitations:** Requires representative training data; assumes normal distribution of classes. **Output:** Multi-class surface type maps with probability estimates. |
| | Bayesian Maximum Likelihood (BML) Classification | Incorporates prior knowledge using Bayes' theorem for SAR feature classification into surface types. **Advantages:** Uses prior class probabilities; theoretically robust framework. **Limitations:** Computationally complex; requires accurate priors and sufficient training data. **Output:** Probabilistic surface type classifications. |
| | Decision Tree + Regression models | Decision-tree classification of imagery followed by regression using polarimetric and texture parameters. **Advantages:** Interpretable decision rules; handles mixed data types. **Limitations:** Accuracy depends on training data quality; regression limited by linear assumptions. **Output:** Classification maps and continuous MPF estimates. |
| | Object-based Classification + Random Forest | Combination of 2 methods: groups neighboring pixels into objects via segmentation; Random Forest classifies objects into multiple classes. **Advantages:** Incorporates spatial context; handles complex class boundaries. **Limitations:** Computationally intensive; accuracy depends on segmentation quality; requires careful parameter tuning. **Output:** Object-based classification maps |
| | Multiscale Segmentation + Aggregation | Segments data into ice/melt pond classes using multiscale approaches (e.g. SAR data); derives binary water/ice maps. **Advantages:** Handles scale-dependent features; automated processing. **Limitations:** Underestimates MPF (misses small ponds); moderate agreement with reference data. **Output:** Binary classification maps and MPF statistics. |
| Deep learning & neural-based architectures | MLP (Multilayer Perceptron) | Neural network with hidden layers trained to replicate spectral unmixing outputs. **Advantages:** Accelerates traditional spectral unmixing; learns non-linear relationships. **Limitations:** Trained on 8-day composites; may not represent full data variability. **Output:** Fractional coverage maps. |
| | Multi-Neural Network (MNN) + Multinomial Logistic Regression (MLR) | MNN classifies reflectance into surface types; MLR predicts MPF from classified data. **Advantages:** Two-stage approach improves accuracy; handles top-of-atmosphere reflectance. **Limitations:** Moderate accuracy (85%); performance affected by temporal averaging. **Output:** Surface type classifications and continuous MPF predictions |



| | E-DNN (Ensemble Deep Neural Network) | Combines multiple neural networks through ensemble learning with 3 hidden layers. **Advantages:** Improved performance through ensemble approach; incorporates all spectral bands. **Limitations:** High computational cost; temporal mismatch between training and prediction data. **Output:** Fractional coverage maps. |
|---|---|---|
| | SAE (Stacked Autoencoder) | Learns compressed spectral representations through multiple encoding/decoding layers. **Advantages:** Improved accuracy over earlier approaches; learns efficient data representations. **Limitations:** Sensitive to feature selection; risk of overfitting; still affected by MODIS limitations. **Output:** Fractional coverage maps. |
| | GA-BPNN (Genetic Algorithm-optimized Back-Propagation Neural Network) | Uses genetic algorithms to optimize network structure; incorporates temporal filtering. **Advantages:** Optimized network architecture; addresses temporal inconsistencies. **Limitations:** Complex training procedure; requires extensive parameter tuning. **Output:** Fractional coverage maps. |
| Visual & statistical-based | Regression models | Statistical modeling that establishes quantitative relationships between different sensor measurements through linear/non-linear regression and texture analysis techniques. Example: Relating microwave backscatter/texture from SA to optical MPF product; Linear regression of passive microwave brightness temperature gradients against ship-based observations. **Advantages:** Combines complementary sensor information; quantifies relationships statistically. **Limitations:** Sensitive to incidence angle; requires co-located. **Output:** Empirical models relating input parameters to target variables (e.g. MPFs) |
| | Visual and Correlations Analyses comparisons | Comparative assessment techniques that examine relationships between different datasets through visual inspection and statistical correlation measures. Example: SAR imagery and optical-based MPF products. **Advantages:** Simple implementation; identifies sensor relationships. **Limitations:** Subjective visual assessment. **Output:** Qualitative relationships and correlation coefficients. |




## Appendix B: Melt pond and MPF observation open source datasets

| Dataset name | Reference | Time range | Spatial Coverage | Spatial Resolution | Format | Data access | Preview Example |
|---|---|---|---|---|---|---|---|
| **WorldView -2 (WW2) MPF** | Wright and Polashenski (2019) | 2000 to 2015 | Beaufort Sea: 72.0° N-128.0° W | 0.46 and 1.84 m | GeoTIFF | https://arcticdata.io/catalog/view/doi:10.18739/A22Z12P4J | |
| The WV2 MPF, based on WorldView satellite imagery, results from being processed using the Open Source Sea-Ice Processing algorithm developed by Wright and Polashenski (2018). | | | | | | | |
| **MEDEA** | Webster et al. (2015) | May to August 1999 to 2014 | Site locations within 69–85.5 °N | 5 to 25 km | PNG, GeoTIFF, JPEG, ASCII, CSV | http://psc.apl.uw.edu/melt-pond-data/ | |
| The MEDEA (Melt Pond Fraction Statistics From High Resolution Satellite) Images, are retrieved by the Polar Science Center, University of Washington, also based on visible bands of high-resolution optical data (1 m resolution). The statistics of melt pond coverages were retrieved in locations in the Beaufort Sea, Chukchi Sea, the Canadian Arctic, the Fram Strait, and the East Siberian Sea from May to August, following Webster et al. (2015). | | | | | | | |
| **NSIDC** | Fetterer et al. (2008) | 1999 to 2000 | 72.8–85.1°N | 1 x 1 m | PNG, GeoTIFF, JPEG, ASCII, CSV | https://nsidc.org/data/g02159/versions/1#anchor-2 | |
| The NSIDC dataset (by the National Snow and Ice Data Center) is based on high resolution imagery classified following Fetterer et al. (2008), and corresponds to the site location of the SHEBA experiment (Beaufort Sea, East Siberian Sea, Canadian Arctic and Fram Strait). | | | | | | | |
| **TransArc** | Nicolaus et al. (2012) | August to October 2011 | 83.1–86.3°N | 50 m to 1 km | CSV | https://doi.pangaea.de/10.1594/PANGAEA.803312 | n.a. |
| The TransArc melt pond observations were collected from the ice breaker RV Polarstern during the Germany Trans-Polar cruise ARK-XXVI/3 (Nicolaus et al., 2012), through hourly observations from the bridge of the research vessel on 29 August and 6 September. | | | | | | | |
| **PANGEA** | Niehaus et al. (2022) | 2017 to 2021 | Up to 82.3°N | 10 x 10 m | NetCDF | https://doi.pangaea.de/10.1594/PANGAEA.950885?format=html#download | |
| The PANGAEA is based on 31 scenes of cloud-free Sentinel-2 data by Niehaus et al. (2022). A special focus was given to cover the Multidisciplinary Drifting Observatory for the Study of Arctic Climate (MOSAiC), in fact all scenes from 2020 cover the expedition path. | | | | | | | |





| HOTRAX | Perovich et al. (2009) | August to September | 74.4–86.1°N | 57 x 70 m | JPEG, PNG | t[https://zenodo.org/record/6602409#.ZFuxlHZBzmE](https://zenodo.org/record/6602409#.ZFuxlHZBzmE) | |
|---|---|---|---|---|---|---|---|
| | The HOTRAX MPF were collected during the Healy Oden Trans-Arctic Expedition (HOTRAX) by the Polar Science Center, University of Washington (Perovich et al., 2009). The dataset contains mapped melt pond zones through deep learning approaches (Sudakow et al., 2022) from aerial photographs obtained during helicopter photography flights, as part of HOTRAX campaign. | | | | | | |
| NPI | Divine et al. (2015) | July to August 2012 | 81.4–82.°N | 60 x 40 m | Tabular | [https://data.npolar.no/dataset/5de6b1e4-b62f-4bd4-889c-8eb7bb862d3b](https://data.npolar.no/dataset/5de6b1e4-b62f-4bd4-889c-8eb7bb862d3b) | n.a. |
| | The NPI data were collected by the Norwegian Polar Institute (NPI) during the ICE12 field campaign on Arctic sea ice north of Svalbard during the summer of 2012.The dataset comprises of fractions of three surface types (bare ice, melt ponds and open ocean water) from which MPF were calculated (Divine et al., 2015), along the flight tracks calculated from images collected by a helicopter-borne camera system. | | | | | | |
| IceWatch | Norwegian Meteorological institute | 2006 to present | Depends on the cruise | Depends on the cruise | CSV, JSON, GeoJSON, ASPeCT, Sigrid3, JPEG | [https://cryo.met.no/en/icewatch](https://cryo.met.no/en/icewatch) | |
| | The IceWatch observations contain cruise data in multiple formats which are available to download of MPF are collected and made available through the IceWatch community. The database is hosted and maintained by the Norwegian Ice service (part of the Norwegian Meteorological Institute) and is continuously growing as it is community contributed via their web-tool named ASSIST. | | | | | | |
| THINICE | Rivière et al. (2024) | August 2022 | 70-82°N, 20°W-40°E | Up to 1 m. | netCDF | [https://ralithinice.aeris-data.fr/data-catalog/](https://ralithinice.aeris-data.fr/data-catalog/) [https://data.ceda.ac.uk/badc/arcticcyclones](https://data.ceda.ac.uk/badc/arcticcyclones) [https://a.windbornesystems.com/syzygy/mst2/](https://a.windbornesystems.com/syzygy/mst2/) | n.a. |
| | The THINICE campaign was the first dedicated to summertime Arctic cyclones and interactions (with clouds, sea ice, polar vortex). Two aircraft as well as WindBorne balloons took extensive and detailed measurements to create this dataset. The dataset includes not only detailed surface sea ice characteristics but also cloud properties, jet properties, turbulent fluxes and more. Data are available through AERIS (RALI-THINICE), CEDA, as well as WindBorne systems. | | | | | | |
| DLUT MPF | Huang et al. (2016) | August-Sep.2008 & July -Sept 2010 | 71.9–81.0°N | 98 × 67 m or ranges from 147 × 100 m to 490 × 335 m | n.a. | Contact the lead author. | n.a. |
| | The observations were collected during two Chinese Arctic Research Expeditions by the Dalian University of Technology (DLUT), in 2008 and 2010, leading to over 9000 images. These images were then classified into three surface types (sea ice/snow, water and melt ponds). | | | | | | |





| JOIS MPF | Tanaka et al. (2016) | 2003-2014 | 68.9–88.2°N | 1453~2397 m² | n.a. | Contact the lead author. | n.a. |
|---|---|---|---|---|---|---|---|
| | Dataset collected from ship-based observations by Joint Ocean Ice Study during on the Canadian Coast Guard Ship Louis St-Laurent using a camera mounted with a view of the horizon and ice pack were classified into 5 classes (water, ice, water and ice, pond and ice, water, pond and ice). | | | | | | |
| PRIC-Lei | Lei et al. (2017) | Summer from2010 to 2016 | 71.7-88.4°N | 1 x 1 km | n.a. | Contact the lead author. | n.a. |
| | PRIC-Lei melt pond observations were collected during the Arctic Research Expeditions by the Polar Research Institute of China (PROC) during summer, where melt ponds along other variables, such as snow thickness, sea ice concentration were documented half-hourly from the bridge of the R/V Xuelong, covering along the cruise. | | | | | | |






**Appendix C: Melt pond fraction remote sensing literature review from summary of main research conclusions and datasets used**

| AOI and/or ice types | Period | Study description | Remote Sensing, Campaign (Ca), Validation (Va) and Comparison (Co) datasets | Reference |
|---|---|---|---|---|
| Pan-Arctic | 2002-2011 & 2017-present | Transferred MPD1 algorithm (see Appendix A) retrieval from MERIS to OLCI data to create a continuous 20-year MPF dataset (MPD2 MPF - see table 2); analyzed spatial and temporal trends | - MERIS<br>- Sentinel-3 OLCI,<br>- (Co): Sentinel-2 MSI | Istomina et al. (2025) |
| Pan-Arctic | 2013-2023 | Simultaneous retrieval of MPF and melt pond depth (MPD), enabling melt pond volume (MPV) estimation. RMSE < 10% (MPF), ~24.5 cm for (MPD). MPF showed positive correlation with downward surface radiation. | - Sentinel-2 MSI; Landsat-8<br>- (Va): SkySat (OSSP), MOSAiC helicopter (PASTA-ice), NSIDC/IceBridge (OSSP) | Xiong and Li (2025) |
| Pan-Arctic | - | MPF, ocean and sea ice fraction retrievals using MPD2 algorithm. Fully physical optimization approach using a priori empirical data (sea ice temperature history). Uncertainty reduced from 12.9% (MPD1) to 7.8% with MPD2. Bias of overestimation decreased as well. | - Sentinel-3 OLCI & SLSTR; ERA5 2-m air temp.; OSI-SAF drift data<br>- (Ca): In situ spectral radiometers from MOSAiC, TARA, S106-ARK31/1 and AlertMAPLI18 expeditions<br>- (Va): Sentinel-2 MSI | Niehaus et al. (2024) |
| MOSAiC Track | 2019-2020 | Mapped melt pond bathymetry independent of pond color/sky conditions. Mean depth deviation: 3.5 cm; > 1,600 ponds analysed.. | - (Ca): Helicopter (CANON EOS 1D) from MOSAiC and from PASCAL (RV Polarstern Cruise PS106/1) expeditions. PASCAL included pond depth measures | Fuchs et al. (2024) |
| CAA (Landfast Ice) | 2018, 2020 | Used a co-pol Ku band scatterometer to detect timing of pond onset. High VV/HH co-pol ratios were associated with MPF, when ponds large enough to cause VV/HH to exceed inherent noise (~0.16 dB) | - ScatSat-1: co-pol (VV/HH); ERA5: 2-m air temperature and wind speeds<br>- (Va): Sentinel-2 (NIR band) | Geldsetzer et al. (2023) |
| Pan-Arctic (including marginal seas) | 2000-2021 | Physically-based MPF retrieval via dynamic spectral unmixing with radiative transfer modeling. Found an increasing trend in MPF in marginal seas. For typical regions MPF and ice cover fraction show contrasting trends. | - MOD09AI (8-day),<br>- (Va): MEDEA; WV2-MPF; NSIDC; TransArc; IceWatch (see Appendix B) | Xiong and Ren (2023) |





| | | | | |
|---|---|---|---|---|
| CAA &North Greenland (MYI) | May 2020 | Monitoring melt onset though MPF from thresholding NDWI (Sentinel-2), and pond depth estimates from ICESat-2 via UMD-MPA and DDA algorithms (Farrell el al., 2020; Herzfeld et al., 2023). MPF peaked at 16 % in late June. | - Sentinel-2 MSI: 2-4 & 8 bands; ICESat-2<br>- (Co): WorldView-2: 2,3,5 & 7 bands | Buckley et al. (2023) |
| Pan-Arctic (MYI & FYI) | 2000-2020 | Increased the temporal span of retrieval of MPF using MODIS daily data. Generated open pan-Arctic MPF dataset NENU-MPF (see Table 2). Accuracies of R2 = 0.76, RMSE = 0.05. | - MODIS09GA: 1-7 bands<br>- (Co): UB-MPF ; UB-OLCI, UH-MPF, BNU-MPF (Table 2)<br>- (V/Ca): ASIMPSM, MEDEA, TransArc, IceWatch and HOTRAX (see Appendix B) | Peng et al. (2022) |
| MOSAiC Track | 2019-2021 | Retrieval of MPF using medium resolution data, (with LinearPolar algorithm – see Appendix A). MPF with a resolution of 10 m and uncertainty of 6%. | - Sentinel-2 MSI: 2 and 5 bands<br>- (Va): SkySat data classified with OSSP (from Wright and Polashenski (2018) ); (Ca): Helicopter-borne Canon EOS 1D Mark III (MOSAiC) | Niehaus et al. (2022) |
| CAA | 2006-2018 | Retrieval of MPF based on brightness temperature. Best channels to be used on the gradient radio: 10 & 18 GHz at near-shore environments. | - AMSR-E/2 brightness temperature data different channels at H-polarisation | Tanaka and Scharien (2022) |
| pan-Arctic (north of 60 N) | 2017 (Mid June-July) | Used MODIS daily product to retrieve and establish relationships with MPF. Improved RMSE by 3.7%, comparatively to other models. Allowed to show MPF's seasonal cycle. | - MOD09AI (8-day): 1 - 5 bands; Air temperature and sea level pressure<br>- (Co:) UH-MPF (Table 2); NSIDC (Appendix B) | Feng et al. (2021) |
| NW Passage locations in CAA | 2016, 2018, 2019 | Determined the best band combination for LinearPolar algorithm (see Appendix A) in Landsat. Comparison between LinearPolar and PCA | - Landsat (8 scenes): 2,3,4 and 4 bands<br>- (Va): Sentinel-2 MSI (6 scenes): 2 and 8 bands | Qin et al. 2021 |
| Central Arctic Ocean (FYI) and CAA (MYI) | Summer 2019 | Assessment of ICESat-2 on detecting ponds with different reflective properties. FYI smooth melt ponds most reflective, MYI larger ponds: most variable. | - ICESat-2: ATL03, ATL07,ATL10<br>- (Co/Ca): Sentinel-2: RGB bands and WV2 | Tilling et al. (2020) |
| CAA Landfast FYI | 2009-2018 (April) | Retrieved peak MPF with time series, and used them to predict summer sea ice. Correlations were found. | - RADARSAT-2: HH-pol, incidence angle between 20.0° and 49.3°<br>- (Ca): LiDAR<br>- (Co.): UH-MPF (Rösel et al., 2012) (see Table 2) | Howell et al. (2020) |



| Region | Period | Description | Data | Reference |
|---|---|---|---|---|
| Northern Canadian Arctic and Greenland (MYI & FYI) | 2016-2017 | Developed high-temporal-resolution (1–4 day) multi-sensor γ° SAR composites (Sentinel-1, RADARSAT-2) for melt onset detection. Compared melt onset dates to ASCAT and passive microwave algorithms. Results showed better performance on FYI. | - Sentinel-1, RADARSAT-2 (SAR)<br>- (Co): ASCAT | Howell et al. (2019) |
| Pan-Arctic (MYI & FYI) | 7 years from to 2019 | Melt pond classification (4 classes) and MPF retrieval for TOA 20-year record data - created MODIS-Lee MPF dataset (see Table 2). Normalised band differences produced the best results. Accuracy: 85.5%, RMSE: 0.18. | - MOD02HKM: 1-4 bands<br>- (Va): WorldView, NSIDC, MEDEA (Appendix A)<br>- (Ca): ARKTIS-XXII-2, PS86 data | Lee et al. (2020) |
| Pan-Arctic (MYI & FYI) | 2000-2019 | Created the longest (to date) MPF dataset - BNU-MPF (see Table 2).<br>The sea ice concentration (SIC) dataset played a minor effect on the MPF retrieval results. RMSE < 0.1 | - MOD09A1: 1-7 bands<br>- (Ca/V): HOTRAX; DLUT; TransArc; PRIC-Lei; NSIDC; NPI; IceWatch and JOIS,<br>- (Va) MEDEA; WV2-MPF( see Appendix B) | Ding et al. (2020) |
| Pan-Arctic (MYI & FYI) | 2000-2017 | MPF retrieved with an E-DNN. RMSE: 0.48-0.67; correlation coefficient: 6-12% depending on MPF observations. Evolution trends: increase. | - MOD09A1: 1,2,3,5 bands<br>- (Ca): HOTRAX, DLUT, TransArc, PRIC-Lei, NSIDC dataset (Appendix A)<br>- (Va): MEDEA (Appendix A) | Ding et al. (2019) |
| CAA (MYI & FYI) | 2017 (4 days) | LinearPolar algorithm (Appendix A) to retrieve MPF (accounting for variable reflectances): more accurate and precise than previous methods, with a 30% lower RMSE value. | - Sentinel-2: 2 and 8 bands<br>- (Co): IceBridge DMS imagery (see section 3.2.1) | Wang et al. (2020) |
| CAA (MYI & FYI) | 2017 (Summer) | Fully constrained (FCLS) to retrieve MPF. Achieved high accuracy: (RMSE~0.06). Evolution of melt ponds on FYI/MYI and relationship with albedo and temperature | - Sentinel-2, 2,4 and 8 bands; Landsat 8 L2<br>- ERA-Interim 2 m temperature reanalysis data<br>- (Va):WorldView-2 (WV2) (Appendix B) | Li et al. (2020) |
| Beaufort/Chukchi Seas, Central A. Ocean, (MYI & FYI) | 2016-2017 (July) | Classification of pixels into 4 classes (undeformed ice, deformed ice, open ocean & melt ponds) and 3 colors. Differences associated with FYI vs MYI. | - (Ca): Airborne Digital Mapping System (DMS) (IceBridge NASA) (see section 3.2.1)<br>- (Co): AMSR2 SIC | Buckley et al. (2020) |
| FYI & MYI | 2009, 2014, 2016, | Developed new algorithm OSSP (Open Source Sea Ice Processing) for 3 classes classification: snow/bare ice, melt ponds/submerged ice,/open water, using new algorithm. Accuracy over 96%. Originated WV2 MPF dataset (see Appendix B). | - WorldView (panchromatic) (75m); WorldView 8-bands (multispectral) (125m)<br>- (Ca) NASA IceBridge, Canon EOS 5D DMS (25m) (see section 3.2.1) | Wright and Polashenski (2018) |



| | | | | |
|---|---|---|---|---|
| Resolute Passage, Canada, Landfast FYI | 2012 | Correlations between the melt pond fractions and late-winter linear and polarimetric SAR parameters and texture measures derived from the SAR. Best RMSE of 0.09. | - Aerial photography<br>- RADARSAT-2: HH, VV, HV, VH, incidence angles from 23.1 to 42.6°<br>- (Ca): Airborne Canon G10 | Ramjan et al. (2018) |
| 10 sites in Arctic Ocean FYI | 2000, 2006, 2012, 2018 | First MPF retrieval from hybrid-polarised compact polarisation (CP) SAR. Systematic overestimations due to the ignored effect of snow. Limitation to inc. angles: >35 degrees. | - RADARSAT-2: quad-pol data (VV/HH) | Li et al. (2017) |
| North of Svalbard, Drifting FYI | 2012 | MPF retrieval from 4 dual-pol X-band. Best results: co-pol ratio at medium wind speeds, VV-pol at low wind speeds. Best inc. angle: 29°, >40° not reliable. | -TerraSAR-X: HH-VV<br>- (Ca): Helicopter-borne Canon 5D Mark II and GPS/INS (ICE2012)<br>- Weather station for wind retrievals | Fors et al. (2017) |
| Pan-Arctic (FYI) | n.a. | MPF estimation at basin scale. Reliable results in MPF zones; overestimation in low MPF zones. | - MOD09<br>- (Va): QuickBird data | Yackel et al. (2017) |
| NW Passage MYI and FYI | 2016 (Winter and spring) | Estimating MPF using only HH-pol $\gamma°$. RMSE: 0.09. Strong correlation between winter backscatter coefficient and MPF: spring MPF. | - Sentinel-1 EW Mode: HH $\gamma°$<br>- GeoEye-1 (GE): 4 bands | Scharien et al. (2017) |
| CAA & Beaufort Sea | 2005-2014 (Jul-Oct) | Estimation of MPF from AMSR-E in comparison with ship-born observations. 89 GHz provided more details in areas of high sea ice concentration. | - AMSR-E: 6.9 GHz, H-pol and 89.0 GHz, V-pol<br>- (Va): MODIS MPF (Table 2)-<br>(Ca): HOTRAX (Appendix B) | Tanaka et al. (2016) |
| Chukchi Sea, (MYI) | 2011 3 days | Classification of melt ponds, water and sea ice. Difficult discrimination of melt ponds vs open (ocean) water. HH-pol contributed the most to the Random Forest. RMSE: of 2.4% | - TerraSAR-X: HH and HV-pol with 32.7° incidence angle<br>- Airborne X-band (9.3 m resolution)<br>- Aerial photographs (Kim et al., 2013) | Han et al. (2016) |
| Central Arctic (FYI & MYI) | 2010 (Summer) | Classification of surface into 3 classes: snow-covered/bare ice, melt ponds and open leads. Collected statistics on melt pond characteristics. | - Helicopter-borne photograph with Canon G9 (CHINARE2010) | Huang et al. (2016) |
| Pan-Arctic (MYI & FYI) | 2009 (June - August) | Deriving MPF from MODIS, to understand how sea ice concentration retrievals are impacted by the presence of MPF | - MOD09GA: 1,3,4 bands<br>- (Co): AMSR-E/Aqua: 16 Ghz | Kern et al. (2016) |
| Central Arctic Ocean; FYI and MYI | 1998 (SHEBA) 2005 (HOTRAX) | Developed image-based graph algorithm to analyze melt pond connectivity and fluid flow using conductance networks. Supports modeling of albedo feedback. | - (Ca): SHEBA, HOTRAX (See Table 1); aerial photographic imagery | Barjatia et al. (2016 |



| | | | | |
|---|---|---|---|---|
| Beaufort/ Chukchi Sea region FYI and MYI | 2011 | Analysed seasonal evolution of melt ponds on Arctic sea ice using, for an entire melt season on drifting first-year and multiyear sea ice. Classification into 4 classes sea ice, thin ice, melt pond and open ocean water. | - Panchromatic satellite data (1 m) - (Ca): Airborne and in situ data (NASA DISTANCE) | Webster et al. (2015) |
| Pan-Arctic | 2002-2012 | Development of a new algorithm to retrieve MPF (without fixed values of spectral reflectances & accounting with bi-directional reflectance & atmospheric corrections). Errors for dark ponds: can exceed 50%. Generated UH-MPF dataset (Table 2). | - MERIS Level 1B: 1-15 bands - (Ca):MELTEX | Zege et al. (2015) |
| Pan-Arctic (MYI & FYI) | 2008, 2006 | Algorithm to retrieve MPF from MERIS data. Unscreened cloud overestimates MPF before melt onset and underestimates MPF during the melt season. FYI floes results are worse due to ice drift. Ambiguities on retrieved MPFs, suggest with addition of temperature could improve results. | - MERIS Level 1B: 1-4, 8, 10, 12-14 bands -(Ca/V): Barrow 2009, MELTEX 2008, NOGRAM-2 2011, NOGRAM-2 2011, C-ICE 2002, HOTRAX 2005, TransArc 2011, POL-ICE 2006 (Appendix B) | Istomina et al. (2015) |
| Campaign CHINARE 2010 | 2010 (Summer) | Classification of high resolution data in 4 classes: (water, submerged ice, melt ponds and submerged ice along ice edges), shadow, and ice/snow. Overall classification accuracy of 95.5%, with a producer's accuracy of 90.8% and a user's accuracy of 91.8% | - Aerial photographs with Canon G9 (helicopter-borne,CHINARE2010) - (V/Ca): Ship-based observations | Miao et al. (2015) |
| North of Svalbard, Nansen Basin, FYI | 2012 July-August | 10000 images (with homogeneous MPF) classified into 4 classes: dark ponds, bright ponds, open water and bare ice. Pond fractions decrease matched open water increase in the marginal ice zone. | - Helicopter-borne photograph with Canon EOS 5D (ICE12) - In situ broadband albedo measurements | Divine et al. (2015) |
| CAA (Landfast FYI) | 2012 May-June | Developed VV/HH-based model to retrieve MPF during 3 ponding stages. HV/HH offer more potential over VV/HH ratio, RMSE 0.05-0.07 comparable to optical approaches | - RADARSAT-2: HH, VV, HV, VH | Scharien et al. (2014) |
| North Fram Strait, Greenland, Svalbard | 2009 June- August | ENVISAT SAR-based MPF retrieval. SAR σ° mosaics were visually compared with MODIS MPF, along with spatio-temporal coinciding data. Hard to depict correlation except for smooth landfast FYI. | - ENVISAT WSM: HH-pol - (Co): UH-MPF dataset (Table 2), MOD09GA RBG:3-6-7 and 2-1-3 bands | Mäkynen et al. (2014) |
| Campaigns (North Svalbard and Fram Strait) | 2010 | Semi-automatic classification of melt ponds, open (ocean) water, thin ice, bare ice, and submerged ice (5 classes) in combination with sea ice thickness measurements. Provided insight on relation between MPF and sea ice thickness. | - Helicopter-borne photograph with Canon EOS 350D | Renner et al. (2013) |



| | | | | |
|---|---|---|---|---|
| Campaign Chukchi Sea | 2011 (Summer) | Mapping melt ponds using very high airborne and space-born high-resolution X-band SAR. Results were comparable with aerial photographs from previous studies. | -(Ca): Helicopter-born X-band NanoSAR: HH-pol (KOPRI-led R/V Araon) and airborne SAR (SHEBA),<br>- (Va): aerial photograph; (Co): TerraSAR-X Stripmap mode (6 m and dual-pol HH and VV) | Kim et al. (2013) |
| Pan-Arctic (MYI & FYI) | 2008 | Estimation of MPF, and ice and water coverage and sea ice concentration for the entire Arctic region, improving Tschudi et al. (2008). Showed good agreement with observations. | - MOD09AI (8-day): 1,3,4 bands<br>- (Va): MOD09AG daily: 2,3,4 bands<br>-(V/Ca): HOTRAX, NSIDC, MELTEX (Appendix B) | Rösel et al. (2012) |
| Northern Beaufort Sea | 2001 (July) | Comparison of MPF retrievals from Landsat and MODIS. Classification in 4 classes (open (ocean) water, snow-covered ice, and two types of melt ponds). Results showed problems with saturated pixels, which is related to sun elevation and the surface type. | - Landsat 7 ETM+: 8 bands<br>- MOD09: 1,3,4 bands | Rösel and Kaleschke (2011) |
| 3 Locations Arctic Ocean | 2017 (August) | Analysis of the potential of radar backscatter data for melt pond identification using different frequencies. MPF estimates become more realistic if X- and Ku-band are used in combination with C-band. | - Helicopter-borne Multi³Scat radar: S-, C-, X-, Ku-band at HH, HV, VV and VH, from incidence angles between 20 and 60°<br>-(Co/Ca): Video data (ARKCCII/2) | Kern et al. (2010) |
| Beaufort/ Chukchi Sea region | 2004 | Estimation of MPF evolution at the basin scale. Areal extension of melt ponds has increased over the study period. | - MOD09: 1-3 bands<br>- (Va): UAV digital camera images | Tschudi et al. (2008) |
| 4 Arctic Ocean sites | 1999-2001 | Surface maps of 2 (water and ice) or 3 (ponds, open water and ice) and MPF statistics of 500 m cell with 1 m resolution. Attempts to relate MPF and SIC from microwave data proved unsuccessful. | - High resolution optical satellite imagery (n.d.) | Fetterer et al. (2008) |
| Campaign Wellington Channel FYI | 1997 | Understanding the capacity of the RADARSAT-1 time series for melt pond coverage. Found correlations between the scattering coefficients and the MPF and retrieved MPF geophysical parameters. | - RADARSAT-1 (C-band): HH, several incidence angles<br>- (Co/Ca): In situ Temperature and wind velocity and aircraft video data | Yackel and Barber (2000) |
| Campaign Wellington Channel FYI | Melt season (Julian Days 181-184) | Semi-automated spectral classification of FYI melt season surface types from digital aerial videography; classified four distinct surface cover types: snow, saturated snow, light and dark-colored melt ponds; analyzed fractional coverage, integrated surface albedo, and melt pond morphology patterns | - Digital aerial videography<br>-(Va): In situ surface albedo measurements<br>-(Co): RADARSAT-1 SAR observations | Yackel et al. (2000) |



## 8. Author contribution


SA: article conceptualization, resources (literature sources), writing (original draft preparation, review and editing). SD: resources (literature sources) and writing (original contributions, review and editing). DF: resources (literature sources) and writing (original contributions, review and editing).

## 9. Competing interests

The authors declare that they have no conflict of interest.

## 10. Acknowledgements

This work was supported by the MIT Portugal Program (under Doctoral Grant Doctoral Grant PRT/BD/154506/2022) awarded to Sara Aparício). Within the Department of Applied Mathematics and Theoretical Physics, University of Cambridge, SD is associated with the Institute of Computing for Climate Change (ICCS) and is supported by Schmidt

Sciences, LLC. We are also grateful to the Portuguese Foundation for Science and Technology (Fundação para a Ciência e a Tecnologia - FCT) for supporting CENSE Unit UID/04085: Center for Environmental and Sustainability Research; https://doi.org/10.54499/UIDB/04085/2020; https://doi.org/10.54499/UIDP/04085/2020) - NOVA University Lisbon & CHANGE - Global Change and Sustainability Institute, LA/P/0121/2020; DOI: https://doi.org/10.54499/LA/P/0121/2020.

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
