# Peer review of "Observational data of Arctic Sea Ice Melt Ponds: a Systematic Review of Acquisition and Processing Approaches"

_EGUsphere, 2025_

## Referee Comment (RC2)

**Review of: egusphere-2025-4480**

Observational data of Arctic Sea Ice Melt Ponds: a Systematic Review of Acquisition and Processing Approaches

By Aparício et al.

**Summary**

The authors provide a review of earth observation (EO) methods used in the study of sea ice melt pond properties. Systematic melt pond property observations using EO are challenging and there are no high quality observations (e.g., ECV or CDR type datasets) despite importance for observing sea ice evolution, making proxy estimates such as albedo or light transmittance, and for predicting sea ice conditions. The paper requires some improvements, as per the major and minor comments provided here, and by others, before consideration for publication in *The Cryosphere*.

**Major Comments**

- 1. Given that the paper is a synthesis, the abstract should contain some statements relating to the outcomes of the synthesis. As it is, there only a statement about EO data gaps, along with an outline of the paper's intentions and some motivation statements.
- 2. It would be advantageous for terminology relating to EO to be tightened up. The authors are focused on what they refer to as the "main" EO methods, which appear to be airborne and satellite-based remote sensing. But in section 3, EO is also used to describe some in situ methods to, which is fine since buoys, weather stations, etc. are EO tools too. The term EO is dropped for remote sensing in the main text is that distinction intended?
- 3. The paragraphs on Lines 59-74 needs improvement since some passages are hard the follow. More specific information is needed. How are melt ponds related to a shift in this Arctic sea ice spring predictability barrier? What *is* the barrier exactly? It can be stated better than "melt ponds ...predict ....September minimum" since melt ponds can't make predictions. More detail should be given on how melt ponds are parameterized in GCMs, especially given the sensitivity, and the overall theme of the paper. What is the link between GCM parameterizations and the Spring predictability barrier, as suggested on Lines 66-67? Overall, seasonal prediction and climate model projections, as they pertain to melt pond properties should be much better described. Otherwise, the motivation statements on Line 87 and Line 100 are not well backed-up.
- 4. The seasonal evolution of melt pond properties in Section 2, including Figure 3, needs to better differentiate between first-year ice and multiyear ice. There is mention of the influence of ice types in the context of topography, which is good, but there still needs to be seasonal stages in terms of melt pond formation and evolution, and expected melt pond albedo or fraction, defined by ice type. E.g., Stage 4 (freeze-up) rarely occurs for first-year sea ice since the ice melts away / disintegrates during summer.
- 5. It is generally hard to follow what satellite missions are current or past. Information about data products availability comes later in Table 2, but prior to that there are many missions discussed without enough detail regarding status.
- 6. Optical (laser) altimetry is addressed but radar altimetry is not. Radar altimetry should be included.

7. In Section 3.2, more emphasis needs to be placed on the salient findings from these campaigns, in the context of melt pond property information retrieval from EO data, where possible. In some cases there are just descriptions of data collected and some observations made (e.g., ICE212). In other cases the descriptions aren't clear (e.g., MOSAiC contributions to spatiotemporal studies – what was learned?).

**Minor Comments**

L20: "... sensor type."

L48-50: Recommended text: "(Right) True colour composite image of Arctic sea ice melt ponds from the Copernicus Sentinel-2 satellite (illustrating the large variability of melt pond, lead and open ocean reflectance). Acquired on 17th June 2024 off the Northeastern coast of Greenland.".

L75: Add mention that SHEBA and MOSAiC are summarized later on.

L78: Change "struggle to" to "do not".

L93: "...melt pond studies..."

L96: Refer to "... passive microwave" since radar used microwaves too.

L103: Add a description of the basic outline of the sections of the paper.

L114-118: Bare ice is referred to as both stable and changing (as melt progresses). Clarify.

L121: "radiometric" doesn't fit here.

L127: "ice surface features" should be changes to "ice properties" since ice surface and volume properties are important (as is described).

L134-135: Are the timings (by month) always appropriate when Arctic sea ice extends across about 40° latitude?

L136: Here and in Figure 3 the melt onset should be "pond onset" since melt onset precedes the formation of ponds through meltwater accumulation. See, e.g.: <a href="https://doi.org/10.1038/s41597-023-02760-5">https://doi.org/10.1038/s41597-023-02760-5</a>

L142: See major comment 4. The seasonal peak in pond fraction for first-year ice can be at the initial flooding stage, before drainage pathways open up. The Polashenski et al., 2012 paper shows this too.

L159: Delete Earth Observation since "(EO)" was defined earlier.

L160: Be consistent with "in situ" being italicized or not, here and elsewhere.

L162: "....considerations of associated ..."

L164: "addresses" (change the tense)

L168: In the Section 3.1 heading "ponds" should be "pond"

L174: Describe how the main applications were determined. In figure 4, "measured parameters" is used; are these the same thing?

L179: The Figure 4 caption is incomplete. In the figure, it is unclear if there is a difference between "detection of onset" and "timing of onset". It is odd that melt pond fraction is not mentioned.

- L183: Name the spaceborne mission.
- L186: It should be "EM spectrum".
- L194: "instruments"
- L183-205: Some of these missions are still operational, and others are defunct. Some clarity in this regard is needed. With regards to high-resolution imagery, is WV referenced as an example only? There have been others used; e.g. Webster et al. 2015 used data from the NTM satellites.
- L222: State why Landsat is not suitable, and if that is for a specific Landsat mission or all of them.
- L223: Sentence "Additionally, for high resolution..." should be re-written for clarity. Are there no data for coastal waters, or limited data?
- L226: Remove "Similarly". The commercial restriction associated with those platforms is not similar to Sentinel-2.
- L230-234: This should be made clearer. It is unclear how fresh snow mimicking melting ice leads to misclassification. What is classified/mis-classified? If it is melt pond fraction or pond/ice, aren't fresh snow or melting ice basically the same class (i.e., *not* melt pond)? It is also unclear why freeze ponds (refrozen ponds or freezing ponds?) lead to melt pond fraction overestimation.
- L240: Sentence on LiDAR should be "LiDAR laser pulses are composed of photons typically emitted at one or both of these wavelengths.".
- L245-248: Mention is made of pond depth and presence, in terms of research. But only pond depth is mentioned in relation to operational products. What about presence?
- L249: "...consist of laser penetration..."
- L249-252: Add detail on how melt pond presence affects existing ICESat-2 data products for sea ice, and what those products are, if known. Are melt pond areas masked out, or generally unreliable in sea ice data products during spring/summer conditions?
- L254: Add a comma after "particles".
- L262-264: The statement about liquid water in snow needs correcting. Snow effects do not create the low signature of melt ponds, since melt ponds are water bodies and their backscatter is dependent on wind/wave roughness.
- L272: Clarify what you mean by early melt detections, in the context of X-band suitability over other frequencies.
- L274: Correct "melt onset" if necessary. I.e., if this is pond onset.
- L275: Change "on" to "for" and make "signatures" to "signature".
- L297: Get rid of the empty space. Also it should be "... poorly distinguishable from smooth sea ice.".
- L301: Provide detail about the swath widths. Smaller than what?
- L304-306: The statement about X-band is hard to follow. Optimal performance for deriving what melt pond related information? MYI monitoring of what melt pond related information?

L306-307: Backscatter should not increase due to specular reflection.

L311-313: It is confusing that X-band systems are limited at winds above 5 m/s but optimal retrieval is found at about 6.3 m/s.

L316: "pond onset"

L311-322: It is hard to follow what melt pond properties are being observed/retrieved here. E.g., optimal retrieval of what melt pond information?

L326: The section on scatterometers should be shortened since most of the current focus is on melt onset, which precedes pond onset. Focus should be placed on melt pond properties.

L350: As for scatterometers, the discussion of melt effects in this section should be shortened and more emphasis placed specifically on melt pond related properties.

L376: Low penetration depth also applies to scatterometers and SARs since they operate in the microwave range. It should be noted above too.

L390: Provide some detail on what criteria were used to include these campaigns, and (potentially) exclude others that have studied melt pond properties and detection techniques.

L488: "named"

L506: "that geophysical inversions" doesn't make sense. Please clarify.

L514: In Figure 8, there is mention of compact polarization but there was no mention of that in the earlier section on SAR approaches.

L515-517: Some of the mentioned studies focus on melt pond fraction prediction, where spring MPF is predicted from winter SAR imagery, e.g., Howell et al., 2020. Others focus on direct estimation of MPF, Tanaka and Scharien, 2022. This should be clarified here, and in the earlier SAR section.

L522: It would be helpful to know if more advanced techniques like ANN represent any increase in efficacy compared to traditional techniques.

L526: "converged"

L530: MPF was defined earlier.

L532: "melt pond" (singular).

L580: Statistically significant "difference".

L581: Use "MPFs".

L598: "datasets"

L631: Use "MPF".

L698: "5. Discussion"

L757: It is unclear why SAR-based methods are specifically identified as lagging in situ methods when, presumably, other satellite-based methods would be similarly lagging. This also comes up on L867.

L765: Use "MPF"

L780: CAA was defined earlier. Use "CAA".

L821-822: Elaborate what these schemes are, as well as schemes in general as they pertain to melt pond representation in models. This will make this section of the review more accessible to readers focused on in situ and satellite studies.

L850: Elaborate on why specifically pond depths, pond fraction, snow cover and type. Is ice topography and permeability less important?

---

## Author Comment (AC1)

**Response to the comments of Reviewer 1**

We thank Reviewer 1 for their relevant feedback. Their detailed comments have led to substantial improvements in the manuscript's organizational logic, focus and clarity. Following their advice, we added significant details that drastically improved the quality, readability and utility of the paper, and these suggestions have strengthened and enhanced this review article.

Our responses to the Reviewer's comments are provided below in **blue text** and are positioned directly beneath each corresponding comment.

This manuscript aims to provide a comprehensive review of observational approaches for quantifying melt ponds on Arctic sea ice. Given the central role of melt ponds in ice-albedo feedback and the difficulty of obtaining consistent observations, a systematic overview comparing the strengths, weaknesses, and future potential of various methods is highly relevant and well-motivated.

However, the current version of the manuscript lacks the focus and structure expected for a review paper. The organization makes it difficult to follow the line of reasoning between sections, and the conclusions remain unclear. The manuscript should better clarify the current state of sensing capabilities and offer practical guidance on which products or methods are best suited for different applications.

**Major Comments**
**Organization and Focus**
- The manuscript is often repetitive, particularly in sections discussing general motivation for melt pond observations. In contrast, the technical comparison of methods lacks sufficient depth.

We thank you for your comment and have made substantial revisions to address potential redundancy and structural issues throughout the manuscript. The Introduction (Section 1) now explicitly defines scope and terminology, consolidates the primary motivation, and clearly outlines the structure of the manuscript. Section 2, which presents the main characteristics of melt ponds and their evolution, is now supported and cross referenced to Appendix C, which has been expanded to include a comprehensive analysis of over 40 studies. Section 3 has undergone many changes. For example, it was restructured with clearer headings that distinguish: (3.1) '*Spaceborne observation*' for satellite remote sensing, (3.2) '*In situ and field campaign observation*' for ground-based and ship-based measurements, and (3.3) '*Post-processing techniques*'.

These sections now focus exclusively on technical sensor capabilities, and include new and improved images following advice of the Reviewer. Regarding technical comparison of methods which was addressed by the Reviewer, Section 3.1, provides enhanced technical specifications for each sensor (physical principles; measured parameters; missions status, data availability; MP signatures and detection mechanisms); Section 3.2, was revised to ensure consistent structure and level of detail for each campaign, while including specifically how each of these campaigns advanced our understanding of melt ponds. Section 3.3, examines methodological approaches; algorithm types, and their applicability to different sensors. We maintained the technical comparison details primarily in updated appendixes to enhance article readability while providing comprehensive depth for interested readers.

Furthermore, if the Reviewer feels specific sections of the appendix should be moved to the main body or any topics therein, we would be happy to hear them.

Section 4 underwent a considerable change as it now focuses exclusively on existing datasets. Section 4.2, following the Reviewer's concern on being beyond scope, was placed entirely as an Appendix, where it provides complementary information that cross references and bridges Section 2, 3 and 4.

Finally, Section 5 was revised to exclude any redundancy, with motivations for melt pond studies now stated exclusively in Section 1. Section 5 now focuses solely on identifying key knowledge gaps and outlining directions for future research. Section 6 remains as the Conclusions section which has undergone its own updates.

- Section 3 should begin with a concise synthesis of overarching challenges common across observational platforms, including:
    1. Temporal coverage (limitations due to clouds, daylight, and repeat time),
    2. Spatial resolution, and
    3. Discrimination and identification of pond boundaries.

      These challenges are especially critical for melt pond fraction (MPF) retrievals.

We agree with the Reviewer that these are important topics. Following the Reviewer's advice, we have added a synthesis that previews these major challenge categories and forward-references where each is addressed in detail, guiding the reader more effectively through the structure progression.

All challenges pointed by the Reviewer are addressed extensively for each sensor type throughout Section 3. Namely, temporal coverage limitations are discussed at lines 206-214* and 249-252, for passive and active optical sensors, respectively and at lines 294-295 for microwave systems. Spatial resolution constraints (and resolution coverage tradeoff) are covered at lines 194-229, for optical sensors and at lines 267-294 for active and passive radar systems, respectively. Finally, pond boundary discrimination challenges are addressed at lines 230-235, for optical sensors and at and 294-322 and 371-386 for active and passive microwave systems. We have also updated this Section now to include another family of sensors, namely altimetry, and it follows the same structure as others, contributing to the completeness of Section 3.

*Please note that the line numbers provided here correspond to the original manuscript. Since substantial revisions were made, the line numbers will differ in the updated version. We hope this clarifies the changes.

**4. Scope and Content**

- It is unclear what the extent of observational datasets that the manuscript aims to include are. The section describing field campaigns should clarify which are included and why. Focusing on those most relevant for benchmarking or validation would

strengthen the section. A map of campaign locations could help illustrate spatial biases.

We thank the Reviewer for their useful feedback, particularly regarding the need to clarify the scope/extension of the observational datasets considered and the suggestion to focus on those most relevant ones (for benchmarking and validation). We have made the following substantial changes:

- We added to the Introduction (Section 1), a clarification stating that the article includes observational datasets derived from spaceborne platforms, airborne campaigns and major field campaigns that have contributed to melt pond research; it also states that we focus on datasets that provide (i) melt pond specific measurements; (ii) that have been widely used in literature for algorithm development, validation or (iii) constitute standalone research products, generated through dedicated methodological or algorithmic developments (e.g., pan-Arctic melt pond fraction estimates). We hope by including those most relevant for benchmarking and validation it strengthens the section accordingly.
- At beginning of Section 4, where observational datasets are described, we synthetize its structure cross referencing the two families of datasets: pan-Arctic, multi-year satellite products (Section 4.1.1, Table 2), and high-resolution regional datasets (Section 4.1.2, Appendix B).
- Section 4.2 was moved to support Appendix C, so that Section 4 is exclusively dedicated to datasets and products.
- With respect to the field campaigns, we have revised the introduction to the campaign section to clarify both the intent of the section and the rationale behind the selection of the listed observational campaigns. The revised section now explains that the campaigns were chosen as key reference field studies that have provided significant in situ melt pond datasets, which are discussed later in the manuscript in the context of their application for validation activities. In particular, the selection criteria are now explicitly stated as (i) substantial contribution to the understanding of melt pond processes, (ii) accessibility and scientific use of the datasets for model and satellite product validation, and (iii) their representation of different observational platforms, regions, and temporal scales. This section itself was also revised, and it now achieves better balance regarding how descriptive each campaign is, while including specifically how each of these campaigns advanced our understanding of melt ponds.
- Finally, following the suggestion of the Reviewer, a map of the campaigns was created and added. We appreciate the originality of showing in a visual manner the spatial biases, and thank the reviewer for this suggestion.

[Figure]

Figure: Arctic sea ice melt pond-relevant campaigns and expeditions

- Although the focus is on Arctic sea ice, some mention of Antarctic melt ponds is warranted. A short discussion of their sparse observations, unique challenges, and potential for future monitoring would improve completeness.

We appreciate the Reviewer's suggestion to include a mention of Antarctic melt ponds to improve completeness of the manuscript. We included some mention of Antarctic melt ponds addressing their sparse observation, unique challenges and trends accordingly, while noting the larger role that melt ponds play in the Arctic.

- The section on parameterization in global climate models seems beyond the scope of this review. It could instead be reframed as part of the motivation or discussion of future directions.

We thank the reviewer for this suggestion and have reframed it as part of the motivation section accordingly.

- **Key Variables and Definitions**
- Introduce the key melt pond variables early in the manuscript, ideally in a table. These should include MPF, depth or volume, connectivity (and open vs. lidded), and melt onset.

We introduced  the key melt pond variables accordingly (around L157). It now explicitly lists and describes the key observable variables, furthermore forward-referencing where their measurement or retrieval approach is discussed in detail.

- Methods for measuring parameters beyond MPF (e.g., pond depth or volume) need greater attention (see Buckley et al., 2023; Fuchs et al., 2024).

We thank the Reviewer for raising this important point about methods for measuring pound depth and volume, giving greater emphasis to these, going beyond MPF. We have added clarification and extra detail to our text that now highlights other parameters measurements - including  substantially about  pond depths. We have also included volume, for example to our bathymetry discussion.
Moreover, we   included the Reviewer's suggested literature, and incorporated additional relevant studies (e.g., Xiong and Li, 2025). Section 5.1 has been revised accordingly and now explicitly addresses these parameters, emphasizing their critical importance and the fact that they remain under-measured.

- Figures and Tables
- The choice of figures should be revisited. A meta-analysis figure summarizing validation efforts or comparisons of MPF products would be valuable. Notably, a figure showing typical seasonal cycles should be included.

We thank the Reviewer for this comment. We added a synthesis of available validation and intercomparison information making clear links to the MPF datasets (with cross references 4.1.1 and to Table 2). We have moreover revised Figure 3 showing the seasonal development of melt pond formation, by adding MPF-related aspects (such as increase of %). These illustrate the typical seasonal cycles of MPF and its differences between MYI and FYI.

- Consider adding:
    - A consolidated table summarizing pros and cons of each remote sensing method for MPF (the current figure does not allow easy comparison).

        We have updated Figure 4 to reflect this under *main applications* for optical systems.

    - Example images comparing different types of retrieved melt pond products and A space–time diagram quantifying temporal and spatial coverage, or the percentage of usable data for each platform.

        We thank the Reviewer for suggesting we could consider adding such images. As we wish to provide a methodological synthesis, we feel this is potentially beyond the scope and aim of our paper. We therefore focus on summarizing reports capabilities and limitations from the literature and improve substantially on these aspects such as through all our updates as provided here.

**Specific Line-by-Line Comments**

- L52: Pond color has been shown not to strongly depend on pond depth. Revise or list last.

Corrected - now listed last.

- L66–67: The statement that melt ponds can break the spring predictability barrier lacks clear supporting evidence.

We thank the Reviewer for pointing this out. We did not intend to suggest that melt ponds definitively break the spring predictability barrier, but more that some studies have hinted they could help in improving such seasonal predictions where forecasting has been imperfect and faced challenges. We have amended the text accordingly.

- L68, L70: Reconsider the choice of references (e.g., Driscoll et al., 2024; Polashenski et al., 2012).

These have been replaced and more suitable references have been chosen.

- Figure 2: If retained, the text should better explain why this observation is relevant.

We have amended our text for further clarification. We thank the Reviewer for highlighting this as it improves our text.

- L123–124: Repetitive—streamline.

Streamlined.

- L137–142: Provide references for albedo values and ensure consistency across lines.

References provided (Perovich et al., 2002; Grenfell and Perovich, 2004; Polashenski et al., 2012), and we have addressed consistency issues when addressing albedo ranges across different surface types and melt pond stages, throughout the manuscript.

- Figure 3: Consider annotating observable variables and how features such as refreezing or lidding affect retrieval.

Figure 3 was updated in order to enhance its utility following Reviewer's recommendation. The modifications include an enhanced caption explicitly identifying observable variables from remote sensing (e.g.pond area/coverage; surface color; ice lids); they also include subtle visual indicators to panels b) and c) highlighting observable figures that affect retrieval accuracy.

- L155–157: Add references.

References added.

- L172: Typo—merge sentences.

Sentences are now merged.

- Figure 4: Well prepared, but note limitations (e.g., LiDAR also requires cloud-free conditions). Clarify whether each method detects or times onset.

We thank the Reviewer for these important suggestions. Figure 4 has been revised to clarify onset-related applications. Regarding limitations cloud and other environmental conditions; melt stage dependencies; and ice type impacts on signatures, these are not included in the figure itself but are comprehensively discussed in Section 3.1 for each sensor type. The updated figure incorporates several additional improvements: (i) inclusion of radar altimetry as a distinct sensor category; (ii) clearer visual distinction between measured parameters (directly observable quantities) and main applications (derived products); and (iii) an expanded caption that describes the figure's structure and content.

[Figure]

*Figure 4: Overview of Earth observation methods for melt pond detection across the electromagnetic spectrum. The figure shows six sensor types (multispectral sensors, LiDARs, radiometers, scatterometers, radar altimeters, and synthetic aperture radars) organized by wavelength range (optical and microwave). For each sensor type, the figure presents: (top row) example satellite missions and operating wavelengths; (middle row) measured parameters and physical principles; (second last row) characteristic melt pond signatures and (bottom row) main applications.*

- L255: Replace "capabilities" with "utility."

Replaced.

- L277: Clarify whether "liquid melt pond fraction" differs from MPF used elsewhere; ensure consistent terminology.

Fixed to melt pond fraction.

- L320: Clarify whether sensor sensitivity differs for FYI vs. MYI, and introduce this distinction earlier.

We thank the Reviewer for this. Accordingly, this distinction has been introduced earlier in Section 3.1.2 when discussing SAR signatures. The text at L320 now explicitly references these ice-type-specific sensitivities. We have clarified that sensor sensitivity indeed differs between FYI and MYI due to their distinct surface roughness, dielectric properties, and melt pond morphologies.

- L366: "Firn" is not relevant for sea ice melt ponds—remove.

Removed.

- L387: Clarify intent and scope of listed observational campaigns.

We have clarified the intent and scope of the listed observational campaigns by revising the text preceding Table 1. The updated text now explicitly states the selection criteria we applied. Specifically, we have emphasized that these are key field campaigns that have significantly contributed in situ melt pond datasets to the scientific community. These campaigns were selected based on their substantial contribution to melt pond observations, data accessibility, and their representation of various observational platforms and temporal scales.

- Figure 6: Consider removing or supplement with date, location, and retrieved pond parameters.

We thank the Reviewer for this suggestion. As a result, we have supplemented Figure 6 with the relevant date and location information. The retrieved parameters are also introduced in the text preceding the figure, following the same convention used for the other images in this section.

L425–434: Reorganize to emphasize what was observed, derived products, and how data were collected.

The entire SHEBA description has been reorganized as suggested. It now emphasizes: (i) the observations made, (ii) the derived products, and (iii) their main applications, while directing the reader to the table where the related products are listed. The section concludes with a description of how the data were collected.

Additionally, this section has been revised to provide more balanced descriptions of each campaign, highlighting specifically how each one advanced our understanding of melt ponds by incorporating key findings from each campaign.

L468, L474: Minor wording edits ("melt pond"; "specifically").

Edited

- L480–484: Redundant—streamline.

Streamlined

- L484: Consider starting a new section; summarize datasets and melt pond variables.

We agree that summarizing datasets and melt pond variables is important for guiding the reader. Following the Reviewer's suggestion, we added clear forward-references after L484 directing readers to Section 4.1, where all available melt pond datasets are systematically summarized: Section 4.1.1 covers pan-Arctic satellite-based MPF datasets (Table 2), and Section 4.1.2 addresses high-resolution regional products (Appendix B).

Section 3.3: Difficult to connect with sensor overview. Suggest merging with Section 4 and organizing by wavelength/sensor type.

We thank the Reviewer for this helpful structural suggestion and agree that strengthening the connections between sensor types will enhance readability. In response, we have reinforced the links between Sections 3.1 and 3.3 to more clearly connect each sensor type with the corresponding processing techniques. These relationships are now also clarified in Figure 8 and its caption. Additionally, we have included a transition between Section 3.3 and Section 4, which now better explains its role as a bridge between the processing techniques (based on sensor characteristics) and the resulting datasets discussed in Section 4. We believe these revisions establish a more explicit connection between sections, based on sensor type, and improve the overall flow of the text.

- L526: Replace "convergent" with "converged."

Corrected.

- L538: Table 4 missing—add or renumber.

Corrected.

- L547–549, L553: Typographical and introduction issues—revise.

Revised.

- Table 2: Clarify whether "MERIS-ZEGE" is identical to MPD1; include sensor type for consistency.

We thank the Reviewer for this. We clarify that *MERIS-ZEGE* is a dataset, whereas *MPD1* refers to an algorithm; they are not the same. Naming convention for datasets was clarified at Table 2, and MPD1 algorithm is now introduced and clarified in Section 4.1.1.

Section 4.2: Scope too broad—consider omitting or condensing to future directions.

We thank the Reviewer for this. Therefore, Section 4.2 is now moved to be part of Appendix C where it can provide some linking information between sections without being in the main part of the narrative.

L666–697: This list adds little—remove or condense.

Lines 666–697 were part of Section 4.2, which has now been moved to Appendix C in response to the previous comment. As a result, this list no longer appears in the main manuscript.

L671: Incomplete sentence—revise.

Revised.

- L714: Clarify meaning of "sensor-based constraints."

We changed this to 'sensor-specific limitations'.

- L734: Highlight the importance of seasonal transitions more clearly.

We thank the reviewer for this. Therefore we revised the text to do this as follows:

*"Capturing rapid seasonal transitions, particularly pond onset, drainage events, and freeze-up, is critical for understanding melt pond evolution and validating model parameterizations, yet current observational capabilities frequently miss these short-lived but crucial phases. Airborne campaigns have demonstrated..."*

- Section 5.2: Possibly redundant—merge with earlier priorities.

We thank the Reviewer for this note. We feel Section 5.2 builds on 5.1 (dedicated to the identification of knowledge gaps), but is intended to address a different purpose within the manuscript architecture: it synthesizes implications beyond the description content of Section 3-4 (since Section 5 is dedicated to Discussion). We have made an effort to clarify this in the text, and we have also addressed the Reviewer note on the possible redundancy.

- L794–800: Serves as general motivation, not melt pond–specific—condense.

We thank the Reviewer for this. We have condensed it. The new condensed revised motivation now reads as:

*'Instruments such as AMSR-E and SSMIS provide near-daily SIC data essential for climate monitoring and modeling. However, melt ponds significantly degrade SIC retrievals during the melt season by lowering surface emissivity, causing algorithms to misinterpret pond-covered ice as open water and systematically underestimate SIC (Cavalieri et al., 1990; Comiso and Kwok, 1996), with additional uncertainties arising from changing snow and ice surface properties. These limitations highlight the need to account for melt pond*

*influence in SIC products, particularly during summer, to improve the accuracy of sea ice monitoring and data assimilation.'*

- L801 onward: Repetitive—edit for conciseness.

We thank the Reviewer for noting this issue. To address this, the revised version now reads as:

*'Multiple studies quantify this impact: Kern et al. (2016) found that at 40% MPF, SIC is underestimated by 14-26% depending on the algorithm, though underestimation becomes negligible below 20% MPF. Kern et al. (2020) compared 10 passive microwave SIC products against MODIS and ship measurements, revealing significant melt-pond-related discrepancies. Zhao et al. (2021) analyzed 60 Arctic cruises (2006-2020) and found that at 50% MPF, SIC was underestimated by 7-20%, but MPF-based corrections significantly reduced this bias. These studies demonstrate that improved characterization of melt pond brightness temperature signatures and their explicit inclusion in retrieval algorithms are essential for advancing satellite-derived SIC accuracy during melt seasons.'*

References
Buckley, E. M. et al. (2023). Observing the evolution of summer melt on multiyear sea ice with ICESat-2 and Sentinel-2. The Cryosphere, 17(9), 3695–3719.
Fuchs, N. et al. (2024). Sea ice melt pond bathymetry reconstructed from aerial photographs using photogrammetry: a new method applied to MOSAiC data. The Cryosphere, 18(7), 2991–3015.

We thank the Reviewer for the suggested references. These studies are featured in the Appendix C, and are also now used more extensively in relation to bathymetry studies.

---

## Author Comment (AC2)

**Response to the comments of Reviewer 2**
We would like to thank Reviewer 2 for their thorough and valuable feedback. Their detailed comments allowed us to improve the structure of the manuscript and expand on several aspects, enhancing its completeness. We believe that the revised version, which integrates these suggestions, improves readability, provides stronger contextualization for the reader resulting in greater practical utility of the paper. We are grateful for the Reviewer's important contribution to strengthen  this article review. Our responses to the Reviewer's comments appear below in **blue text**, positioned directly beneath each respective comment.

Review of: egusphere-2025-4480
Observational data of Arctic Sea Ice Melt Ponds: a Systematic Review of Acquisition and Processing Approaches By Aparício et al. Summary The authors provide a review of earth observation (EO) methods used in the study of sea ice melt pond properties. Systematic melt pond property observations using EO are challenging and there are no high quality observations (e.g., ECV or CDR type datasets) despite importance for observing sea ice evolution, making proxy estimates such as albedo or light transmittance, and for predicting sea ice conditions. The paper requires some improvements, as per the major and minor comments provided here, and by others, before consideration for publication in The Cryosphere.

**Major Comments**
1. Given that the paper is a synthesis, the abstract should contain some statements relating to the outcomes of the synthesis. As it is, there only a statement about EO data gaps, along with an outline of the paper's intentions and some motivation statements.

We thank the Reviewer for this feedback. We agree that the abstract should better reflect the key outcomes from our synthesis rather than giving a stronger focus on the intentions and motivations. Following Reviewers' feedback, the abstract now provides the reader with a clearer understanding of what this synthesis revealed about melt ponds observations and it reads as follows:

'*This review synthesizes current methods for acquiring and processing Earth Observation (EO) data relevant to Arctic sea ice melt ponds (MPs), pools of meltwater that form on the ice surface during the polar summer. By reducing albedo, MPs amplify the ice–albedo feedback and alter the sea ice surface energy budget, exerting a strong influence on the Arctic climate system. Despite their importance, melt pond parameterizations remain underdeveloped in many sea ice models, and robust observational records are essential for improving sea ice predictions in a rapidly changing polar environment.*

*We review the main EO methods used in MP studies, including active and passive optical sensors (multispectral and LiDAR), and microwave instruments (synthetic aperture radar, radiometers, radar altimetry, and scatterometers). We synthesize melt pond signatures across the electromagnetic spectrum, distill the underlying physical mechanisms governing sensor responses, and outline the strengths and limitations of each sensor type. In situ observations from field campaigns, together with key processing techniques, are discussed alongside available MP datasets from satellite missions and ground-based campaigns.*

*Our synthesis reveals that optical systems currently dominate pan-Arctic melt pond fraction (MPF) products despite their limitations (e.g. light availability and cloud presence), while microwave systems face fundamental challenges despite their all-weather capabilities. Data processing has evolved from statistical approaches to spectral unmixing, physics-based algorithms, and machine-learning techniques. Intercomparison of pan-Arctic datasets reveals discrepancies during peak melt season and pronounced regional variability. Trade-offs exist between pan-Arctic datasets, which provide continuous monitoring at coarser resolutions, and high-resolution regional products offering meter-scale accuracy but limited coverage.*

*EO data gaps remain a major challenge: while major field campaigns have provided temporal and spatial snapshots of Arctic conditions they offer limited geographic coverage. Conversely, satellite remote sensing provides pan-Arctic coverage but validation datasets remain concentrated in coastal areas and first-year ice, leaving central Arctic and multi-year ice zones less well documented. Temporal constraints hinder capturing rapid melt transitions, and the lack of standardized classification and validation limits algorithm development and intercomparisons.*

*This review provides the first side-by-side overview of melt pond datasets, sensors, processing approaches, and field campaigns, highlighting spatio-temporal patterns of in situ coverage and enabling direct comparison of coverage, uncertainties, and accessibility that are otherwise scattered across sources. By compiling existing datasets and methods, identifying knowledge gaps and outlining priority research needs, this review paper provides a state-of-the-art review of melt pond observations, designed to support refinement of parameterizations and the development of multi-modal modeling approaches, crucial for closing observational gaps and advancing the understanding and prediction of Arctic change".*

2. It would be advantageous for terminology relating to EO to be tightened up. The authors are focused on what they refer to as the "main" EO methods, which appear to be airborne and satellite-based remote sensing. But in section 3, EO is also used to describe some in situ methods to, which is fine since buoys, weather stations, etc. are EO tools too. The term EO is dropped for remote sensing in the main text – is that distinction intended?

We appreciate that the Reviewer has highlighted this important terminological inconsistency, which can be potentially confusing. We have revised the manuscript to establish clear and consistent terminology throughout. We now explicitly define our scope in the introduction, we use '*Earth observation*' as the broader term encompassing all observational methods for monitoring melt ponds, including both remote sensing (spaceborne and airborne platforms) and in situ measurements (ship-based observation and field campaigns). When we refer specifically to satellite and airborne sensors we use '*remote sensing*' or spaceborn/airborne observations to be precise.

The term 'main EO methods', in our original text referred to the primary remote sensing approaches (optical & microwave sensors on satellite and airborne platforms), but we acknowledge this was unclear given that Section 3.2 discusses in situ methods, which are also forms of Earth observation.

For clarity and consistency, we have restructured Section 3 with clearer headings that distinguish: (3.1) '*Spaceborne observation*' for satellite remote sensing, (3.2) '*In situ and field campaign observation*' for ground-based and ship-based measurements , and (3.3) '*Post-processing techniques*' which apply to both. We believe that these revisions (which are listed below) eliminate the terminological ambiguity and make our scope and distinction clear throughout the manuscript.

Regarding the specific revision made:
- The introduction now explicitly defines scope and terminology
- Section 3 title is now revised to: '*Earth observation methods for melt pond detection and monitoring*'
- Section 3.1 is revised to '*Spaceborne observations*'
- Section 3.2 is revised to '*In situ and field campaign observations*'
- Finally, through the text we now use '*remote sensing*' when referring specifically to satellite/airborne sensors and 'Earth *observation*'

3. The paragraphs on Lines 59-74 needs improvement since some passages are hard the follow. More specific information is needed. How are melt ponds related to a shift in this Arctic sea ice spring predictability barrier? What is the barrier exactly? It can be stated better than "melt ponds …predict …September minimum" since melt ponds can't make predictions. More detail should be given on how melt ponds are parameterized in GCMs, especially given the sensitivity, and the overall theme of the paper. What is the link between GCM parameterizations and the Spring predictability barrier, as suggested on Lines 66-67? Overall, seasonal prediction and climate model projections, as they pertain to melt pond properties should be much better described. Otherwise, the motivation statements on Line 87 and Line 100 are not well backed-up.

In reference to the comments regarding Lines 59-74 and Lines 66-67, we  have described in more detail the influence of melt ponds on both seasonal prediction and climate predictions, and clarified all our statements. We apologise for confusion caused, and awkward wording (for instance that melt ponds themselves 'predict'). Furthermore, we have added more detail on how melt ponds are parametrized in GCMs, addressing all the comments here which we are grateful for. We have also modified statements on L87 and L100 and incorporated the changes as suggested.

4. The seasonal evolution of melt pond properties in Section 2, including Figure 3, needs to better differentiate between first-year ice and multiyear ice. There is mention of the influence of ice types in the context of topography, which is good, but there still needs to be seasonal stages in terms of melt pond formation and evolution, and expected melt pond albedo or fraction, defined by ice type. E.g., Stage 4 (freeze-up) rarely occurs for first-year sea ice since the ice melts away / disintegrates during summer.

We thank the Reviewer for this important observation. We agree that the seasonal evolution should be more explicitly differentiated between FYI and MYI, as these ice types follow distinct pathways. In the revised version, for instance, Section 2 has been updated to address ice-type-specific characteristics at each stage of evolution, including typical albedo ranges, defined by ice type. We have further updated our manuscript to note that  Stage 4 (freeze-up) is primarily a characteristic of MYI. Furthermore, this revision has also been reflected by updates to the Figure 3 caption and throughout the seasonal discussion.

5. It is generally hard to follow what satellite missions are current or past. Information about data products availability comes later in Table 2, but prior to that there are many missions discussed without enough detail regarding status.

We agree that addressing this lack of information will improve the clarity regarding current data availability for users. For this reason, we have revised Section 3.1 to include the operational status and temporal coverage for each satellite mission when first mentioned. Specifically, we now indicate: (i) mission status, i.e. whether it is operational or has been discontinued; (ii) the operational period and (iii) data availability status. These additions now appear before Table 2 in order to provide the readers with essential context about mission status as they read through the technical descriptions of each sensor system.

Moreover, this section was further revised to include an additional family of sensors (as discussed in the response to comment 6) and to address our response to comment L194 (in the minor comments section).

6. Optical (laser) altimetry is addressed but radar altimetry is not. Radar altimetry should be included.

Following the Reviewer's suggestion, we have given  radar altimetry its own dedicated section, following the same structure as other satellite missions  (i.e. it includes its way of operating, signature, mission examples, and data availability). We have included  relevant studies, such as Dawson et al. (2022), Landy et al. (2022) and Kwok et al. (2018), and additionally, for the sake of consistency, we also include this family of sensors in our Figure 4, dedicated to the overview of melt pond signatures and parameters across the different remote sensing systems. Refer to our response to comment L174 to view the updated image and caption.

7. In Section 3.2, more emphasis needs to be placed on the salient findings from these campaigns, in the context of melt pond property information retrieval from EO data, where possible. In some cases there are just descriptions of data collected and some observations made (e.g., ICE212). In other cases the descriptions aren't clear (e.g., MOSAiC contributions to spatiotemporal studies – what was learned?).

We have revised Section 3.2, to ensure a consistent structure and level of detail for each campaign, while specifically including how each of these campaigns advanced our understanding of melt ponds. In response to the Reviewer's suggestion, the revised section now includes the key findings from each campaign: MOSAiC (e.g., its contributions to understanding melt pond evolution patterns and relation between pond depth and the use of MOSAiC data for validation and algorithm improvement); SHEBA (e.g., its contributions to findings regarding the albedo-pond coverage relationship); ICE212 (e.g., its relevance to spectral differences between bright and dark melt ponds and the usage of data for melt pond validation purposes); HOTRAX (e.g., comprehensive documentation of pond fraction across a broad transect of ice conditions); IceBridge (e.g. the relevance of a multi-sensor approach to demonstrate the importance of sensor complementarity for characterising pond depth, extent and bathymetry); THINICE (e.g. how observations revealed pond formation and

drainage patterns, which provided new insights ingto melt pond behaviour under extreme weather).

Additionally, in response to Reviewer 1's suggestion, a new figure has been generated illustrating the tracks and locations of the campaigns.

**Minor Comments**

- L20: "… sensor type."

Fixed.

- L48-50: Recommended text: "(Right) True colour composite image of Arctic sea ice melt ponds from the Copernicus Sentinel-2 satellite (illustrating the large variability of melt pond, lead and open ocean reflectance). Acquired on 17th June 2024 off the Northeastern coast of Greenland.".

Added.

- L75: Add mention that SHEBA and MOSAiC are summarized later on.

The mention was added, referring to the section where it happens.

- L78: Change "struggle to" to "do not".

Changed.

- L93: "…melt pond studies…"

Was fixed (also on L434 and L635).

- L96: Refer to "… passive microwave" since radar used microwaves too.

Fixed.

- L103: Add a description of the basic outline of the sections of the paper.

The description was added at the end of Section1/Introduction as follows:

*'Section 2 reviews essential melt pond optical properties and their seasonal evolution. Section 3 covers Earth observation methods, including spaceborne sensors across the electromagnetic spectrum (3.1), in situ and field campaign observations (3.2), and post-processing techniques (3.3). Section 4 compiles available datasets (distinguishing between pan-Arctic satellite products (4.1) and high-resolution regional datasets (now Section 4.2\*). Section 5 discusses current knowledge gaps (5.1) outlines the role of melt ponds on future cryospheric research, and highlights potential future research trends (5.2). Section 6 concludes with key findings and emphasises the importance of enhanced melt pond observations for advancing Arctic climate research.'*

*This section number is different from the original manuscript due to changes on its structure.

- L114-118: Bare ice is referred to as both stable and changing (as melt progresses). Clarify.

We apologise for the confusion, the text was revised as follows:

*"Bare ice (light orange curve in Figure 2) has a substantial surface scattering layer that provides characteristically high reflectance in the blue-green region of the spectrum (0.75-0.8 at 450-500 nm) (Perovich et al., 2002). While the spectral signature remains relatively stable for this surface type, the magnitude of reflectance gradually decreases as melt progresses (Smith et al., 2022), primarily due to increasing snow grain size associated with rising liquid water content (Warren, 1982, 2019)."*

- L121: "radiometric" doesn't fit here.

Removed.

- L127: "ice surface features" should be changes to "ice properties" since ice surface and volume properties are important (as is described).

Changed.

- L134-135: Are the timings (by month) always appropriate when Arctic sea ice extends across about 40° latitude?

We expanded the text to address this question as follows:

*"Melt ponds typically begin forming in late spring to early summer and develop to cover large portions of the sea ice in the Arctic. They then deepen and expand, and refreeze by late Summer to early Autumn. The exact timing varies depending on latitude and regional climate conditions, with earlier onset taking place in lower-latitude seasonal ice zones."*

- L136: Here and in Figure 3 the melt onset should be "pond onset" since melt onset precedes the formation of ponds through meltwater accumulation. See, e.g.: https://doi.org/10.1038/s41597-023-02760-5 and L142: See major comment 4. The seasonal peak in pond fraction for first-year ice can be at the initial flooding stage, before drainage pathways open up. The Polashenski et al., 2012 paper shows this too.

We thank the reviewer for this observation. We have revised both the text and Figure 3 to reflect the ice-type-specific differences in pond fraction evolution. The text now clarifies that for first-year ice (FYI), the seasonal peak in pond fraction can occur during the initial flooding stage (Stage a), before drainage pathways open, whereas multi-year ice (MYI) typically exhibits its peak pond fraction later in the season after drainage and interconnection

(Stage c). Figure 3's caption and stage descriptions have been updated to explicitly describe the differences between FYI and MYI, with reference to Polashenski et al. (2012).

- L159: Delete Earth Observation since "(EO)" was defined earlier.

Deleted.

- L160: Be consistent with "in situ" being italicized or not, here and elsewhere.

Fixed

- L162: "... .considerations of associated …"

Corrected.

- L164: "addresses" (change the tense)

Fixed.

- L168: In the Section 3.1 heading "ponds" should be "pond"

Fixed.

- L174: Describe how the main applications were determined. In figure 4, "measured parameters" is used; are these the same thing? And L179: The Figure 4 caption is incomplete. In the figure, it is unclear if there is a difference between "detection of onset" and "timing of onset". It is odd that melt pond fraction is not mentioned.

We have updated the text to explain that the main applications were determined through our systematic literature review (which can be found in Appendix C - previously Section 4.2). To better clarify the distinction between parameters used and main applications we made considerable changes to Figure 4's layout and caption. This allows for a clearer and more intuitive understanding of the difference between parameters (i.e. directly measured physical quantities) and the main applications (i.e. higher-level derived applications). Figure 4 was further revised with the inclusion of radar altimetry, a clearer distinction between the signatures and applications rows, and the caption was updated.

[Figure]

*Figure 4: Overview of Earth observation methods for melt pond detection across the electromagnetic spectrum. The figure shows six sensor types (multispectral sensors, LiDARs, radiometers, scatterometers, radar altimeters, and synthetic aperture radars) organized by wavelength range (optical and microwave). For each sensor type, the figure presents: (top row) example satellite missions and operating wavelengths; (middle row) measured parameters and physical principles; (second last row) characteristic melt pond signatures and (bottom row) main applications.*

- L183: Name the spaceborne mission.

The mission has been named.

- L186: It should be "EM spectrum".

Updated.

- L194: "instruments" L183-205: Some of these missions are still operational, and others are defunct. Some clarity in this regard is needed. With regards to high-resolution imagery, is WV referenced as an example only? There have been others used; e.g. Webster et al. 2015 used data from the NTM satellites.

Regarding the operational status, we have added: operational dates and current status for all satellite missions mentioned in section 3.1, clearly indicating which missions are currently operational versus discontinued ones. These additions appear throughout section 3.1 providing readers with immediate context about mission status as they encounter each sensor system, as clarified in our answer to major comment 5.

About high-resolution imagery, the Reviewer is correct in that WV was referenced as an example (which we clarify in the text also), and we acknowledge that other high-resolution commercial satellites have been used in melt pond studies. The high-resolution imagery text was revised as follows:

*'For very high-resolution imagery, commercial satellites from the DigitalGlobe's WorldView (WV) constellation (2007–present, multiple satellites), which is an example that has enabled local analyses of melt pond properties and classification (Wright and Polashenski, 2018), providing finer spatial insights that complement the broader-scale insights provided by coarser-resolution data. Other high-resolution platforms, including governmental satellites, have also contributed to melt pond studies, as documented in the literature compiled in Appendix C (e.g., Webster et al., 2015).'*

- L222: State why Landsat is not suitable, and if that is for a specific Landsat mission or all of them.

The text was revised to explain the unsuitability statement regarding Landsat missions, which will include mentions of: (i) swath widths; (ii) limitations associated with cloudless opportunities, (iii) and Landast-7-specific additional limitations caused by the Scan Line Corrector failure in 2003, which created persistent data gaps and affected ~22% of each scene (as described in Markus et al., 2003).

- L223: Sentence "Additionally, for high resolution…" should be re-written for clarity. Are there no data for coastal waters, or limited data?

We have rewritten this sentence for clarity. Sentinel-2 MSI is restricted to coastal waters within 20 km of the shore, meaning that data coverage is limited rather than absent in these regions. The revised text now clearly states this spatial limitation and its implications for monitoring melt ponds on sea ice.

- L226: Remove "Similarly". The commercial restriction associated with those platforms is not similar to Sentinel-2.

Removed.

- L230-234: This should be made clearer. It is unclear how fresh snow mimicking melting ice leads to misclassification. What is classified/mis-classified? If it is melt pond fraction or pond/ice, aren't fresh snow or melting ice basically the same class (i.e., not melt pond)? It is also unclear why freeze ponds (refrozen ponds or freezing ponds?) lead to melt pond fraction overestimation.

We agree that the misclassification mechanism requires a clearer explanation. The revised text addresses both misclassification examples asked by the Reviewer, explaining how/why they can be misclassified (i.e. the spectral mechanism leading to the misclassification) and the result (which is overestimation in both cases).The revised text now reads as follows:

*"Fresh snow with high liquid water content exhibits increased NIR absorption similar to wet melting ice, making both surfaces spectrally similar to shallow or bright melt ponds in certain band combinations. When classification algorithms rely heavily on NIR reflectance characteristics, these wet snow or melting ice surfaces can be incorrectly classified as melt ponds, leading to MPF overestimation (Istomina et al., 2025). Similarly, refrozen melt ponds (ice-lidded ponds) retain low NIR reflectance similar to liquid melt ponds despite their frozen state, as the thin ice lid does not substantially alter the spectral signature. Classification algorithms based on reflectance alone cannot distinguish between liquid and ice-covered ponds, resulting in ice-lidded ponds being classified as active melt ponds and thus contributing to MPF overestimation, particularly during early-season refreezing events and late-season freeze-up (Xiong and Ren, 2023)."*

- L240: Sentence on LiDAR should be "LiDAR laser pulses are composed of photons typically emitted at one or both of these wavelengths.".

Updated.

- L245-248: Mention is made of pond depth and presence, in terms of research. But only pond depth is mentioned in relation to operational products. What about presence?

The parameter 'pond depth' received particular attention since, amongst the multiple pan-Arctic data products on the presence of melt ponds or MPF (as summarized for instance in Table 2), none focused on depths. Regarding ICESat-2 mission, its capabilities remain at research level, with no operational data products providing automated melt pond presence classification or depth measurements. With regards to presence, we revised the text, to acknowledge this distinct difference on data availability and operational products, while also inviting the reader to consult Appendix C, showcasing numerous studies involving the development of products.

- L249: "…consist of laser penetration…"

Fixed.

- L249-252: Add detail on how melt pond presence affects existing ICESat-2 data products for sea ice, and what those products are, if known. Are melt pond areas masked out, or generally unreliable in sea ice data products during spring/summer conditions?

The revised text now addresses how the presence of melt ponds affects ICESat-2 data products and further discusses the limitations of this type of sensor. Specifically, it now refers to relevant ICESat-2 data products (e.g., ATL03, ATL10) and explains how melt pond-covered areas are treated in sea-ice products during the spring and summer seasons, following, for example, Tilling et al. (2020) and Buckley et al. (2023).

- L254: Add a comma after "particles".

Added.

- L262-264: The statement about liquid water in snow needs correcting. Snow effects do not create the low signature of melt ponds, since melt ponds are water bodies and their backscatter is dependent on wind/wave roughness.

We thank the Reviewer for this remark. The text has been revised to focus exclusively on melt ponds, and it clarifies that their lower backscatter under calm conditions is primarily controlled by specular reflection from the smooth water surface, with the returned signal being sensitive to wind- and wave- induced roughness (in addition to radar configuration).

- L272: Clarify what you mean by early melt detections, in the context of X-band suitability over other frequencies.

The clarification has been added as follows:

*'Comparatively, X-band SAR systems like TerraSAR-X (9.65 GHz, 3.1 cm) provide higher spatial resolution (0.25-40 m) and enhanced sensitivity to fine-scale surface features compared to C-band (5.6 cm) or L-band (23.5 cm) systems. This is because radar backscatter is most sensitive to surface roughness features with scales comparable to the wavelength. For this reason X-band can detect smaller-scale surface changes such as initial ponding, and subtle surface roughness variations that characterize melt onset and early pond formation (Fig. 5), making it particularly suitable for detecting the initial stages of melt pond development (Kern et al., 2010; Fors et al., 2017).'*

- L274: Correct "melt onset" if necessary. I.e., if this is pond onset.

Corrected.

- L275: Change "on" to "for" and make "signatures" to "signature".

Corrected.

- L297: Get rid of the empty space. Also it should be "… poorly distinguishable from smooth sea ice.".

Corrected.

- L301: Provide detail about the swath widths. Smaller than what?

We have provided the actual swath widths adding clarification on size comparison.

- L304-306: The statement about X-band is hard to follow. Optimal performance for deriving what melt pond related information? MYI monitoring of what melt pond related information?

We revised the text in order to improve its readability while clarifying the two points, as follows:

*'The optimal incidence angle for X-band melt pond observations depends on the specific application and ice type. For melt pond fraction (MPF) retrieval on first-year ice, Fors et al. (2017) found optimal performance at 29-40° incidence angles using TerraSAR-X single-pol data. For multi-year ice, Han et al. (2016) demonstrated that TerraSAR-X dual-pol data performs best at 20-30° incidence angles for discriminating melt ponds from surrounding ice and monitoring pond evolution.'*

- L306-307: Backscatter should not increase due to specular reflection.

Corrected.

- L316: "pond onset"

Corrected.

- L311-322 and 311-313: It is hard to follow what melt pond properties are being observed/retrieved here. E.g., optimal retrieval of what melt pond information?

We have restructured this paragraph to explicitly state what is being retrieved.

- L326 and L350: The section on scatterometers should be shortened since most of the current focus is on melt onset, which precedes pond onset. Focus should be placed on melt pond properties. As for scatterometers, the discussion of melt effects in this section should be shortened and more emphasis placed specifically on melt pond related properties.

The scatterometer discussion was condensed to focus on melt pond properties as suggested.

- L376: Low penetration depth also applies to scatterometers and SARs since they operate in the microwave range. It should be noted above too.

We agree on its importance, and we added this clarification earlier in the SAR section (around L265-270) as follows:

*"It should be noted that low penetration depth into liquid water (~1 mm at frequencies ≥6 GHz) is a fundamental constraint across all microwave sensors operating in these frequency ranges, including SAR, scatterometers, and radiometers (Ulaby et al., 1986). This physical limitation prevents sensors from distinguishing seawater from meltwater in ponds and affects all microwave-based melt pond observations."*

We then referenced this earlier statement in the radiometer section (L376) rather than introducing it for the first time there.

- L390: Provide some detail on what criteria were used to include these campaigns, and (potentially) exclude others that have studied melt pond properties and detection techniques.

The criteria were included when introducing Table 1 as follows:

*"Table 1 summarizes field campaigns that contributed significant melt pond observational data, selected based on the following criteria: (1) campaigns that acquired spatially extensive in situ melt pond measurements (coverage, depth, albedo) or high-resolution imagery suitable for melt pond mapping; (2) datasets that have been widely used in the literature for algorithm development or validation (as documented in Appendix C); and (3) multi-instrument campaigns providing complementary observations relevant to remote sensing validation. Campaigns focused solely on ice thickness, oceanographic, or atmospheric measurements without melt pond-specific data collection were excluded. This selection represents the most significant melt pond observational efforts but is not exhaustive of all Arctic field campaigns."*

- L488: "named"

Corrected.

- L506: "that geophysical inversions" doesn't make sense. Please clarify.

The revised sentence now reads as follows:

*"Microwave data processing employs distinct methodologies that include 1) geophysical inversions to retrieve surface properties, 2) conversion of backscatter coefficients to MPF and empirical relationships, and 3) correlations between co-polarization ratios and compact polarization metrics (Yackel and Barber, 2000; Scharien et al., 2014; Li et al., 2017)"*

- L514: In Figure 8, there is mention of compact polarization but there was no mention of that in the earlier section on SAR approaches.

We have updated an earlier section (SAR section, around L262-290) to include a mention of compact polarization.

- L515-517: Some of the mentioned studies focus on melt pond fraction prediction, where spring MPF is predicted from winter SAR imagery, e.g., Howell et al., 2020. Others focus on direct estimation of MPF, Tanaka and Scharien, 2022. This should be clarified here, and in the earlier SAR section.

In order to clarify this important distinction, the 515-517 text was revised as follows:

*"Since 2014, two distinct approaches for MPF retrieval using multiple sensors have emerged: (1) direct estimation methods that use correlation and regression to relate coincident or near-simultaneous microwave signatures to optical-based MPF (Mäkynen et al., 2014; Tanaka et al., 2016; Fors et al., 2017; Scharien et al., 2017; Ramjan et al., 2018; Tanaka and Scharien, 2022), and (2) predictive methods that forecast future MPF from earlier-season SAR data, such as predicting spring/summer pond conditions from winter SAR imagery (Howell et al., 2020). These fusion methods link microwave and optical-based MPF and are well-suited for integration since they are essentially empirical mappings that can incorporate information from multiple sensors."*

Given the relevance of this clarification, it will also be reflected in the earlier SAR section (around L287-290) with the following revised text:

*"While most SAR studies focus on contemporaneous melt pond detection during the melt season, recent work has demonstrated the potential for predictive applications, using winter and early spring SAR characteristics to forecast subsequent melt pond development (Howell et al., 2020)."*

- L522: It would be helpful to know if more advanced techniques like ANN represent any increase in efficacy compared to traditional techniques.

The impacts on the efficacy of more advanced techniques were added as follows:

*"Deep learning methods have generally demonstrated improved performance over traditional techniques. For MPF retrieval, ANNs achieved RMSE values of 0.05-0.10 (Lee et al., 2020; Ding et al., 2020; Peng et al., 2022) compared to 0.10-0.18 for spectral unmixing approaches (Rösel et al., 2012; Zege et al., 2015). For classification tasks, CNNs achieved accuracies of 92-96% (Wright and Polashenski, 2018) versus 85-90% for traditional supervised classifiers (Fetterer et al., 2008; Divine et al., 2015). However, these improvements come at the cost of requiring large labeled training datasets and reducing skill in physical interpretation. The performance gains are most pronounced when training data are abundant and representative of diverse conditions."*

- L526: "converged"

Corrected.

- L530: MPF was defined earlier.

Fixed.

- L532: "melt pond" (singular).

Corrected.

- L580: Statistically significant "difference".

Corrected.

- L581: Use "MPFs".

Corrected.

- L598: "datasets"

Corrected.

- L631: Use "MPF".

Corrected.

- L698: "5. Discussion"

Corrected.

- L757: It is unclear why SAR-based methods are specifically identified as lagging in situ methods when, presumably, other satellite-based methods would be similarly lagging. This also comes up on L867.

We agree this phrasing was unclear. All satellite-based methods, including both optical and SAR, suffer from limited in situ validation data relative to the spatial and temporal scales needed for comprehensive algorithm development and validation. The revised text now clarifies that the distinction for SAR is not about lacking in situ data per se (which affects all remote sensing methods), but rather about SAR-based methods facing additional fundamental challenges in retrieving MPF at operational scales (L287-294), including: difficulty distinguishing melt ponds from open water due to similar backscatter characteristics; strong dependencies on environmental conditions and sensor parameters; and limited validated correlations beyond smooth FYI surfaces (Mäkynen et al., 2014; Scharien et al., 2014; Li et al., 2017).

While available in situ validation datasets are predominantly optical in nature (aerial photography, visual surveys, albedo measurements from field campaigns, Section 3.2, Table 1), this alone does not explain why SAR has not achieved the same level of operational MPF product development as optical methods (Table 2). The key point is that SAR faces signal interpretation challenges that have proven more difficult to resolve than those encountered by optical methods, despite SAR's theoretical advantages of all-weather, day-night capability.

The text at L757 and L867 has been revised to focus on this comparison between satellite-based approaches rather than on comparing satellite methods to in situ observations.

- L765: Use "MPF"

Corrected.

- L780: CAA was defined earlier. Use "CAA".

Corrected.

- L821-822: Elaborate what these schemes are, as well as schemes in general as they pertain to melt pond representation in models. This will make this section of the review more accessible to readers focused on in situ and satellite studies.

We appreciate this valuable recommendation, and agree that revising the text will make it more accessible. We have therefore revised the text in order to: (i) add the explanation of the meaning of melt pond 'schemes', (ii) add a brief description of the main categories of melt pond schemes, followed by (iii) the need for specification of explicit schemes and (iv) the importance behind of the choice of scheme regard impacts on models outputs.

- L850: Elaborate on why specifically pond depths, pond fraction, snow cover and type. Is ice topography and permeability less important?

Thank you for pointing this out. We revised the discussion of topography, noting that it can be observed remotely and validated, and added a remark highlighting other equally important parameters (e.g. sea-ice permeability) that do not meet these same criteria. The revised text now reads as:

*"Quantifying the physical processes driving melt pond fraction evolution remains a key challenge: understanding the interplay between the main ponds characteristics (pond depth and pond fraction) and the sea ice surface characteristics (e. g. snow cover and type, topography) would improve the comprehension of the links between surface physics and MPF variability. Emerging efforts to couple remote sensing observation with sea ice components of GCMs, hold the potential of improving and simulating pond onset, evolution and refreeze states. The significant influence of physical features like snow cover (Webster et al., 2015) or snow depth (Kim et al., 2018; Toyoda et al., 2024) underscores the importance coupling these parameters in a modeling framework to resolve inconsistencies between MPF datasets and improve melt onset detection. The parameters listed can be directly observed (either remotely or in field campaign) and validated, and can be used in model simulations. Other parameters, such ice permeability, while crucial for drainage processes (Polashenski et al., 2012), remain difficult to observe directly or validate systematically and must therefore be represented through process-based parameterizations."*

---

## Author Comment (AC3)

**Response to the comments of Commenter 1**

CC1 General Comment: This is an impressive piece of work and I am glad that it is already online as a citable discussion paper. I went through the manuscript - not necessarily with a reviewer's eye but with the intention to learn something new and to provide ideas how readability and usability of this publication could eventually be improved further. Perhaps one or more comments could assist here.

We would like to thank the Commenter for his very helpful insights, which helped us shape further the structure of the manuscript while taking additional care on different aspects that strengthen the manuscript. We are very grateful for the contribution, which we carefully addressed and took into consideration. Our responses to each point suggested by the Commenter appear below in **blue text**, positioned directly beneath each respective comment.

- L1-2: This is a very high-level title which, at least from my side, raises very high expectations. While you manage to come up with an impressive compilation of various data sets and also attempt to describe sensing and processing techniques I suggest to come up with an alternative structure to guide the reader a bit better. I was wondering whether it would make sense to begin - as you did - with the relevance of melt ponds for studying the Arctic climate and selected (!) studies where observations of melt ponds changed and/or influences our knowledge about Arctic summer sea ice conditions. I think in that section if would not matter whether you are referring to satellite, airborne or ground-based studies - simply because all have their different application areas and examples. Then, I would come up with the section where you describe the 3 main different observational tools: ground-based, airborne, spaceborne in 3 sub-sections. In each of these you would refer to the measurement technique, provide a list of the sensors and their characteristics, experiments / expeditions / satellites and at the end of each of these 3 subsections come up with the limitations of use / knowledge gaps / and room for improvement.
- Following this structure suggestion together with inputs. I would try to provide tables and/or appendices that are clearly linked to these 3 sub-sections, i.e. in-situ, air-borne and satellite.

We appreciate this thoughtful structural suggestion to organize by observational platform, and we have carefully considered this alternative organization. Section 1 (Introduction) was revised to better consolidate the motivations for melt ponds, while outlining the structure of the manuscript, and Section 2 (Melt Pond Properties and Seasonal Evolution) was updated to integrate their relevance and key findings. These two initial sections underwent significant changes, following the Commenter's suggestion (while also considering Reviewers 1 and 2, by incorporating an overview/synthesis from melt pond studies - which were previously examined in a dedicated Section 4.2, and are now part of Appendix C).

Following suggestions from Reviewers 1 and 2, we have improved the logical flow of Section 3 within its existing structure by clearly distinguishing between spaceborne observations (3.1), in situ/field campaigns (3.2), and processing techniques (3.3), while strengthening the links between them. Specifically, we have added a synthesis paragraph at the beginning of Section 3 that previews overarching challenges with forward references to where each is

addressed in detail for each sensor type. We have improved consistency in terminology (i.e., using EO as a broader term and 'remote sensing' when referring specifically to satellite/airborne observations, with explicit definitions in the introduction). We also enhanced cross-referencing between sections to make connections more explicit, particularly linking Section 3.1 (sensor capabilities) to Section 3.3 (processing techniques) and to Section 4 (datasets). This revision also ensures that the focus of this section stays on existing datasets. These changes directly address the reviewer's comment regarding the need for clearer linkages to the main text. We hope that the substantial structural revisions, including the platform-level organization emphasized by the reviewers, and the prioritization of physical measurement principles, ensure consistency with the reviewers' feedback while improving the overall flow and readability of the manuscript.

- I don't think it makes a lot of sense to try to distinguish between pan-Arctic and regional spaceborne applications because in the long run, applications such as from Sentinel-2 MSI might become pan-Arctic as well once there is enough coverage. I guess, if described properly, readers will understand that 10 km x 10 km large super-high resolution satellite images are not pan-ARctic and are not suitable for climate studies but - like the air-borne data sets - are perfect for algorithm development and evaluation.

We have considered this feedback alongside suggestions from Reviewers 1 and 2. While maintaining the pan-Arctic versus regional categorization for its current practical utility in distinguishing dataset characteristics and intended applications, we acknowledge the Commenter's important point about the evolving nature of these distinctions. We have added a clarifying statement noting that certain platforms currently categorized as regional (such as Sentinel-2 MSI) may achieve pan-Arctic coverage in the long term as data accumulation continues, and that this distinction should be understood as reflecting current data availability rather than fundamental sensor limitations. The revised text ensures that all spaceborne applications (such as S2 MSI-based products, which were also pointed out by the Commenter) are appropriately categorized, with clear descriptions of their spatial coverage, temporal extent, and suitability for different research applications (e.g., algorithm development and validation versus long-term climate studies).

- Figure 4: Signature row needs more information; mixes onset and mature melt pond existence; overly simplistic and partly misleading

We recognize that Figure 4 attempts to synthesize complex, stage-dependent signatures into a single schematic, inevitably simplifying the temporal evolution. To address this limitation, the text preceding Figure 4 has been revised to explicitly acknowledge (i) temporal variability between melt onset and mature pond stages, (ii) signature sensitivity to environmental conditions, and (iii) dependence on ice type (FYI versus MYI). Readers are also explicitly referred to Section 3.1, where these aspects are discussed in greater detail.

In response to the reviewer's concern, Figure 4 has been further revised to improve clarity and completeness: (i) clarification of onset detection applications was made within the image; (ii) radar altimetry has been added, (iii) the figure layout has been restructured to more clearly distinguish between *retrieved parameters* (i.e., directly measured physical

quantities) and *main applications* (i.e., higher-level derived uses) and (iv) the rows containing mission examples, retrieved parameters, signatures, and main applications, derived from our systematic literature review and supported by Appendix C (formerly Section 4.2), have been refined to ensure better separation and consistency across sensor types.

Finally, the figure caption has been revised to better contextualize the schematic nature of the figure and to clarify its intended scope and limitations.

[Figure]

*Figure 4: Overview of Earth observation methods for melt pond detection across the electromagnetic spectrum. The figure shows six sensor types (multispectral sensors, LiDARs, radiometers, scatterometers, radar altimeters, and synthetic aperture radars) organized by wavelength range (optical and microwave). For each sensor type, the figure presents: (top row) example satellite missions and operating wavelengths; (middle row) measured parameters and physical principles; (second last row) characteristic melt pond signatures and (bottom row) main applications.*

- L196: Add Lee et al., 2020

Added.

- L249-252: Clouds are an issue for ICESat-2; limited daily coverage may miss seasonal development steps

We thank the Commenter for this relevant note, which we added to the revised text.

- L254-256: Statement about microwave atmospheric independence not entirely correct; water vapor affects ~22 GHz and ~90 GHz; wind and salinity effects

The text was updated to correct microwave atmospheric independence, and addresses water vapor, wind-induced surface roughening, and water salinity effects.

- L295++: Be more specific about penetration depth (dry ice/snow only, not water; melt pond depth has no influence on radar backscatter); clarify "seasonality"; provide rule of thumb about wind-roughened vs. smooth pond detection

We expanded on this by clarifying the actual influences of penetration. The term "seasonality" was further clarified following the Commenter's suggestion. Additionally, we added guidance on the detection thresholds for wind-roughened versus smooth ponds.

- L360-369: Don't mix land surfaces and sea ice; concentrate on sea ice; mention sensors first, then applications

L360-369 was revised to concentrate on sea ice and maintained consistency with the other missions' descriptions.

- L378-380: Check The Cryosphere 2016 and 2020 for more recent literature on melt pond uncertainties in SIC retrieval; 2nd sentence needs reference

More recent literature from The Cryosphere (Kern et al., 2016, 2020) was included addressing melt pond uncertainties in SIC retrieval, and references were added to the previously uncited sentence.

- L489/490: Link does not work

Link updated.

- L500: Link back to earlier sensor descriptions; structural improvement needed

We added explicit links to clarify that this section presents the main technical approaches for processing melt pond observations (detailed in Appendix A), building on the sensor characteristics and signatures described in Section 3.1 (spaceborne observations) and Section 3.2 (in situ and field campaign observations). The section was restructured to establish clear connections between processing families (Figure 8) and sensor types and measured parameters introduced in Figure 4.

- L501: "Early approaches" doesn't make sense when citing 2015+

Changed.

- 505: Kern et al. 2016 did not develop the approach; should cite Tschudi et al., 2008 and Rösel et al., 2012

Corrected.

- L515-527: Combine methods and data sources (satellite sensors) in one go; clarify organizational logic

We added transitional text clarifying the organizational logic of the section and improving readability. The revised structure now presents methods and data sources within a sensor-specific framework progressing to multi-sensor integration and sensor-independent approaches, reflecting the methodological evolution of the field.

- L563/564: No systematic independent intercomparison exists; be careful about accuracy claims; dataset producers have biased views

We appreciate this note, and the text was revised addressing the dataset common challenges; the lack of a systematic independent intercomparison study; and mentioning that datasets with reported high accuracy were based primarily on validation by the data producers.

- L602: Refer to datasets by author team name unless there was an actual expedition

Updated.

- L602-615: Don't mix satellite, airborne, and ground-based observations in this section

L602-615 were restructured to group them by platform following the suggestion of the Commenter, improving readability.

- Section 4.2: Goes beyond review scope; should focus on datasets and processing approaches; much repetition of earlier content; place 1-2 key studies per dataset upfront

We have carefully considered the Commenter's suggestions and implemented a substantial revision. Specifically, Section 4.2 has been relocated to Appendix C, in alignment with Reviewer 1's feedback. Although this appendix extends beyond the immediate scope of the review, it compiles over 40 studies on melt ponds, providing valuable context that reinforces both the motivation and relevance of melt pond research. Furthermore, it offers insights into the methodological evolution within the field and aids in identifying future research directions, thereby establishing clear links across the manuscript. Throughout the text, readers are referred to this appendix for further consultation.

- L703-707: Doesn't this downgrade SHEBA and MOSAiC importance? Their role needs emphasis

The text was revised to ensure it does not unintentionally downplay these critical campaigns, while emphasizing their irreplaceable role in providing detailed measurements that cannot be obtained remotely, and their importance for melt pond studies. Moreover, following additional suggestions from Reviewers 1 and 2, this section was revised to better balance the level of description across campaigns and to explicitly highlight how each advanced our understanding of melt ponds by including their key findings.

- L757-762: Little hope to disentangle SAR signatures between melt-pond-free ice, melt ponds, and leads

Indeed this is a fundamental challenge, which has been now emphasized at L757-762

- L765: (1) SIA is physically correct quantity, not SIE; (2) Melt ponds don't influence SIE computation (>15% threshold); they influence SIA; this lack of SIC accuracy drove community to SIE over SIA

We appreciate this correction, which has been made in the revised version of the manuscript.

- L863-865: (1) Are commercial satellite images openly accessible? Coverage? (2) Emphasize the impressive amount of such data available; (3) These are NOT in-situ measurements—they require retrieval; communicate uncertainties; OIB validation is inter-comparison, not true validation

We addressed each point: satellite openness, coverage, licensing needs, the volume of available data were included. We explicitly noted that both airborne and satellite MPF products are retrieved values (not in situ measurements) requiring algorithms with associated uncertainties. We clarified that comparisons between satellite-based and airborne-derived MPF constitute inter-comparisons between two retrieval products at different spatial resolutions, not true validation against ground truth measurements.

- L885-890: Effort required not worth the outcome; better to have 2-3 datasets for intercomparison to assess uncertainty

The text has been revised to clarify that standardization is intended to *enable* efficient intercomparison of a small number of representative datasets (e.g., 2–3), to assess classification uncertainty, rather than focusing on a single dataset or exhaustive analyses.

- Appendix B: (1) Check names (TransArc may be in IceWatch); verify IceWatch availability to present; (2) Rename "PANGEA" dataset since multiple datasets are in PANGAEA database; (3) Check ship-based observation link (https://www.cen.uni-hamburg.de/en/icdc/data/cryosphere/seaiceparameter-shipobs.html); (4) Is list exhaustive? Why two appendices (B and C)? Consider combining

(1) Appendix B has been revised to correct and clarify dataset names. We have verified IceWatch availability, and added the recommended ship-based observation link. We thank the reviewer for this suggestion.
(2) PANGEA dataset name was revised.
(3) The list is not intended to be exhaustive but representative.
(4) Appendices B and C are retained because they serve different purposes: Appendix B summarizes observational datasets, while Appendix C lists research studies and applications, some of which use or led to the datasets in Appendix B.